# Transformed Low-Rank Parameterization Can Help Robust Generalization for Tensor Neural Networks

**Andong Wang**
RIKEN AIP
andong.wang@riken.jp

**Chao Li**
RIKEN AIP
chao.li@riken.jp

**Mingyuan Bai**
RIKEN AIP
mingyuan.bai@riken.jp

**Zhong Jin**
China University of Petroleum-Beijing at Karamay
zhongjin@cupk.edu.cn

**Guoxu Zhou**[*]
Guangdong University of Technology
gx.zhou@gdut.edu.cn

**Qibin Zhao**[*]
RIKEN AIP
qibin.zhao@riken.jp

## Abstract

Multi-channel learning has gained significant attention in recent applications, where neural networks with t-product layers (t-NNs) have shown promising performance through novel feature mapping in the transformed domain. However, despite the practical success of t-NNs, the theoretical analysis of their generalization remains unexplored. We address this gap by deriving upper bounds on the generalization error of t-NNs in both standard and adversarial settings. Notably, it reveals that t-NNs compressed with exact transformed low-rank parameterization can achieve tighter adversarial generalization bounds compared to non-compressed models. While exact transformed low-rank weights are rare in practice, the analysis demonstrates that through adversarial training with gradient flow, highly over-parameterized t-NNs with the ReLU activation can be implicitly regularized towards a transformed low-rank parameterization under certain conditions. Moreover, this paper establishes sharp adversarial generalization bounds for t-NNs with approximately transformed low-rank weights. Our analysis highlights the potential of transformed low-rank parameterization in enhancing the robust generalization of t-NNs, offering valuable insights for further research and development.

## 1 Introduction

Multi-channel learning is a task to extract representations from the data with multiple channels, such as multispectral images, time series, and multi-view videos, in an efficient and robust manner [24, 39, 60, 61]. Among the methods tackling this task, neural networks with t-product layers (t-NNs, see Eq. (5) for a typical example) [12, 36] came to the stage very recently with remarkable efficiency and robustness in various applications such as graph learning, remote sensing, and more [1, 11, 14, 32, 38, 39, 53, 58]. What distinguishes t-NNs from other networks is the inclusion of t-product layers, founded on the algebraic framework of tensor singular value decomposition (t-SVD) [19, 40, 60, 61]. Unlike traditional tensor decompositions, t-SVD explores the transformed low-rankness, i.e., the low-rank structure of a tensor in the transformed domain under an invertible linear transform [18]. The imposed transform in t-product layers provides additional expressivity to neural networks, while

---

[*]Qibin Zhao and Guoxu Zhou are the corresponding authors.

37th Conference on Neural Information Processing Systems (NeurIPS 2023).

the controllable transformed low-rank structure in t-NNs enables a flexible balance between model accuracy and robustness [36, 38, 53].

Despite the impressive empirical performance of t-NNs, the theoretical foundations behind their success remain unclear. The lack of systematic theoretical analysis hinders deeper comprehension and exploration of more effective applications and robust performance of t-NNs. Furthermore, the inclusion of the additional transform in t-NNs renders the theoretical analysis more technically challenging compared to existing work on general neural networks [29, 37, 43, 55]. To address this challenge, we establish for the first time a theoretical framework for t-NNs to understand both the standard and robust generalization behaviors, providing both theoretical insights and practical guidance for the efficient and robust utilization of t-NNs. Specifically, we address the following fundamental questions:

- *Can we theoretically characterize the generalization behavior of general t-NNs?* Yes. We derive the upper bounds on the generalization error for *general* t-NNs in both standard and adversarial settings in Sec. 3.
- *How does exact transformed low-rankness influence the robust generalization of t-NNs?* In Sec. 4.1, our analysis shows that t-NNs with *exactly* transformed low-rank weights exhibit lower adversarial generalization bounds and require fewer samples, highlighting the benefits of transformed low-rank weights in t-NNs for improved robustness and efficiency.
- *How does adversarial learning of t-NNs affect the transformed ranks of their weight tensors?* In Sec. 4.2, we deduce that weight tensors tend to be of transformed low-rankness *approximately* for highly over-parameterized t-NNs with ReLU activation under adversarial training with gradient flow.
- *How is robust generalization impacted by approximately transformed low-rank weight tensors in t-NNs?* In Sec. 4.3, we establish sharp adversarial generalization bounds for t-NNs with *approximately* transformed low-rank weights by carefully bridging the gap with exact transformed low-rank parameterization. This finding again underscores the importance of incorporating transformed low-rank weights as a means to enhance the robustness of t-NNs.

## 2 Notations and Preliminaries

In this section, we introduce the notations and provide a concise overview of t-SVD and t-NNs, which play a central role in the subsequent analysis.

**Notations.** We use lowercase, lowercase boldface, and uppercase boldface letters to denote scalars, *e.g.*, $a \in \mathbb{R}$, vectors, *e.g.*, $\mathbf{a} \in \mathbb{R}^m$, and matrices, *e.g.*, $\mathbf{A} \in \mathbb{R}^{m \times n}$, respectively. Following the standard notations in Ref. [19], a 3-way tensor of size $d \times 1 \times \mathsf{c}$ is also called a *t-vector* and denoted by underlined lowercase, *e.g.*, $\underline{\mathbf{x}}$, whereas a 3-way tensor of size $m \times n \times \mathsf{c}$ is also called a *t-matrix* and denoted by underlined uppercase, *e.g.*, $\underline{\mathbf{X}}$. We use a t-vector $\underline{\mathbf{x}} \in \mathbb{R}^{d \times 1 \times \mathsf{c}}$ to represent a multi-channel example, where $\mathsf{c}$ denotes the number of channels and $d$ is the number of features for each channel.

Given a matrix $\mathbf{A} \in \mathbb{R}^{m \times n}$, its Frobenius norm (F-norm) and spectral norm are defined as $\|\mathbf{A}\|_{\mathrm{F}} := \sqrt{\sum_{i=1}^{\min\{m,n\}} \sigma_i^2}$ and $\|\mathbf{A}\| := \max_i \sigma_i$, respectively, where $\sigma_i$, $i = 1, \cdots, \min\{m, n\}$ are its singular values. The *stable rank* of a non-zero matrix $\mathbf{A}$ is defined as the squared ratio of its F-norm and spectral norm $r_{\mathrm{stb}}(\mathbf{A}) := \|\mathbf{A}\|_{\mathrm{F}}^2 / \|\mathbf{A}\|^2$. Given a tensor $\underline{\mathbf{T}}$, define its $l_p$-norm and F-norm respectively as $\|\underline{\mathbf{T}}\|_{l_p} := \|\mathrm{vec}(\underline{\mathbf{T}})\|_{l_p}$, and $\|\underline{\mathbf{T}}\|_{\mathrm{F}} := \|\mathrm{vec}(\underline{\mathbf{T}})\|_{l_2}$, where $\mathrm{vec}(\cdot)$ denotes the vectorization operation of a tensor [21]. Given $\underline{\mathbf{T}} \in \mathbb{R}^{m \times n \times \mathsf{c}}$, let $\underline{\mathbf{T}}_{:,:,i}$ denote its $i^{\mathrm{th}}$ frontal slice. The inner product between two tensors $\underline{\mathbf{A}}, \underline{\mathbf{B}}$ is defined as $\langle \underline{\mathbf{A}}, \underline{\mathbf{B}} \rangle := \mathrm{vec}(\underline{\mathbf{A}})^{\top} \mathrm{vec}(\underline{\mathbf{B}})$. The frontal-slice-wise product of two tensors $\underline{\mathbf{A}}, \underline{\mathbf{B}}$, denoted by $\underline{\mathbf{A}} \odot \underline{\mathbf{B}}$, equals a tensor $\underline{\mathbf{T}}$ such that $\underline{\mathbf{T}}_{:,:,i} = \underline{\mathbf{A}}_{:,:,i}\underline{\mathbf{B}}_{:,:,i}$, $i = 1, \cdots, \mathsf{c}$ [19]. We use $|\cdot|$ as the absolute value for a scalar and cardinality for a set. We use $\circ$ to denote the function composition operation. Additional notations will be introduced upon their first occurrence.

### 2.1 Tensor Singular Value Decomposition

The framework of tensor singular value decomposition (t-SVD) is based on the t-product under an invertible linear transform $M$ [18]. In recent studies, the transformation matrix $\mathbf{M}$ defining the transform $M$ is *restricted to be orthogonal* [50] for better properties, which is also followed in this

paper. Given any *orthogonal matrix* $\mathbf{M} \in \mathbb{R}^{c \times c}$, define the associated linear transform $M(\cdot)$ with its inverse $M^{-1}(\cdot)$ on any $\underline{\mathbf{T}} \in \mathbb{R}^{m \times n \times c}$ as

$$M(\underline{\mathbf{T}}) := \underline{\mathbf{T}} \times_3 \mathbf{M}, \quad \text{and} \quad M^{-1}(\underline{\mathbf{T}}) := \underline{\mathbf{T}} \times_3 \mathbf{M}^{-1}, \tag{1}$$

where $\times_3$ denotes the tensor matrix product on mode-3 [18].

**Definition 1** (t-product [18]). *The t-product of any $\underline{\mathbf{A}} \in \mathbb{R}^{m \times n \times c}$ and $\underline{\mathbf{B}} \in \mathbb{R}^{n \times k \times c}$ under transform $M$ in Eq. (1) is denoted and defined as $\underline{\mathbf{A}} *_M \underline{\mathbf{B}} = \underline{\mathbf{C}} \in \mathbb{R}^{m \times k \times c}$ such that $M(\underline{\mathbf{C}}) = M(\underline{\mathbf{A}}) \odot M(\underline{\mathbf{B}})$ in the transformed domain. Equivalently, we have $\underline{\mathbf{C}} = M^{-1}(M(\underline{\mathbf{A}}) \odot M(\underline{\mathbf{B}}))$ in the original domain.*

**Definition 2** ($M$-block-diagonal matrix). *The $M$-block-diagonal matrix of any $\underline{\mathbf{T}} \in \mathbb{R}^{m \times n \times c}$, denoted by $\widetilde{\mathbf{T}}_M$, is the block diagonal matrix whose diagonal blocks are the frontal slices of $M(\underline{\mathbf{T}})$:*

$$\widetilde{\mathbf{T}}_M := \mathtt{bdiag}(M(\underline{\mathbf{T}})) := \begin{bmatrix} M(\underline{\mathbf{T}})_{:,:,1} & & & \\ & M(\underline{\mathbf{T}})_{:,:,2} & & \\ & & \ddots & \\ & & & M(\underline{\mathbf{T}})_{:,:,c} \end{bmatrix} \in \mathbb{R}^{mc \times nc}.$$

In this paper, we also follow the definition of t-transpose, t-identity tensor, t-orthogonal tensor, and f-diagonal tensor given by Ref. [18], and thus the t-SVD is introduced as follows.

**Definition 3** (t-SVD, tubal rank [18]). *Tensor Singular Value Decomposition (t-SVD) of $\underline{\mathbf{T}} \in \mathbb{R}^{m \times n \times c}$ under the invertible linear transform $M$ in Eq. (1) is given as follows*

$$\underline{\mathbf{T}} = \underline{\mathbf{U}} *_M \underline{\mathbf{S}} *_M \underline{\mathbf{V}}^\top, \tag{2}$$

*where $\underline{\mathbf{U}} \in \mathbb{R}^{m \times m \times c}$ and $\underline{\mathbf{V}} \in \mathbb{R}^{n \times n \times c}$ are t-orthogonal, and $\underline{\mathbf{S}} \in \mathbb{R}^{m \times n \times c}$ is f-diagonal. The tubal rank of $\underline{\mathbf{T}}$ is defined as the number of non-zero tubes of $\underline{\mathbf{S}}$ in its t-SVD in Eq. (2), i.e., $r_t(\underline{\mathbf{T}}) := |\{i \mid \underline{\mathbf{S}}(i,i,:) \neq \mathbf{0}, i \leq \min\{m,n\}\}|$.*

For any $\underline{\mathbf{T}} \in \mathbb{R}^{m \times n \times c}$ with the tubal rank $r_t(\underline{\mathbf{T}})$, we have following relationship between its t-SVD and the matrix SVD of its $M$-block-diagonal matrix [26, 50]:

$$\underline{\mathbf{T}} = \underline{\mathbf{U}} *_M \underline{\mathbf{S}} *_M \underline{\mathbf{V}}^\top \Leftrightarrow \widetilde{\mathbf{T}}_M = \widetilde{\mathbf{U}}_M \cdot \widetilde{\mathbf{S}}_M \cdot \widetilde{\mathbf{V}}_M^\top, \quad \text{and} \quad c \cdot r_t(\underline{\mathbf{T}}) \geq \text{rank}(\widetilde{\mathbf{T}}_M). \tag{3}$$

As the $M$-block-diagonal matrix $\widetilde{\mathbf{T}}_M$ is defined after transforming tensor $\underline{\mathbf{T}}$ from the original domain to the transformed domain, the relationship $c \cdot r_t(\underline{\mathbf{T}}) \geq \text{rank}(\widetilde{\mathbf{T}}_M)$ indicates that the tubal rank can be chosen as a measure of transformed low-rankness [26, 50].

## 2.2 Neural Networks with t-Product Layer (t-NNs)

In this subsection, we will introduce the formulation of the t-product layer in t-NNs, which is designed for multi-channel feature learning.

**Multi-channel feature learning via t-product.** Suppose we have a multi-channel example represented by a t-vector $\underline{\mathbf{x}} \in \mathbb{R}^{d \times 1 \times c}$, where $c$ is the number of channels and $d$ is the number of features. We define an $L$-layer t-NN feature extractor $\mathbf{f}(\underline{\mathbf{x}})$, to extract $d_L$ features for each channel of $\underline{\mathbf{x}}$:

$$\mathbf{f}(\underline{\mathbf{x}}) = \mathbf{f}^{(L)}(\underline{\mathbf{x}}); \quad \mathbf{f}^{(l)}(\underline{\mathbf{x}}) = \sigma(\underline{\mathbf{W}}^{(l)} *_M \mathbf{f}^{(l-1)}(\underline{\mathbf{x}})), \, l = 1, \cdots, L; \quad \mathbf{f}^{(0)}(\underline{\mathbf{x}}) = \underline{\mathbf{x}}, \tag{4}$$

where the $l$-th layer $\mathbf{f}^{(l)}$ first conducts t-product with weight tensor (t-matrix) $\underline{\mathbf{W}}^{(l)} \in \mathbb{R}^{d_l \times d_{l-1} \times c}$ on the output of the $(l-1)$-th layer as multi-channel features[2] $\mathbf{f}^{(l-1)}(\underline{\mathbf{x}}) \in \mathbb{R}^{d_{l-1} \times 1 \times c}$ to obtain a $(d_l \times 1 \times c)$-dimensional representation and then uses the entry-wisely ReLU activation[3] $\sigma(x) = \max\{x, 0\}$ for nonlinearity.

**Remark.** *Unlike Refs. [36], [32] and [53] whose nonlinear activation is performed in the transformed domain, the t-NN model in Eq. (4) considers the nonlinear activation in the original domain and hence is consistent with traditional neural networks.*

---

[2]For simplicity, let $d_0 = d$ by treating the input example $\underline{\mathbf{x}}$ as the 0-th layer $\mathbf{f}^{(0)}$.

[3]Although we consider ReLU activation in this paper, most of the main theoretical results (*e.g.*, Theorems 3, 5, 6, 12, and 14) can be generalized to general Lipschitz activations with slight modifications in the proof.

By adding a linear classification module with weight $\mathbf{w} \in \mathbb{R}^{cd_L}$ after the feature exaction module in Eq. (4), we consider the following t-NN predictor whose sign can be utilized for binary classification:

$$f(\mathbf{x}; \underline{\mathbf{W}}) := \mathbf{w}^\top \mathtt{vec}(\mathbf{f}^{(L)}(\mathbf{x})) \in \mathbb{R}. \qquad (5)$$

Let $\underline{\mathbf{W}} := \{\underline{\mathbf{W}}^{(1)}, \cdots, \underline{\mathbf{W}}^{(L)}, \mathbf{w}\}$ be the collection of all the weights[4]. With a slight abuse of notation, let $\|\underline{\mathbf{W}}\|_{\mathrm{F}} := \sqrt{\|\mathbf{w}\|_2^2 + \sum_{l=1}^{L} \|\underline{\mathbf{W}}^{(l)}\|_{\mathrm{F}}^2}$ denote the Euclidean norm of all the weights. The function class of general t-NNs whose weights are bounded in the Euclidean norm is defined as

$$\mathfrak{F} := \left\{ f(\mathbf{x}; \underline{\mathbf{W}}) \mid \|\mathbf{w}\|_2 \le B_w, \quad \|\underline{\mathbf{W}}^{(l)}\|_{\mathrm{F}} \le B_l, \quad l = 1, \cdots, L \right\}, \qquad (6)$$

with positive constants $B_w$ and $B_l$, $l = 1, \cdots, L$. Let $B_{\underline{\mathbf{W}}} := B_w \prod_{l=1}^{L} B_l$ for simplicity.

## 3 Standard and Robust Generalization Bounds for t-NNs

This section establishes both the standard and robust generalization bounds for any t-NN $f \in \mathfrak{F}$.

### 3.1 Standard Generalization for General t-NNs

Suppose we are given a training multi-channel dataset $S$ consisting of $N$ example-label pairs $\{(\mathbf{x}_i, y_i)\}_{i=1}^{N} \subset \mathbb{R}^{d \times 1 \times c} \times \{\pm 1\}$ *i.i.d.* drawn from an underlying data distribution $P_{\mathbf{x},y}$.

**Assumption 1.** *Every input example $\mathbf{x} \in \mathbb{R}^{d \times 1 \times c}$ has an upper bounded F-norm, i.e., $\|\mathbf{x}\|_{\mathrm{F}} \le B_x$, where $B_x$ is a positive constant.*

When a loss function $\ell(f(\mathbf{x}_i), y_i)$ is considered as the measure of the classification quality, we define the empirical and population risk for any $f \in \mathfrak{F}$ as $\hat{\mathcal{L}}(f) := N^{-1} \sum_{i=1}^{N} \ell(f(\mathbf{x}_i), y_i)$ and $\mathcal{L}(f) := \mathbb{E}_{P_{(\mathbf{x},y)}} [\ell(f(\mathbf{x}), y)]$, respectively. Similar to Ref. [30], we make assumptions on the loss as follows.

**Assumption 2.** *The loss $\ell(h(\mathbf{x}), y)$ can be expressed as $\ell(h(\mathbf{x}), y) = \exp(-\mathfrak{f}(yh(\mathbf{x}))$ for any t-NN $h \in \mathfrak{F}$, such that:*

*(A.1)* *the range of loss $\ell(\cdot, \cdot)$ is $[0, B]$, where $B$ is a positive constant;*

*(A.2)* *function $\mathfrak{f} : \mathbb{R} \to \mathbb{R}$ is $C^1$-smooth;*

*(A.3)* *$\mathfrak{f}'(x) \ge 0$ for any $x \in \mathbb{R}$;*

*(A.4)* *there exists $b_{\mathfrak{f}} \ge 0$ such that $x\mathfrak{f}'(x)$ is non-decreasing for $x \in (b_{\mathfrak{f}}, +\infty)$, and the derivative $x\mathfrak{f}'(x) \to +\infty$ as $x \to +\infty$;*

*(A.5)* *let $\mathfrak{g} : [\mathfrak{f}(b_{\mathfrak{f}}), +\infty) \to [b_{\mathfrak{f}}, +\infty)$ be the inverse function of $\mathfrak{f}$ on the domain $[b_{\mathfrak{f}}, +\infty)$. There exist $b_{\mathfrak{g}} \ge \max\{2\mathfrak{f}(b_{\mathfrak{f}}), \mathfrak{f}(2b_{\mathfrak{f}})\}$ and $K \ge 1$, such that $\mathfrak{g}'(x) \le K\mathfrak{g}'(\theta x)$ and $\mathfrak{f}'(y) \le K\mathfrak{f}'(\theta y)$ for any $x \in (b_{\mathfrak{g}}, +\infty), y \in (\mathfrak{g}(b_{\mathfrak{g}}), +\infty)$ and $\theta \in [1/2, 1)$.*

Assumption *(A.1)* is a natural assumption in generalization analysis [3, 59], and Assumptions *(A.2)*-*(A.5)* are the same as Assumption (B3) in Ref. [30]. According to Assumption *(A.2)*, the loss function $\ell(\cdot, \cdot)$ satisfies the $L_\ell$-Lipschitz continuity

$$|\ell(h(\mathbf{x}_1), y_1) - \ell(h(\mathbf{x}_2), y_2)| \le L_\ell |y_1 h(\mathbf{x}_1) - y_2(\mathbf{x}_2)|, \quad \text{with } L_\ell = \sup_{|q| \le B_{\tilde{f}}} \mathfrak{f}'(q) e^{-\mathfrak{f}(q)}, \quad (7)$$

where $B_{\tilde{f}}$ is an upper bound on the output of any t-NN $h \in \mathfrak{F}$. The Lipschitz continuity is also widely assumed for generalization analysis of DNNs [55, 59]. Assumption 2 is satisfied by commonly used loss functions such as the logistic loss and the exponential loss.

The generalization gap $\mathcal{L}(f) - \hat{\mathcal{L}}(f)$ of any function $f \in \mathfrak{F}$ can be bounded as follows.

**Lemma 3** (Generalization bound for t-NNs). *Under Assumptions 1 and 2, it holds for any $f \in \mathfrak{F}$ that*

$$\mathcal{L}(f) - \hat{\mathcal{L}}(f) \le \frac{L_\ell B_x B_{\underline{\mathbf{W}}}}{\sqrt{N}} (\sqrt{2 \log(2(L+1))} + 1) + 3B\sqrt{\frac{t}{2N}}, \qquad (8)$$

*with probability at least $1 - 2e^{-t}$ for any $t > 0$.*

---

[4]Here for the ease of notation presentation, we use the tensor notation $\underline{\mathbf{W}}$ instead of the set notation $\mathcal{W}$.

**Remark.** *When the input example has channel number $c = 1$, the generalization bound in Theorem 3 is consistent with the F-norm-based bound in Ref. [8].*

## 3.2 Robust Generalization for General t-NNs

We study the adversarial generalization behavior of t-NNs in this section. We first make the following assumption on the adversarial perturbations.

**Assumption 4.** *Given an input example $\underline{\mathbf{x}}$, the adversarial perturbation is chosen within a radius-$\xi$ ball of norm $R_a(\cdot)$ with compatibility constant [35] defined as $C_{R_a} := \sup_{\underline{\mathbf{x}} \neq \mathbf{0}} R_a(\underline{\mathbf{x}})/\|\underline{\mathbf{x}}\|_F$.*

The assumption allows for much broader adversary classes than the commonly considered $l_p$-attacks [54, 55]. For example, if one treats the multi-channel data $\underline{\mathbf{x}} \in \mathbb{R}^{d \times 1 \times c}$ as a matrix of dimensionality $d \times c$ and attacks it with nuclear norm attacks [17], then the constant $C_{R_a} = \sqrt{\min\{d, c\}}$.

Given an example-label pair $(\underline{\mathbf{x}}, y)$, the adversarial loss for any predictor $f$ is defined as $\tilde{\ell}(f(\underline{\mathbf{x}}), y) = \max_{R_a(\underline{\mathbf{x}}' - \underline{\mathbf{x}}) \leq \xi} \ell(f(\underline{\mathbf{x}}'), y)$. The empirical and population adversarial risks are thus defined as $\hat{\mathcal{L}}^{\mathrm{adv}}(f) := N^{-1} \sum_{i=1}^{N} \tilde{\ell}(f(\underline{\mathbf{x}}_i), y_i)$ and $\mathcal{L}^{\mathrm{adv}}(f) := \mathbb{E}_{P_{(\underline{\mathbf{x}}, y)}}[\tilde{\ell}(f(\underline{\mathbf{x}}), y)]$, respectively. The adversarial generalization performance is measured by the adversarial generalization gap (AGP) defined as $\mathcal{L}^{\mathrm{adv}}(f) - \hat{\mathcal{L}}^{\mathrm{adv}}(f)$. Let $B_{\tilde{f}} := (B_x + \xi C_{R_a}) B_{\underline{\mathbf{W}}}$. For any $f \in \mathfrak{F}$, its AGP is bounded as follows.

**Theorem 5** (Adversarial generalization bound for t-NNs). *Under Assumptions 1, 2, and 4, there exists a constant $C$ such that for any $f \in \mathfrak{F}$, it holds with probability at least $1 - 2e^{-t}$ ($\forall t > 0$):*

$$\mathcal{L}^{\mathrm{adv}}(f) - \hat{\mathcal{L}}^{\mathrm{adv}}(f) \leq \frac{CL_\ell B_{\tilde{f}}}{\sqrt{N}} \sqrt{c \sum_{l=1}^{L} d_{l-1} d_l \log(3(L+1))} + 3B\sqrt{\frac{t}{2N}}. \tag{9}$$

**Remark.** *When the input example has channel number $c = 1$ and the attacker uses $l_p$-attack, the adversarial generalization bound in Theorem 5 degenerates to the one in Theorem 4 of Ref. [55].*

# 4 Transformed Low-rank Parameterization for Robust Generalization

## 4.1 Robust Generalization with Exact Transformed Low-rank Parameterization

According to Theorem 5, the AGP bound scales with the square root of the parameter complexity, specifically as $O(\sqrt{c(\sum_l d_{l-1} d_l)/N})$. This implies that achieving the desired adversarial accuracy may require a large number $N$ of training examples. Furthermore, high parameter complexity leads to increased energy consumption, storage requirements, and computational cost when deploying large t-NN models, particularly on resource-constrained embedded and mobile devices.

To this end, we propose a transformed low-rank parameterization scheme to compress the original t-NN models $\mathfrak{F}$. Specifically, given a vector of pre-set ranks $\mathbf{r} = (r_1, \cdots, r_L)^\top \in \mathbb{R}^L$ where $r_l \leq \min\{d_l, d_{l-1}\}$, we consider the following subset of the original t-NNs:

$$\mathfrak{F}_{\mathbf{r}} := \left\{ f \mid f \in \mathfrak{F}, \text{and } r_t(\underline{\mathbf{W}}^{(l)}) \leq r_l, \ l = 1, \cdots, L \right\}. \tag{10}$$

In the function set $\mathfrak{F}_{\mathbf{r}}$, the weight tensor $\underline{\mathbf{W}}^{(l)}$ of the $l$-th layer has the upper bounded tubal rank, which means low-rankness in the transformed domain[5]. We bound the AGP for any $f \in \mathfrak{F}_{\mathbf{r}}$ as follows.

**Theorem 6** (Adversarial generalization bound for t-NNs with transformed low-rank weights). *Under Assumptions 1, 2, and 4, there exists a constant $C'$ such that*

$$\mathcal{L}^{\mathrm{adv}}(f_{\mathbf{r}}) - \hat{\mathcal{L}}^{\mathrm{adv}}(f_{\mathbf{r}}) \leq \frac{C' L_\ell B_{\tilde{f}}}{\sqrt{N}} \sqrt{c \sum_{l=1}^{L} r_l(d_{l-1} + d_l) \log(9(L+1))} + 3B\sqrt{\frac{t}{2N}}, \tag{11}$$

*holds for any $f_{\mathbf{r}} \in \mathfrak{F}_{\mathbf{r}}$ with probability at least $1 - 2e^{-t}$ ($\forall t > 0$).*

---

[5]For empirical implementations, one can adopt similar rank learning strategy to Ref. [15] to select a suitable rank parameter $\mathbf{r}$. Due to the scope of this paper, we leave this for future work.

Comparing Theorem 6 with Theorem 5, we observe that the adversarial generalization bound under transformed low-rank parameterization has a better scaling, specifically $O(\sqrt{c \sum_l r_l(d_{l-1} + d_l)/N})$. This also implies that a smaller number $N$ of training examples is required to achieve the desired accuracy, as well as reduced energy consumption, storage requirements, and computational cost. Please refer to Sec. A.1 in the appendix for numerical evidence.

## 4.2 Implicit Bias of Gradient Flow for Adversarial Training of Over-parameterized t-NNs

Although Theorem 6 shows *exactly* transformed low-rank parameterization leads to lower bounds, the well trained t-NNs on real data rarely have exactly transformed low-rank weights. In this section, we prove that the highly over-parameterized t-NNs, trained by adversarial training with gradient flow (GF), are *approximately* of transformed low-rank parameterization under certain conditions.

First, the proposed t-NN $f(\underline{\mathbf{x}}; \underline{\mathbf{W}})$ is said to be (positively) *homogeneous* as the condition $f(\underline{\mathbf{x}}; a\underline{\mathbf{W}}) = a^{L+1} f(\underline{\mathbf{x}}; \underline{\mathbf{W}})$ holds for any positive constant $a$. Motivated by Ref. [29], we focus on the scale invariant adversarial perturbations defined as follows.

**Definition 4** (Scale invariant adversarial perturbation [29]). *An adversarial perturbation $\underline{\boldsymbol{\delta}}_i(\underline{\mathbf{W}})$ is said to be scale invariant for $f(\underline{\mathbf{x}}; \underline{\mathbf{W}})$ at any given example $\underline{\mathbf{x}}_i$ if it satisfies $\underline{\boldsymbol{\delta}}_i(a\underline{\mathbf{W}}) = \underline{\boldsymbol{\delta}}_i(\underline{\mathbf{W}})$ for any positive constant $a$.*

**Lemma 7.** *The $l_2$-FGM [34], FGSM [9], $l_2$-PGD and $l_\infty$-PGD [31] perturbations for the t-NNs are all scale invariant.*

Then, we consider adversarial training of t-NNs with scale invariant adversarial perturbations by GF, which can be seen as gradient descent with infinitesimal step size. When using GF for the ReLU t-NNs, $\underline{\mathbf{W}}$ changes continuously with time, and the trajectory of parameter $\underline{\mathbf{W}}$ during training is an arc $\underline{\mathbf{W}} : [0, \infty) \to \mathbb{R}^{\dim(\underline{\mathbf{W}})}, t \mapsto \underline{\mathbf{W}}(t)$ that satisfies the differential inclusion [7, 30]

$$\frac{\mathrm{d}\underline{\mathbf{W}}(t)}{\mathrm{d}t} \in -\partial^\circ \hat{\mathcal{L}}^{\mathrm{adv}}(\underline{\mathbf{W}}(t)) \tag{12}$$

for $t \geq 0$ *a.e.*, where $\partial^\circ \hat{\mathcal{L}}^{\mathrm{adv}}$ denotes the Clarke's subdifferential [7] with respect to $\underline{\mathbf{W}}(t)$. If $\hat{\mathcal{L}}^{\mathrm{adv}}(\underline{\mathbf{W}})$ is actually a $C^1$-smooth function, the above differential inclusion reduces to

$$\frac{\mathrm{d}\underline{\mathbf{W}}(t)}{\mathrm{d}t} = -\frac{\partial \hat{\mathcal{L}}^{\mathrm{adv}}(\underline{\mathbf{W}}(t))}{\partial \underline{\mathbf{W}}(t)} \tag{13}$$

for any $t \geq 0$, which corresponds to the GF with differential in the usual sense. However, for simplicity, we follow Refs. [45, 46] and still use Eq. (13) to denote Eq. (12) with a slight abuse of notation, even if $\hat{\mathcal{L}}^{\mathrm{adv}}$ does not satisfy differentiability but only local Lipschitzness [6].

We also make an assumption on the training data as follows.

**Assumption 8** (Existence of a separability of adversarial examples during training). *There exists a time $t_0$ such that $\hat{\mathcal{L}}^{\mathrm{adv}}(t_0) \leq N^{-1}\ell(b_{\mathfrak{f}})$.*

This assumption is a generalization of the separability condition in Refs. [29, 30]. Adversarial training can typically achieve this separability in practice, *i.e.*, the model can fit adversarial examples of the training dataset, making the above assumption reasonable. Then, we obtain the following lemma.

**Lemma 9** (Convergence to the direction of a KKT point). *Consider the hypothesis class $\mathfrak{F}$ in Eq. (6). Under Assumptions 2 and 8, the limit point of normalized weights $\{\underline{\mathbf{W}}(t)/ \|\underline{\mathbf{W}}(t)\|_{\mathrm{F}} : t \geq 0\}$ of the GF for Eq. (13), i.e., the empirical adversarial risk with scale invariant adversarial perturbations $\underline{\boldsymbol{\delta}}_i(\underline{\mathbf{W}})$, is aligned with the direction of a KKT point of the minimization problem:*

$$\min_{\underline{\mathbf{W}}} \frac{1}{2} \|\underline{\mathbf{W}}\|_{\mathrm{F}}^2, \qquad \text{s.t. } y_i f(\underline{\mathbf{x}}_i + \underline{\boldsymbol{\delta}}_i(\underline{\mathbf{W}}); \underline{\mathbf{W}}) \geq 1, \quad i = 1, \cdots, N. \tag{14}$$

Building upon Lemma 9, we can establish that highly over-parameterized t-NNs undergoing adversarial training with GF will exhibit an implicit bias towards transformed low-rank weights.

---

[6] Note that the ReLU function is not differentiable at 0. Practical implementations of gradient methods define the derivative $\sigma'(0)$ to be a constant in $[0, 1]$. In this work we assume for convenience that $\sigma'(0) = 0$.

**Theorem 10** (Implicit low-rankness for t-NNs induced by GF). *Suppose there is an example $\underline{\mathbf{x}}_i$ satisfying $\|\underline{\mathbf{x}}_i\|_F \leq 1$ in the training set $S = \{(\underline{\mathbf{x}}_i, y_i)\}_{i=1}^N$. Suppose there is a $(J+1)$-layer ($J \geq 2$) ReLU t-NN, denoted by $g(\underline{\mathbf{x}}; \underline{\mathbf{V}})$ with parameters $\underline{\mathbf{V}} = (\underline{\mathbf{V}}^{(1)}, \cdots, \underline{\mathbf{V}}^{(J)}, \mathbf{v})$, satisfying the conditions:*

**(C.1)** *the dimensionality of the weight tensor $\underline{\mathbf{V}}^{(j)} \in \mathbb{R}^{m_j \times m_{j-1} \times \mathsf{c}}$ of the $j$-th t-product layer satisfies $m_j \geq 2$, $j = 1, \cdots, J$;*

**(C.2)** *there is a constant $B_v > 0$, such that the Euclidean norm of the weights $\underline{\mathbf{V}} = (\underline{\mathbf{V}}^{(1)}, \cdots, \underline{\mathbf{V}}^{(L)}, \mathbf{v})$ satisfy $\|\underline{\mathbf{V}}^{(j)}\|_F \leq B_v$ for any $j = 1, \cdots, J$ and $\|\mathbf{v}\|_2 \leq B_v$;*

**(C.3)** *for all $i \in \{1, \cdots, N\}$, we have $y_i g(\underline{\mathbf{x}}_i + \underline{\boldsymbol{\delta}}_i(\underline{\mathbf{V}}); \underline{\mathbf{V}}) \geq 1$.*

*Then, we consider the class of over-parameterized t-NNs $\mathfrak{F} = \{f(\underline{\mathbf{x}}; \underline{\mathbf{W}})\}$ defined in Eq. (5) satisfying*

**(C.4)** *the number $L$ of t-product layers is much greater than $J$;*

**(C.5)** *the dimensionality of weight $\underline{\mathbf{W}}^{(l)} \in \mathbb{R}^{d_l \times d_{l-1} \times \mathsf{c}}$ satisfies $d_l \gg \max_{j \leq J}\{m_j\}$ for any $l \leq L$.*

*Let $\underline{\mathbf{W}}^* = (\underline{\mathbf{W}}^{*(1)}, \cdots, \underline{\mathbf{W}}^{*(L)}, \mathbf{w}^*)$ be a global optimum of Problem (14). Namely, $\underline{\mathbf{W}}^*$ parameterizes a minimum-norm t-NN $f(\underline{\mathbf{x}}; \underline{\mathbf{W}}^*) \in \mathfrak{F}$ that labels the perturbed training set correctly with margin 1 under scale invariant adversarial perturbations. Then, we have*

$$\frac{L}{\sum_{l=1}^{L}\left(r_{\text{stb}}(\widetilde{\mathbf{W}}_M^{*(l)})\right)^{-1/2}} \leq \frac{1}{\left(1 + \frac{1}{L}\right)\left(\frac{1}{B_v}\right)^{\frac{J+1}{L+1}}\sqrt{\frac{L+1}{(J+1)+(\mathsf{c}m_J)(L-J)} - \frac{1}{L}}},$$

*where $\widetilde{\mathbf{W}}_M^{*(l)}$ denotes the $M$-block-diagonal matrix of weight tensor $\underline{\mathbf{W}}^{*(l)}$ for any $l = 1, \cdots, L$.*

By the above theorem, when $L$ is sufficiently large, the harmonic mean of the square root of the stable rank of $\widetilde{\mathbf{W}}_M^{*(l)}$, *i.e.*, the $M$-block-diagonal matrix of weight tensor $\underline{\mathbf{W}}^{*(l)}$, is approximately bounded by $\sqrt{\mathsf{c}m_J}$, which is significantly smaller than the square root of the dimensionality $\sqrt{\min\{\mathsf{c}d_l, \mathsf{c}d_{l-1}\}}$ according to condition **(C.5)** in Theorem 10. Thus, $f(\underline{\mathbf{x}}; \underline{\mathbf{W}}^*)$ has a nearly low-rank parameterization in the transformed domain. In our case, the weights $\underline{\mathbf{W}}(t)$ generated by GF tend to have an infinite norm and to converge in direction to a transformed low-rank solution. Moreover, note that the ratio between the spectral norm and the F-norm is invariant to scaling, and hence it suggests that after a sufficiently long time, GF tends to reach a t-NN with transformed low-rank weight tensors. Refer to Sec. A.2 for numerical evidence supporting Theorem 10.

### 4.3 Robust Generalization with Approximate Transformed Low-rank Parameterization

Theorem 10 establishes that for highly over-parameterized adversarial training with GF, well-trained t-NNs exhibit approximately transformed low-rank parameters under specific conditions. In this section, we analyze the AGP of t-NNs that possess an approximately transformed low-rank parameterization[7].

Initially, by employing low-tubal-rank tensor approximation [20], one can always compress an *approximately* low-tubal-rank parameterized t-NN $f$ by a t-NN $g \in \mathfrak{F}_\mathbf{r}$ with an *exact* low-tubal-rank parameterization, ensuring a small distance between $g$ and $f$ in the parameter space. Now, the question is: *Can the small parametric distance between $f$ and $g$ also indicate a small difference in their adversarial generalization behaviors?* To answer this question, we first define the $(\delta, \mathbf{r})$-approximate low-tubal-rank parameterized functions.

**Definition 5** (($\delta, \mathbf{r}$)-approximate low-tubal-rank parameterization). *A t-NN $f(\underline{\mathbf{x}}; \underline{\mathbf{W}}) \in \mathfrak{F}$ with weights $\underline{\mathbf{W}} = (\mathbf{w}, \underline{\mathbf{W}}^{(1)}, \cdots, \underline{\mathbf{W}}^{(L)})$ is said to satisfy the $(\delta, \mathbf{r})$-approximate low-tubal parameterization with tolerance $\delta > 0$ and rank $\mathbf{r} = (r_1, \cdots, r_L)^\top \in \mathbb{N}^L$, if there is a t-NN $g(\underline{\mathbf{x}}; \underline{\mathbf{W}}_\mathbf{r}) \in \mathfrak{F}_\mathbf{r}$ whose weights $\underline{\mathbf{W}}_g = (\mathbf{w}, \underline{\mathbf{W}}_{r_1}^{(1)}, \cdots, \underline{\mathbf{W}}_{r_L}^{(L)})$ satisfy $\|\underline{\mathbf{W}}_{r_l}^{(l)} - \underline{\mathbf{W}}^{(l)}\|_F \leq \delta$ for any $l = 1, \cdots, L$.*

Furthermore, let's consider the collection of t-NNs with approximately low-tubal-rank weights

$$\mathfrak{F}_{\delta, \mathbf{r}} := \{f \in \mathfrak{F} \mid f \text{ satisfies the } (\delta, \mathbf{r})\text{-approximate low-tubal-rank parameterization}\}. \tag{15}$$

---

[7]We use the tubal rank as a measure of low-rankness in the transformed domain for notation simplicity. One can also consider the average rank [52] or multi-rank [50] for more refined bounds with quite similar techniques.

Subsequently, we analyze the AGP for any $f \in \mathfrak{F}_{\delta,\mathbf{r}}$ in terms of its low-tubal-rank compression $g \in \mathfrak{F}_{\mathbf{r}}$. The idea is motivated by the work on compressed bounds for non-compressed but compressible models [43], originally developed for generalization analysis of NNs for standard training.

Under Assumption 2, we first define $\mathfrak{F}_{\delta,\mathbf{r}}^{\mathrm{adv}} := \{\tilde{f} : (\mathbf{x}, y) \mapsto \min_{R_{\mathrm{a}}(\mathbf{x}' - \mathbf{x}) \leq \xi} y f(\mathbf{x}') \mid f \in \mathfrak{F}_{\delta,\mathbf{r}}\}$ as the adversarial version of $\mathfrak{F}_{\delta,\mathbf{r}}$. To analyze the AGP of $f \in \mathfrak{F}_{\delta,\mathbf{r}}$ through $g \in \mathfrak{F}_{\mathbf{r}}$, we instead consider their adversarial counterparts $\tilde{f} \in \mathfrak{F}_{\delta,\mathbf{r}}^{\mathrm{adv}}$ and $\tilde{g} \in \mathfrak{F}_{\mathbf{r}}^{\mathrm{adv}}$, where $\mathfrak{F}_{\mathbf{r}}^{\mathrm{adv}}$ is defined as $\mathfrak{F}_{\mathbf{r}}^{\mathrm{adv}} := \{\tilde{g} : (\mathbf{x}, y) \mapsto \min_{R_{\mathrm{a}}(\mathbf{x} - \mathbf{x}') \leq \xi} y g(\mathbf{x}') \mid g \in \mathfrak{F}_{\mathbf{r}}\}$. Define the Minkowski difference of $\mathfrak{F}_{\delta,\mathbf{r}}^{\mathrm{adv}}$ and $\mathfrak{F}_{\mathbf{r}}^{\mathrm{adv}}$ as $\mathfrak{F}_{\delta,\mathbf{r}}^{\mathrm{adv}} - \mathfrak{F}_{\mathbf{r}}^{\mathrm{adv}} := \{\tilde{f} - \tilde{g} \mid \tilde{f} \in \mathfrak{F}_{\delta,\mathbf{r}}^{\mathrm{adv}}, \tilde{g} \in \mathfrak{F}_{\mathbf{r}}^{\mathrm{adv}}\}$. The empirical $L_2$-norm of a t-NN $h \in \mathfrak{F}$ on the training data $S = \{(\mathbf{x}_i, y_i)\}_{i=1}^N$ is defined as $\|h\|_S := \sqrt{N^{-1} \sum_{i=1}^N h^2(\mathbf{x}_i, y_i)}$, and the population $L_2$-norm is $\|h\|_{L_2} := \sqrt{\mathbb{E}_{P(\mathbf{x},y)}[h^2(\mathbf{x}, y)]}$. Define the local Rademacher complexity of $\mathfrak{F}_{\delta,\mathbf{r}}^{\mathrm{adv}} - \mathfrak{F}_{\mathbf{r}}^{\mathrm{adv}}$ of radius $\mathfrak{r} > 0$ as $\dot{R}_{\mathfrak{r}}(\mathfrak{F}_{\delta,\mathbf{r}}^{\mathrm{adv}} - \mathfrak{F}_{\mathbf{r}}^{\mathrm{adv}}) := \bar{R}_N(\{h \in \mathfrak{F}_{\delta,\mathbf{r}}^{\mathrm{adv}} - \mathfrak{F}_{\mathbf{r}}^{\mathrm{adv}} \mid \|h\|_{L_2} \leq \mathfrak{r}\})$, where $\bar{R}_N(\mathcal{H})$ denotes the average Rademacher complexity of a function class $\mathcal{H}$ [4].

The first part of the upcoming Theorem 12 shows that a small parametric distance between $f$ and $g$ leads to a small empirical $L_2$-distance in the adversarial output space. Specifically, for any $f(\mathbf{x}; \underline{\mathbf{W}}) \in \mathfrak{F}_{\delta,\mathbf{r}}$ with compression $g(\mathbf{x}; \underline{\mathbf{W}}_{\mathbf{r}})$, their (adversarial) empirical $L_2$-distance $\|\tilde{f}(\mathbf{x}; \underline{\mathbf{W}}) - \tilde{g}(\mathbf{x}; \underline{\mathbf{W}}_{\mathbf{r}})\|_S$ can be bounded by a small constant $\hat{\mathfrak{r}} > 0$ in linearity of $\delta$. We also aim for a small population $L_2$-distance by first assuming the local Rademacher complexity $\dot{R}_{\mathfrak{r}}(\mathfrak{F}_{\delta,\mathbf{r}}^{\mathrm{adv}} - \mathfrak{F}_{\mathbf{r}}^{\mathrm{adv}})$ can be bounded by a concave function of $\mathfrak{r}$, following common practice in Rademacher complexity analysis [4, 43].

**Assumption 11.** *For any $\mathfrak{r} > 0$, there exists a function $\phi(\mathfrak{r}) : [0, \infty) \to [0, \infty)$ such that $\dot{R}_{\mathfrak{r}}(\mathfrak{F}_{\delta,\mathbf{r}}^{\mathrm{adv}} - \mathfrak{F}_{\mathbf{r}}^{\mathrm{adv}}) \leq \phi(\mathfrak{r})$ and $\phi(2\mathfrak{r}) \leq 2\phi(\mathfrak{r})$.*

We further define $\mathfrak{r}_* = \mathfrak{r}_*(t) := \inf\{\mathfrak{r} > 0 \mid 16 B_{\tilde{f}} \mathfrak{r}^{-2} \phi(\mathfrak{r}) + B_{\tilde{f}} \mathfrak{r}^{-1} \sqrt{2t/N} + 2t B_{\tilde{f}}^2 \mathfrak{r}^{-2}/N \leq 1/2\}$ for any $t > 0$, such that the population $L_2$-norm of any $h \in \mathfrak{F}_{\delta,\mathbf{r}}^{\mathrm{adv}} - \mathfrak{F}_{\mathbf{r}}^{\mathrm{adv}}$ can be bounded by $\|h\|_{L_2}^2 \leq 2(\|h\|_S^2 + \mathfrak{r}_*^2)$ using the peeling argument [42, Theorem 7.7]. We then establish an adversarial generalization bound for approximately low-tubal-rank t-NNs as follows.

**Theorem 12** (Adversarial generalization bound for general approximately low-tubal-rank t-NNs).
*(I). For any $f \in \mathfrak{F}_{\delta,\mathbf{r}}$ with adversarial proxy $\tilde{f} \in \mathfrak{F}_{\delta,\mathbf{r}}^{\mathrm{adv}}$, there exists a function $g \in \mathfrak{F}_{\mathbf{r}}$ with adversarial proxy $\tilde{g} \in \mathfrak{F}_{\mathbf{r}}^{\mathrm{adv}}$, such that the empirical $L_2$-distance $\|\tilde{f} - \tilde{g}\|_S \leq \delta B_{\tilde{f}} \sum_{l=1}^L B_l^{-1} =: \hat{\mathfrak{r}}$.*
*(II). Let $\dot{\mathfrak{r}} := \sqrt{2(\hat{\mathfrak{r}}^2 + \mathfrak{r}_*^2)}$. Under Assumptions 1, 2, 4, 11, there exist constants $C_1, C_2 > 0$ satisfying*

$$
\mathcal{L}^{\mathrm{adv}}(f) - \hat{\mathcal{L}}^{\mathrm{adv}}(f) \leq \underbrace{\frac{C_1 L_\ell B_{\tilde{f}}}{\sqrt{N}} \sqrt{\mathsf{c} \sum_{l=1}^L r_l(d_{l-1} + d_l) \log(9(L+1))} + B\sqrt{\frac{t}{2N}}}_{\text{main term}}
$$
$$
+ \underbrace{C_2\left(\Phi(\dot{\mathfrak{r}}) + L_\ell \dot{\mathfrak{r}} \sqrt{\frac{t}{N}} + \frac{t L_\ell B_{\tilde{f}}}{N}\right)}_{\text{bias term}},
$$
(16)

*for any $f \in \mathfrak{F}_{\delta,\mathbf{r}}$ with probability at least $1 - 4e^{-t}$ for any $t > 0$, where $\Phi(\mathfrak{r})$ is defined as*

$$
\Phi(\mathfrak{r}) := \bar{R}_N\left(\{\ell \circ \tilde{f} - \ell \circ \tilde{g} \mid \tilde{f} \in \mathfrak{F}_{\delta,\mathbf{r}}^{\mathrm{adv}}, \tilde{g} \in \mathfrak{F}_{\mathbf{r}}^{\mathrm{adv}}, \|\tilde{f} - \tilde{g}\|_{L_2} \leq \mathfrak{r}\}\right).
$$

The main term of the bound quantifies the complexity of functions in $\mathfrak{F}_{\mathbf{r}}$ with exact low-tubal-rank parameterization in adversarial settings, which can be significantly smaller than that of $\mathfrak{F}_{\delta,\mathbf{r}}$. On the other hand, the bias term captures the sample complexity required to bridge the gap between approximately low-tubal-rank parameterized $\mathfrak{F}_{\delta,\mathbf{r}}$ and exactly low-tubal-rank parameterized $\mathfrak{F}_{\mathbf{r}}$. As we usually observe $\mathfrak{r}_*^2 = o(1/\sqrt{N})$, setting $\hat{\mathfrak{r}} = o_p(1)$ allows the bias term to decay faster than the main term, which is $O(1/\sqrt{N})$. Theorem 12 suggests that *a small parametric distance between $f \in \mathfrak{F}_{\delta,\mathbf{r}}$ and $g \in \mathfrak{F}_{\mathbf{r}}$ also implies a small difference in their adversarial generalization behaviors.*

**A special case.** We also showcase a specific scenario where the weights of t-product layers exhibit a polynomial spectral decay in the transformed domain, leading to a considerably small AGP bound.

**Assumption 13.** *Consider the setting where any t-NN $f(\mathbf{x}; \underline{\mathbf{W}}) \in \mathfrak{F}_{\delta, \mathbf{r}}$ has tensor weights $\underline{\mathbf{W}}^{(l)}$ ($l = 1, \cdots, L$) whose singular values in the transformed domain satisfy $\sigma_j(M(\underline{\mathbf{W}}^{(l)})_{:,:,k}) \leq V_0 \cdot j^{-\alpha}$, where $V_0 > 0$ is a constant, and $\sigma_j(\cdot)$ is the j-th largest singular value of a matrix.*

Under Assumption 13, the weight tensor $\underline{\mathbf{W}}^{(l)}$ can be approximated by its optimal tubal-rank-$r_l$ approximation $\underline{\mathbf{W}}_{r_l}^{(l)}$ for any $1 \leq r_l \leq \min\{d_l, d_{l-1}\}$ with error $\|\underline{\mathbf{W}}^{(l)} - \underline{\mathbf{W}}_{r_l}^{(l)}\|_F \leq \sqrt{\mathsf{c}/(2\alpha-1)}V_0(r_l-1)^{(1-2\alpha)/2}$ [20], which can be much smaller than $\|\underline{\mathbf{W}}_{r_l}^{(l)}\|_F$ when $\alpha > 1/2$ is sufficiently large. Thus, we can find an exactly low-tubal-rank parameterized $g \in \mathfrak{F}_{\mathbf{r}}$ for any $f \in \mathfrak{F}_{\delta, \mathbf{r}}$ satisfying Assumption 13, such that the parametric distance between $g$ and $f$ is quite small. The following theorem shows that the small parametric distance also leads to a small AGP.

**Theorem 14.** *Under Assumptions 1, 2, 4, and 13, if we let $\hat{\mathfrak{r}} = V_0 B_{\tilde{f}} \sum_{l=1}^{L}(r_l+1)^{-\alpha} B_l^{-1}$, then for any t-NN $f \in \mathfrak{F}_{\delta, \mathbf{r}}$, there exists a function $g \in \mathfrak{F}_{\mathbf{r}}$ whose t-product layer weights have tubal-rank exactly no greater than $r_l$, satisfying $\|\tilde{f} - \tilde{g}\|_S \leq \hat{\mathfrak{r}}$. Further, there is a constant $C_\alpha$ only depending on $\alpha$ such that the AGP, i.e., $\mathcal{L}^{\mathrm{adv}}(f) - \hat{\mathcal{L}}^{\mathrm{adv}}(f)$, of any $f \in \mathfrak{F}_{\delta, \mathbf{r}}$ can be upper bounded by*

$$C_\alpha L_\ell \left\{ B_{\tilde{f}} E_1 + \hat{\mathfrak{r}} \sqrt{E_1} + E_2^{\frac{2\alpha}{2\alpha+1}} \left( B_{\tilde{f}}^{\frac{2\alpha-1}{2\alpha+1}} + 1 \right) + \hat{\mathfrak{r}}^{\frac{2\alpha}{2\alpha+1}} \sqrt{E_2} + (\hat{\mathfrak{r}} + \frac{B}{L_\ell})\sqrt{\frac{t}{N}} + \frac{1+tB_{\tilde{f}}}{N} \right\},$$

*for any $t > 0$ with probability at least $1 - 4e^{-t}$, where $E_1 = N^{-1}\mathsf{c}\sum_{l=1}^{L} r_l(d_l + d_{l-1})\log(9NLB_{\tilde{f}}/\sqrt{\mathsf{c}})$ and $E_2 = N^{-1}\mathsf{c}\sum_{l=1}^{L} \left( LV_0 B_{\tilde{f}} B_l^{-1} \right)^{1/\alpha}(d_l + d_{l-1})\log(9NLB_{\tilde{f}}/\sqrt{\mathsf{c}})$.*

This suggests that by choosing a sufficiently large $\alpha > 1/2$, where each weight tensor has a tubal-rank close to 1, we can attain a superior generalization error bound. It is important to note that the rank $r_l$ can be arbitrarily chosen, and there exists a trade-off relationship between $\hat{\mathfrak{r}}$ and $E_1$. Therefore, by selecting the rank appropriately for a balanced trade-off, we can obtain an optimal bound as follows.

**Corollary 15.** *Under the same assumption to Theorem 14, if we choose the parameter $\mathbf{r}$ of tubal ranks in $\mathfrak{F}_{\mathbf{r}}$ by $r_l = \min\{\lceil \left(LV_0 B_{\tilde{f}} B_l^{-1}\right)^{1/\alpha} \rceil, d_l, d_{l-1}\}$, then there is a constant $C_\alpha$ only depending on $\alpha$ such that the AGP of any $f \in \mathfrak{F}_{\delta, \mathbf{r}}$ can be upper bounded as*

$$\mathcal{L}^{\mathrm{adv}}(f) - \hat{\mathcal{L}}^{\mathrm{adv}}(f) \leq C_\alpha L_\ell \left\{ B_{\tilde{f}}^{1-1/(2\alpha)} \sqrt{\frac{\mathsf{c}\sum_{l=1}^{L} \left(LV_0 B_l^{-1}\right)^{1/\alpha}(d_l + d_{l-1})\log(9NLB_{\tilde{f}}/\sqrt{\mathsf{c}})}{N}} \right.$$
$$\left. + E_2^{\frac{2\alpha}{2\alpha+1}} \left( B_{\tilde{f}}^{\frac{2\alpha-1}{2\alpha+1}} + 1 \right) + \sqrt{E_2} + \frac{B}{L_\ell}\sqrt{\frac{t}{N}} + \frac{1+tB_{\tilde{f}}}{N} \right\},$$

*with probability at least $1 - 4e^{-t}$ for any $t > 0$.*

It is worth highlighting that the bound exhibits a linear dependency on the number of neurons in the t-product layers, represented as $O(\sqrt{\mathsf{c}\sum_l(d_l + d_{l-1})/N})$. In contrast, Theorem 5 demonstrates a dependency on the total number of parameters, denoted as $O(\sqrt{\mathsf{c}\sum_l d_l d_{l-1}/N})$. This observation suggests that employing the low-tubal-rank parameterization can potentially enhance adversarial generalization for t-NNs.

# 5 Related Works

**T-SVD-based data and function representation.** The unique feature of t-SVD-based data representation, in contrast to classical low-rank decomposition methods, is the presence of low-rankness in the transformed domain. This transformed low-rankness is crucial for effectively modeling real multi-channel data with both smoothness and low-rankness [24, 49, 50]. Utilized in t-product layers in DNNs [32, 36, 53], t-SVD has also been a workhorse for function representation and achieves impressive empirical performance. While t-SVD-based signal processing models have been extensively studied theoretically [13, 24, 27, 40, 50, 60], the t-SVD-based learning model itself has not been thoroughly scrutinized until this paper. Hence, this study represents the first theoretical analysis of t-SVD-based learning models, contributing to the understanding of their theoretical foundations.

**Theoretical analysis methods.** Our analysis draws on norm-based generalization analysis [37] and implicit regularization of gradient descent-based learning [46] as related theoretical analysis methods. Norm-based generalization analysis plays a crucial role in theoretical analysis across various domains, including standard generalization analysis of DNNs [8], compressed models [22], non-compressed models [43], and adversarial generalization analysis [3, 55, 59]. Our work extends norm-based tools to analyze both standard and adversarial generalization in t-NNs, going beyond the traditional use of matrix products. For implicit regularization of gradient descent based learning, extensive past research has been conducted on implicit bias of GF for both standard and adversarial training of homogeneous networks building on matrix product layers, respectively [16, 30, 45]. We non-trivially extend these methods to analyze t-NNs and reveals that GF for over-parameterized ReLU t-NNs produces nearly transformed low-rank weights under scale invariant adversarial perturbations.

Our theoretical results notably deviate from the standard error bounds for fully connected neural networks (FNNs) in several ways:

- The generalization bounds in Lemma 3 and Theorem 5 for t-NNs diverge from their counterparts for FNNs in Refs. [8, 55, 59] due to the channel number c in t-NNs. Moreover, Theorem 5 encompasses a wider range of adversary classes than the $l_p$-attacks in the aforementioned references.

- The uniqueness of Theorem 6, compared to Refs. [55, 59], stems from its consideration of weight low-rankness in the adversarial generalization bound, suggesting possible robustness improvements in generalization.

- Our exploration of the implicit bias in GF for adversarial training presents a novel angle: the bias towards approximate transformed low-rankness in t-NNs. While Ref. [29] focuses on the implicit bias in adversarial training for FNNs, centered on KKT point convergence with exponential loss, our work delves deeper, considering a wider array of loss functions in adversarial training for t-NNs.

- A crucial distinction in our adversarial generalization bounds, detailed in Section 4.3, from non-adversarial bounds for FNNs [43] is the integration of the localized Rademacher complexity. This encompasses the Minkowski difference between adversarial counterparts of both approximately and exactly low-tubal-rank t-NNs as seen in Theorem 12.

## 6 Concluding Remarks

A thorough investigation of the generalization behavior of t-NNs is conducted for the first time. We derive upper bounds for the generalization gaps of standard and adversarially trained t-NNs and propose compressing t-NNs with a transformed low-rank structure for more efficient adversarial learning and tighter bounds on the adversarial generalization gap. Our analysis shows that adversarial training with GF in highly over-parameterized settings results in t-NNs with approximately transformed low-rank weights. We further establish sharp adversarial generalization bounds for t-NNs with approximately transformed low-rank weights. Our findings demonstrate that utilizing the transformed low-rank parameterization can significantly enhance the robust generalization of t-NNs, carrying both theoretical and empirical significance.

**Limitations.** While this paper adheres to the norm-based framework for capacity control [8, 37], it is worth noting that the obtained generalization bounds may be somewhat conservative. However, this limitation can be mitigated by employing more sophisticated analysis techniques, as evidenced by recent studies [2, 25, 56, 57].

**Discussions.** The inclination of adversarial training towards low-rank/sparse weights, and the reciprocal effects of parameter reduction on robustness, are currently at the forefront of ongoing research. This domain has witnessed a spectrum of observations and results [6, 23, 41, 51]. In this study, we propose that employing low-rank parameterization can enhance the adversarial robustness of t-NNs, as evidenced by our analysis of uniform adversarial generalization error bounds. However, despite these promising results, it is crucial to emphasize the necessity of a more exhaustive exploration of low-rank parameterization. Its implications, particularly when considered in the context of approximation, estimation, and optimization, are profound and warrant further dedicated research efforts. Such a comprehensive investigation will undoubtedly enhance our understanding and fully unlock the potential of low-rank parameterization in neural networks.

## Acknowledgments

We extend our deepest gratitude to Linfeng Sui and Xuyang Zhao for their indispensable support in implementing the Python code for t-NNs during the rebuttal phase. Our sincere appreciation also goes to both the area chair and reviewers for their unwavering dedication and meticulous attention given to this paper. This research was supported by RIKEN Incentive Research Project 100847-202301062011, by JSPS KAKENHI Grant Numbers JP20H04249 and JP23H03419, and in part by National Natural Science Foundation of China under Grants 62103110 and 62073087.

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

Appendix
# Transformed Low-Rank Parameterization Can Help Robust Generalization for Tensor Neural Networks

In the appendix, we begin by presenting numerical evaluations of our theoretical findings. Subsequently, we introduce the additional notations and preliminaries related to t-SVD, followed by the proofs of the propositions mentioned in the main text.

Our analysis and proofs pertaining to t-NNs differ from those for FNNs as follows:

- Firstly, unlike the analysis for FNNs' generalization bounds based on Rademacher complexity in Refs. [8, 55, 59], we derive specific lemmas for standard and adversarial generalization bounds in t-NNs. This is due to the unique structure of (low-rank) t-product layers. We reformulate the t-product through an operator-like expression in Lemma 16, paving the way for pivotal Lemma 17, supporting the t-product-based "peeling argument." Additionally, we introduce Lemmas 37, 38, and Lemma 33 to handle t-product layer output norms and covering low-tubal-rank tensors.

- Secondly, proving the implicit bias of GF for adversarial training of t-NNs, specifically the approximately transformed low-rankness, is nontrivial in comparison to the proof in Ref. [29] for the implicit bias of adversarial training for FNNs. As we consider more general loss functions for t-NNs in contrast to the exponential loss for FNNs in Ref. [29], we first derive a more general convergence result to the direction of a KKT point for t-NNs Lemma 9, and then goes deeper by using a constructive approach to establish the approximately transformed low-rankness in Theorem 10.

- Thirdly, differing from Ref. [43] which focuses on standard FNN generalization, our approach delves into t-NNs' adversarial generalization. We achieve this by introducing the $(\delta, \mathbf{r})$-parameterization, bounding localized Rademacher complexity for a Minkowski set in adversarial settings, and using low-tubal-rank approximations for tensor weights.

## Contents

# A   Numerical Evaluations of the Theoretical Results

This section presents numerical evaluations for our theoretical results. All training process is conducted on nVidia A100 GPU. For additional information and access to the demo code, please visit the following URL: https://github.com/pingzaiwang/Analysis4TNN/.

## A.1   Effects of Exact Transformed Low-rank Weights on the Adversarial Generalization Gap

To validate the adversarial generalization bound in Theorem 6, we have conducted experiments on the MNIST dataset to explore the relationship between adversarial generalization gaps (AGP), weight tensor low-rankness, and training sample size. We consider binary classification of 3 and 7, with FSGM [9] attacks of strength $20/255$. The t-NN consists of three t-product layers and one FC layer, with weight tensor dimensions of $28 \times 28 \times 28$ for $\underline{\mathbf{W}}^{(1)}$, $\underline{\mathbf{W}}^{(2)}$, and $\underline{\mathbf{W}}^{(3)}$, and $784$ for the FC weight $\mathbf{w}$. As an input to the t-NN, each MNIST image of size $28 \times 28$ is treated as a t-vector of size $28 \times 1 \times 28$.

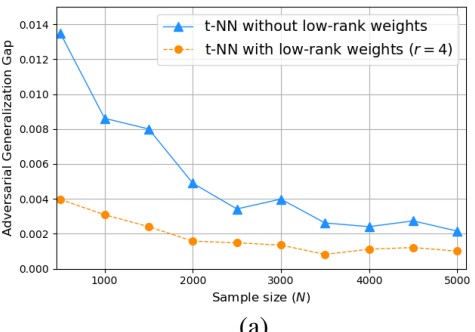 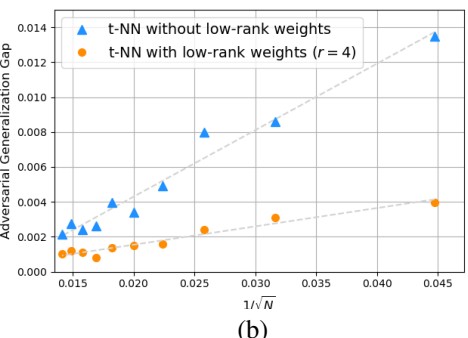

(a)                        (b)

Figure 1: The adversarial generalization gaps plotted against the training sample size ($N$) for t-NNs, both with and without transformed low-rank weight tensors, using the MNIST dataset. In (a), the adversarial generalization gaps are presented against the sample size $N$, while in (b), they are plotted against $1/\sqrt{N}$.

Theorem 6 emphasizes: (*i*) lower weight tensor rank leads to smaller bounds on the adversarial generalization gaps, and (*ii*) the bound diminishes at a rate of $O(1/\sqrt{N})$ as $N$ increases. We explored this by conducting experiments, controlling the upper bounds of the tubal-rank to 4 and 28 for low and full tubal-rank cases, and systematically increasing the number of training samples.

Fig. 1 presents the results. The curves indicate that t-NNs with lower rank weight tensors have smaller robust generalization errors. Interestingly, the adversarial generalization errors seem to follow a linear relationship with $1/\sqrt{N}$, approximately validating the generalization error bound in Theorem 6 by approximating the scaling behavior of the empirical errors.

## A.2   Implicit Bias of GF-based Adversarial Training to Approximately Transformed Low-rank Weight Tensors

We carried out experiments to confirm two theoretical statements related to the analysis of GF-based adversarial training.

**Statement A.2.1** Theorem 10 reveals that, under specific conditions, well-trained t-NNs with highly over-parameterized adversarial training using GF show nearly transformed low-rank parameters.

**Statement A.2.2** Lemma 22 asserts that the empirical adversarial risk approaches zero, and the F-norm of the weights grows infinitely as $t$ approaches infinity.

In continuation of the experimental settings in Sec. A.1, we focus on binary classification on MNIST under FGSM attacks. The t-NN is structured with three t-product layers and one FC layer, with weight dimensions set to $D \times 28 \times 28$ for $\underline{\mathbf{W}}^{(1)}$, $D \times D \times 28$ for $\underline{\mathbf{W}}^{(2)}$ and $\underline{\mathbf{W}}^{(3)}$, and $28D$ for the FC weight $\mathbf{w}$. Our experiments involve setting values of $D$ to 128 and 256, respectively, and we track

the effective rank of each weight tensor, the empirical adversarial risk, and the F-norm of the weights as the number of epochs progresses. Since implementing gradient flow with infinitely small step size is impractical in real experiments, we opt for SGD with a constant learning rate and batch-size of $80$, following the setting on fully connected layers in Ref. [29].

For Statement A.2.1, we present preliminary results illustrating the progression of the stable ranks of the $M$-block-diagonal matrix of tensor weights in Fig. 2 for the settings $D \in \{128, 256\}$. Notably, these results show that the effective ranks decrease as more epochs are executed, thereby confirming the influence of implicit bias on transformed low-rankness, as described in Statement A.2.1.

For Statement A.2.2, we present initial numerical findings depicting the progress of the empirical adversarial risk and the F-norm of the weights in Figs. 3 and 4, respectively. These results exhibit a consistent pattern with the theoretical descriptions outlined in Statement A.2.2 and the numerical results reported in Ref. [29] for adversarial training and Ref. [30] for standard training of FNNs. Specifically, we observe a decreasing trend in the empirical risk function and an increasing trend in the weight tensor's F-norm, which align with the expected behavior based on our theoretical framework and corroborate the numerical results presented in Refs. [29, 30].

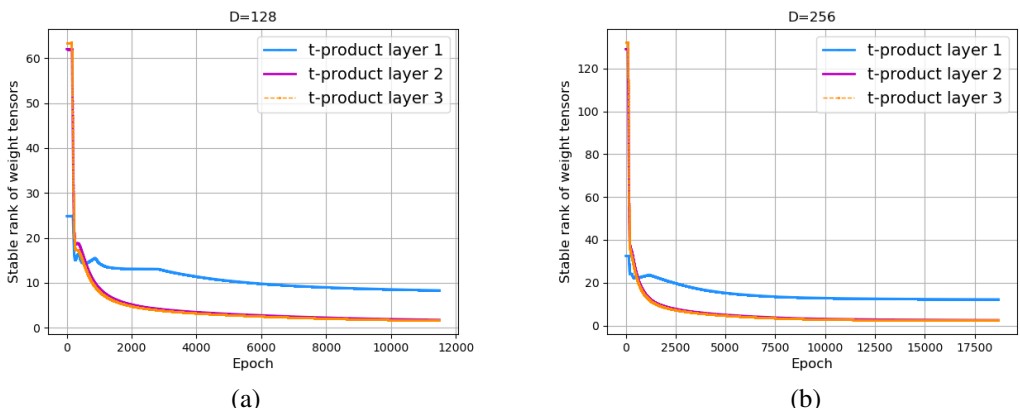

Figure 2: Curves of the stable ranks of the $M$-block-diagonal matrix derived from the weight tensors, plotted against epoch numbers, using the MNIST dataset. Two t-NN size settings are showcased: (a) $D = 128$ and (b) $D = 256$.

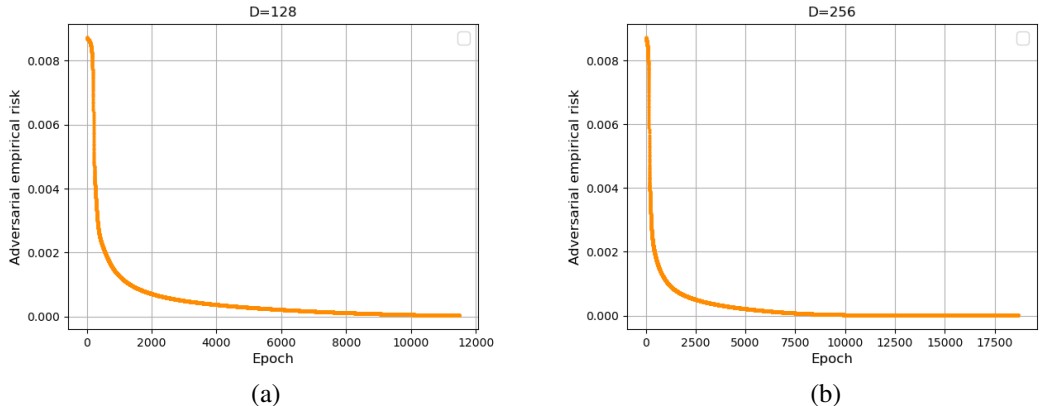

Figure 3: Curves of the adversarial training loss of t-NNs plotted against epoch numbers, using the MNIST dataset. Two t-NN size settings are showcased: (a) $D = 128$ and (b) $D = 256$.

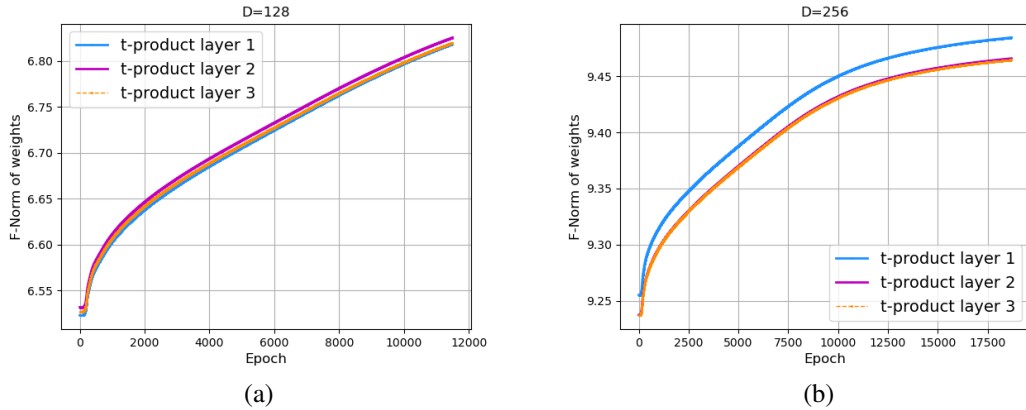

(a)                                                                (b)

Figure 4: Curves of the F-norms of the $M$-block-diagonal matrix derived from the weight tensors, plotted against epoch numbers, using the MNIST dataset. Two t-NN size settings are showcased: (a) $D = 128$ and (b) $D = 256$.

## A.3 Additional Regularization for a Better Low-rank Parameterized t-NN

It is natural to ask: *instead of using adversarial training with GF in highly over-parameterized settings to train a approximately transformed low-rank t-NN, is it possible to apply some extra regularizations in training to achieve a better low-rank parameterization*?

Yes, it is possible to apply additional regularizations during training to achieve a better low-rank representation in t-NNs. Instead of relying solely on adversarial training with gradient flow in highly over-parameterized settings, these extra regularizations can potentially promote and enforce low-rankness in the network.

To validate the concern regarding the addition of an extra regularization term, we performed a preliminary experiment. In this experiment, we incorporated the tubal nuclear norm [48] as an explicit regularizer to induce low-rankness in the transformed domain. Specifically, we add the tubal nuclear norm regularization to the t-NN with three t-product layer $D = 128$ in Sec. A.2 with a regularization parameter $0.01$, and keep the other settings the same as Sec. A.2. We explore how the stable ranks of tensor weights evolve with the epoch number with/without tubal nuclear norm regularization.

The experimental results are depicted in Fig. 5. According to Fig. 5, it becomes evident that the introduction of the explicit low-rank regularizer significantly enforced low-rankness in the transform domain of the weight tensors.

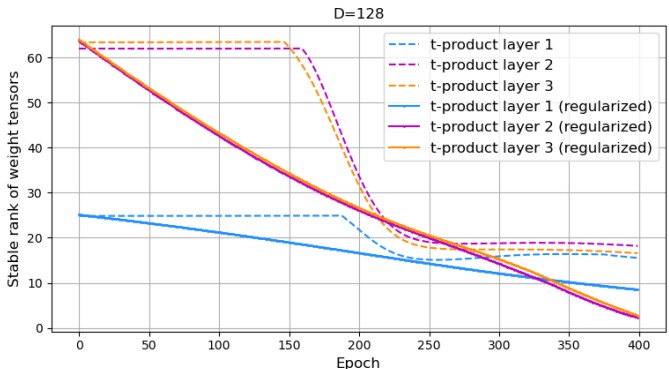

Figure 5: Curves of the stable ranks of the $M$-block-diagonal matrix of the weight tensors versus the epoch number in the scenario where $D = 128$, both with and without the tubal nuclear norm regularization, using the MNIST dataset.

# B  Notations and Preliminaries of t-SVD

## B.1  Notations

For simplicity, we use $c, c_0, C$ etc. to denote constants whose values can vary from line to line. We use We first give the most commonly used notations in Table 1.

Table 1: List of main notations

---

*Notations for t-SVD*

| | | | |
|---|---|---|---|
| $\underline{\mathbf{x}} \in \mathbb{R}^{d \times 1 \times \mathsf{c}}$ | a t-vector | $\underline{\mathbf{T}} \in \mathbb{R}^{m \times n \times \mathsf{c}}$ | a t-matrix |
| $\mathbf{M} \in \mathbb{R}^{\mathsf{c} \times \mathsf{c}}$ | an orthogonal matrix | $M(\cdot)$ | transform via $\mathbf{M}$ in Eq. (1) |
| $*_M$ | t-product | $\widetilde{\underline{\mathbf{T}}}_M$ | $M$-block diagonal matrix of $\underline{\mathbf{T}}$ |
| $\underline{\mathbf{T}}_{:,:,i}$ | $i$-th frontal slice of $\underline{\mathbf{T}}$ | $r_{\mathrm{t}}(\cdot)$ | tensor tubal rank |
| $\lVert \cdot \rVert_{\mathrm{sp}}$ | tensor spectral norm | $\lVert \cdot \rVert_{\mathrm{F}}$ | tensor F-norm |
| $\lVert \cdot \rVert$ | matrix spectral norm | | |

---

*Notations for data representation*

| | | | |
|---|---|---|---|
| $\mathsf{c}$ | number of channels | $d$ | number of features per channel |
| $\underline{\mathbf{x}}_i \in \mathbb{R}^{d \times 1 \times \mathsf{c}}$ | a multi-channel example | $y_i \in \{\pm\}$ | label of multi-channel data $\underline{\mathbf{x}}_i$ |
| $S$ | training sample of size $N$ | $\underline{\boldsymbol{\delta}}_i$ | scale invariant adv. perturbation |
| $R_{\mathrm{a}}(\cdot)$ | norm used for attack | $\underline{\mathbf{x}}_i'$ | adv. perturbed version of $\underline{\mathbf{x}}_i$ |
| $B_x$ | upper bound on $\lVert \underline{\mathbf{x}} \rVert_{\mathrm{F}}$ | $\xi$ | radius of $R_{\mathrm{a}}(\cdot)$ for adv. attack |

---

*Notations for network structure*

| | |
|---|---|
| $L$ | number of t-product layers of a general t-NN |
| $\underline{\mathbf{W}}^{(l)}$ | weight tensor of $l$-th t-product layer with dimensionality $d_l \times d_{l-1} \times \mathsf{c}$ |
| $\mathbf{w}$ | weight vector of fully connected layer with dimensionality $\mathsf{c}d_L$ |
| $f(\underline{\mathbf{x}}; \underline{\mathbf{W}})$ | a general t-NN with weights $\underline{\mathbf{W}} = (\underline{\mathbf{W}}^{(1)}, \cdots, \underline{\mathbf{W}}^{(L)}, \mathbf{w})$ |
| $B_{\underline{\mathbf{W}}}$ | bound on product of Euclidean norms of weights of $f \in \mathfrak{F}$, i.e., $B_{\underline{\mathbf{W}}} = B_w \prod_{l=1}^{L} B_l$ |

---

*Notations for model analysis*

| | |
|---|---|
| $\tilde{f}(\underline{\mathbf{x}}, y)$ | adversarial version of $f(\underline{\mathbf{x}})$ which maps $(\underline{\mathbf{x}}, y)$ to $\inf_{R_{\mathrm{a}}(\underline{\mathbf{x}}' - \underline{\mathbf{x}}) \leq \xi} y_i f(\underline{\mathbf{x}}')$ |
| $B_{\tilde{f}}$ | bound on the output of $\tilde{f}(\underline{\mathbf{x}}, y)$ given as $B_{\tilde{f}} := (B_x + \xi \mathsf{C}_{R_{\mathrm{a}}}) B_{\underline{\mathbf{W}}}$ |
| $\ell(f(\underline{\mathbf{x}}), y)$ | loss function with range $[0, B]$, and Lipstchitz constant $L_\ell$ (See Assumption 2) |
| $\hat{\mathcal{L}}, \mathcal{L}$ | standard empirical and population risk, respectively |
| $\hat{\mathcal{L}}^{\mathrm{adv}}, \mathcal{L}^{\mathrm{adv}}$ | empirical and population risk, respectively |
| $\mathfrak{F}, \mathfrak{F}^{\mathrm{adv}}$ | function class of t-NNs and its adversarial version, respectively |
| $\mathfrak{F}_{\mathbf{r}}, \mathfrak{F}_{\mathbf{r}}^{\mathrm{adv}}$ | function class of low-tubal-rank parameterized t-NNs and adversarial version, resp. |

---

*Notations for implicit bias analysis (Sec. 4.2)*

| | |
|---|---|
| $\tilde{q}_i, \tilde{q}_m, \tilde{\gamma}$ | example robust, sample robust, and smoothly normalized robust margin, resp. |
| $\mathfrak{f}, \mathfrak{g}, b_{\mathfrak{f}}, b_{\mathfrak{g}}, K$ | auxiliary functions and constants to chareterize $\ell(\cdot, \cdot)$ (See Assumption 2) |

---

*Notations for the analysis of apprximately transformed low-rank parameterized models (Sec. 4.3)*

| | |
|---|---|
| $\bar{R}_N, \hat{R}_S, \dot{R}_{\mathbf{r}}$ | average, empirical, and localized Rademacher complexity, resp. |
| $\mathfrak{F}_{\delta, \mathbf{r}}, \mathfrak{F}_{\delta, \mathbf{r}}^{\mathrm{adv}}$ | function class of nearly low-tubal-rank parameterized t-NNs and adv. version, resp. |
| $\lVert f \rVert_S, \lVert f \rVert_{L_2}$ | empirical $L_2$-norm on sample $S$ and population $L_2$-norm of a function $f$, resp. |
| $\boldsymbol{\varepsilon} = \{\varepsilon_i\}_{i=1}^N$ | *i.i.d.* Rademacher variables, i.e., $\varepsilon_i$ equals to 1 or $-1$ with equal probability |

---

## B.2 Additional Preliminaries of t-SVD

We give additional notions and propositions about t-SVD omitted in the main body of the paper.

**Definition 6** ([18]). *The t-transpose of $\underline{\mathbf{T}} \in \mathbb{R}^{m \times n \times c}$ under the $M$ transform in Eq. (1), denoted by $\underline{\mathbf{T}}^\top$, satisfies*

$$M(\underline{\mathbf{T}}^\top)_{:,:,k} = (M(\underline{\mathbf{T}})_{:,:,k})^\top, \ k = 1, \cdots, c.$$

**Definition 7** ([18]). *The t-identity tensor $\underline{\mathbf{I}} \in \mathbb{R}^{m \times m \times c}$ under $M$ transform in Eq. (1) is the tensor such that each frontal slice of $M(\underline{\mathbf{I}})$ is a $c \times c$ identity matrix, i.e,*

$$M(\underline{\mathbf{I}})_{:,:,k} = \mathbf{I}, \ k = 1, \cdots, c.$$

Given the appropriate dimensions, it is trivial to verify that $\underline{\mathbf{T}} *_M \underline{\mathbf{I}} = \underline{\mathbf{T}}$ and $\underline{\mathbf{I}} *_M \underline{\mathbf{T}} = \underline{\mathbf{T}}$.

**Definition 8** ([18]). *A tensor $\underline{\mathbf{Q}} \in \mathbb{R}^{d \times d \times d_3}$ is t-orthogonal under $M$ transform in Eq. (1) if it satisfies*

$$\underline{\mathbf{Q}}^\top *_M \underline{\mathbf{Q}} = \underline{\mathbf{Q}} *_M \underline{\mathbf{Q}}^\top = \underline{\mathbf{I}}.$$

**Definition 9** ([19]). *A tensor is called f-diagonal if all its frontal slices are diagonal matrices.*

**Definition 10** (Tensor t-spectral norm [28]). *The tensor t-spectral norm of any tensor $\underline{\mathbf{T}}$ under $M$ transform in Eq. (1) is defined as the matrix spectral norm of its $M$-block-diagonal matrix $\widetilde{\mathbf{T}}_M$, i.e.,*

$$\|\underline{\mathbf{T}}\|_{\mathrm{sp}} := \left\| \widetilde{\mathbf{T}}_M \right\|.$$

**Lemma 16.** *For any t-matrix $\underline{\mathbf{W}} \in \mathbb{R}^{m \times n \times c}$ and t-vector $\underline{\mathbf{x}} \in \mathbb{R}^{n \times 1 \times c}$, the t-product $\underline{\mathbf{W}} *_M \underline{\mathbf{x}}$ defined under $M$ transform in Eq. (1) is equivalent to a linear operator $\mathrm{op}(\underline{\mathbf{W}})$ on $\mathrm{unfold}(\underline{\mathbf{x}})$ in the orginal domain defined as follows*

$$\mathrm{op}(\underline{\mathbf{W}})(\underline{\mathbf{x}}) = (\mathbf{M}^{-1} \otimes \mathbf{I}_m) \left[ \mathrm{bdiag}\big( (\mathbf{M} \otimes \mathbf{I}_m)\mathrm{unfold}(\underline{\mathbf{W}}) \big) (\mathbf{M} \otimes \mathbf{I}_n) \right] \mathrm{unfold}(\underline{\mathbf{x}}), \quad (17)$$

*where $\otimes$ denotes the Kronecker product, and the operations of $\mathrm{unfold}(\underline{\mathbf{W}})$ and $\mathrm{unfold}(\underline{\mathbf{x}})$ are given explicitly as follows*

$$\mathrm{unfold}(\underline{\mathbf{W}}) = \begin{bmatrix} \underline{\mathbf{W}}_{:,:,1} \\ \underline{\mathbf{W}}_{:,:,2} \\ \vdots \\ \underline{\mathbf{W}}_{:,:,c} \end{bmatrix} \in \mathbb{R}^{mc \times n}, \quad \mathrm{unfold}(\underline{\mathbf{x}}) = \begin{bmatrix} \underline{\mathbf{x}}_{:,1,1} \\ \underline{\mathbf{x}}_{:,1,2} \\ \vdots \\ \underline{\mathbf{x}}_{:,1,c} \end{bmatrix} \in \mathbb{R}^{nc}.$$

Since Eq. (17) is a straightforward reformulation of the definition of t-product in [36, Definition 6.3], the proof is simply omitted.

According to Lemma 16, we have the following remark on the relationship between t-NNs and fully connected neural networks (FNNs).

**Remark** (Connnection with FNNs). *The t-NNs and FNNs can be treated as special cases of each other.*

*(I) When the channel number $c = 1$, the t-product becomes to standard matrix multi-lication and the proposed t-NN predictor Eq. (5) degenerates to an $(L+1)$-layer FNN, which means the FNN is a special case of the t-NN.*

*(II) On the other hand, by the definition of t-product, the t-NN $f(\cdot; \underline{\mathbf{W}})$ in Eq. (5) has the compounding representation as an FNN:*

$$f_{\underline{\mathbf{W}}} = \mathbf{w} \circ \sigma \circ \mathrm{op}(\underline{\mathbf{W}}^{(L)}) \circ \sigma \circ \mathrm{op}(\underline{\mathbf{W}}^{(L-1)}) \circ \cdots \circ \sigma \circ \mathrm{op}(\underline{\mathbf{W}}^{(1)}).$$

*Thus, t-NN can also be seen as a special case of FNN.*

## C   Standarad and Adversarial Generalization Bounds for t-NNs

### C.1   Standarad Generalization Bound for t-NNs

**Lemma 17.** *Consider the ReLU activation. For any t-vector-valued function set $\mathcal{H}$ and any convex and monotonically increasing function $g : \mathbb{R} \to [0, \infty)$,*

$$\mathbb{E}_{\boldsymbol{\varepsilon}} \left[ \sup_{\mathbf{h} \in \mathcal{H}, \underline{\mathbf{W}} : \|\underline{\mathbf{W}}\|_{\mathrm{F}} \leq R} g \left( \left\| \sum_{i=1}^{N} \varepsilon_i \sigma(\underline{\mathbf{W}} *_M \mathbf{h}(\underline{\mathbf{x}}_i)) \right\|_{\mathrm{F}} \right) \right] \leq 2\mathbb{E}_{\boldsymbol{\varepsilon}} \left[ \sup_{\mathbf{h} \in \mathcal{H}} g \left( R \left\| \sum_{i=1}^{N} \varepsilon_i \mathbf{h}(\underline{\mathbf{x}}_i) \right\|_{\mathrm{F}} \right) \right],$$

*where $R > 0$ is a constant.*

*Proof.* This lemma is a direct corollary of Lemma 1 in Ref. [8] by using Eq. (17). $\qquad \square$

*Proof of Lemma 3.* According to Lemma 29, we can upper bound the generalization error of $\ell \circ f$ for any $f \in \mathfrak{F}$ through the (empirical) Rademacher complexity $\hat{R}_S(\ell \circ \mathfrak{F})$ where $\ell \circ \mathfrak{F} := \{\ell \circ f \mid f \in \mathfrak{F}\}$. Further regarding the $L_\ell$-Lipschitzness[8] of the loss function $\ell$, we have $\hat{R}_S(\ell \circ \mathfrak{F}) \leq L_\ell \hat{R}_S(\mathfrak{F})$ by the Talagrand's contraction lemma (Lemma 30). Then, it remains to bound $\hat{R}_S(\mathfrak{F})$.

To upper bound $\hat{R}_S(\mathfrak{F})$, we follow the proof of [8, Theorem 1]. By Jensen's inequality, the (scaled) Rademacher complexity $N\hat{R}_S(\mathfrak{F}) = \mathbb{E}_{\boldsymbol{\varepsilon}} \sup_{f \in \mathfrak{F}} \sum_{i=1}^{N} \varepsilon_i f(\underline{\mathbf{x}}_i)$ satisfies

$$\frac{1}{\lambda} \log \exp \left( \lambda \cdot \mathbb{E}_{\boldsymbol{\varepsilon}} \sup_{f \in \mathfrak{F}} \sum_{i=1}^{N} \varepsilon_i f(\underline{\mathbf{x}}_i) \right) \leq \frac{1}{\lambda} \log \left( \mathbb{E}_{\boldsymbol{\varepsilon}} \sup_{f \in \mathfrak{F}} \exp \lambda \sum_{i=1}^{N} \varepsilon_i f(\underline{\mathbf{x}}_i) \right), \qquad (18)$$

where $\lambda > 0$ is an arbitrary parameter. Then, we can use a "peeling" argument [8, 37] as follows.

The Rademacher complexity can be upper bounded as

$$N\hat{R}_S(\mathfrak{F}) = \mathbb{E}_{\boldsymbol{\varepsilon}} \sup_{\mathbf{f}^{(L)}, \|\mathbf{w}\|_2 \leq B_w} \sum_{i=1}^{N} \varepsilon_i \mathbf{w}^\top \mathbf{f}^{(L)}(\underline{\mathbf{x}}_i)$$

$$\leq \frac{1}{\lambda} \log \mathbb{E}_{\boldsymbol{\varepsilon}} \sup_{\substack{\|\mathbf{w}\|_2 \leq B_w \\ \mathbf{f}^{(L-1)}, \|\underline{\mathbf{W}}^{(L)}\|_{\mathrm{F}} \leq B_L}} \exp \left( \lambda \sum_{i=1}^{N} \varepsilon_i \mathbf{w}^\top \sigma \left( \underline{\mathbf{W}}^{(L)} *_M \mathbf{f}^{(L-1)}(\underline{\mathbf{x}}_i) \right) \right)$$

$$\leq \frac{1}{\lambda} \log \mathbb{E}_{\boldsymbol{\varepsilon}} \sup_{\substack{\mathbf{f}^{(L-1)} \\ \|\underline{\mathbf{W}}^{(L)}\|_{\mathrm{F}} \leq B_L}} \exp \left( B_w \lambda \left\| \sum_{i=1}^{N} \varepsilon_i \sigma \left( \underline{\mathbf{W}}^{(L)} *_M \mathbf{f}^{(L-1)}(\underline{\mathbf{x}}_i) \right) \right\|_{\mathrm{F}} \right)$$

$$\leq \frac{1}{\lambda} \log \left( 2 \cdot \mathbb{E}_{\boldsymbol{\varepsilon}} \sup_{\substack{\mathbf{f}^{(L-2)} \\ \|\underline{\mathbf{W}}^{(L-1)}\|_{\mathrm{F}} \leq B_{L-1}}} \exp \left( B_w B_L \lambda \left\| \sum_{i=1}^{N} \varepsilon_i \left( \underline{\mathbf{W}}^{(L-1)} *_M \mathbf{f}^{(L-2)}(\underline{\mathbf{x}}_i) \right) \right\|_{\mathrm{F}} \right) \right)$$

$$\leq \cdots$$

$$\leq \frac{1}{\lambda} \log \left( 2^{L-1} \cdot \mathbb{E}_{\boldsymbol{\varepsilon}} \sup_{\|\underline{\mathbf{W}}^{(1)}\|_{\mathrm{F}} \leq B_1} \exp \left( B_w \prod_{l=2}^{L} B_l \lambda \left\| \sum_{i=1}^{N} \varepsilon_i \left( \underline{\mathbf{W}}^{(1)} *_M \mathbf{f}^{(0)}(\underline{\mathbf{x}}_i) \right) \right\|_{\mathrm{F}} \right) \right)$$

$$\leq \frac{1}{\lambda} \log \left( 2^L \cdot \mathbb{E}_{\boldsymbol{\varepsilon}} \exp \left( B_w \prod_{l=1}^{L} B_l \lambda \left\| \sum_{i=1}^{N} \varepsilon_i \underline{\mathbf{x}}_i \right\|_{\mathrm{F}} \right) \right).$$

$$(19)$$

Letting $B_{\underline{\mathbf{W}}} = B_w \prod_{l=1}^{L} B_l$, define a random variable

$$Z = B_{\underline{\mathbf{W}}} \cdot \left\| \sum_{i=1}^{N} \varepsilon_i \underline{\mathbf{x}}_i \right\|_{\mathrm{F}}$$

---

[8]This is a natural consequence of *(A.2)* in Assumption 2. See Eq. (7).

as a function of random variables $\{\varepsilon_i\}_i$. Then

$$\frac{1}{\lambda} \log\left(2^L \cdot \mathbb{E} \exp \lambda Z\right) = \frac{L \log 2}{\lambda} + \frac{1}{\lambda} \log\left(\mathbb{E} \exp \lambda(Z - \mathbb{E}Z)\right) + \mathbb{E}Z. \tag{20}$$

By Jensen's inequlity, $\mathbb{E}Z$ can be upper bounded by

$$\mathbb{E}Z = B_{\underline{\mathbf{W}}} \mathbb{E} \left\| \sum_{i=1}^N \varepsilon_i \underline{\mathbf{x}}_i \right\|_{\mathrm{F}} \le B_{\underline{\mathbf{W}}} \sqrt{\mathbb{E} \left\| \sum_{i=1}^N \varepsilon_i \underline{\mathbf{x}}_i \right\|_{\mathrm{F}}^2} = B_{\underline{\mathbf{W}}} \sqrt{\mathbb{E}_{\boldsymbol{\varepsilon}} \left[ \sum_{i,j=1}^N \varepsilon_i \varepsilon_j (\mathrm{vec}(\underline{\mathbf{x}}_i))^\top (\mathrm{vec}(\underline{\mathbf{x}}_j)) \right]}$$

$$= B_{\underline{\mathbf{W}}} \sqrt{\sum_{i=1}^N \|\underline{\mathbf{x}}_i\|_{\mathrm{F}}^2}.$$

To handle the $\log\left(\mathbb{E} \exp \lambda(Z - \mathbb{E}Z)\right)$ term in Eq. (20), note that $Z$ is a deterministic function of the *i.i.d.* random variables $\{\varepsilon_i\}_i$, and satisfies

$$Z(\varepsilon_1, \cdots, \varepsilon_i, \cdots, \varepsilon_m) - Z(\varepsilon_1, \cdots, -\varepsilon_i, \cdots, \varepsilon_m) \le 2B_{\underline{\mathbf{W}}} \|\underline{\mathbf{x}}_i\|_{\mathrm{F}}.$$

This means that $Z$ satisfies a bounded-difference condition, which by the proof of [5, Theorem 6.2], implies that $Z$ is sub-Gaussian, with the variance factor

$$v = \frac{1}{4}(2B_{\underline{\mathbf{W}}} \|\underline{\mathbf{x}}_i\|_{\mathrm{F}})^2 = B_{\underline{\mathbf{W}}}^2 \sum_{i=1}^N \|\underline{\mathbf{x}}_i\|_{\mathrm{F}}^2.$$

and satisfies

$$\frac{1}{\lambda} \log\left(\mathbb{E} \exp \lambda(Z - \mathbb{E}Z)\right) \le \frac{1}{\lambda} \frac{\lambda^2 B_{\underline{\mathbf{W}}}^2 \sum_{i=1}^N \|\underline{\mathbf{x}}_i\|_{\mathrm{F}}^2}{2} = \frac{\lambda B_{\underline{\mathbf{W}}}^2 \sum_{i=1}^N \|\underline{\mathbf{x}}_i\|_{\mathrm{F}}^2}{2}.$$

Choosing $\lambda = \frac{\sqrt{2L \log 2}}{B_{\underline{\mathbf{W}}} \sqrt{\sum_{i=1}^N \|\underline{\mathbf{x}}_i\|_{\mathrm{F}}^2}}$ and using the above inequality, we get that Eq. (19) can be upper bounded as follows

$$\frac{1}{\lambda} \log\left(2^L \cdot \mathbb{E} \exp \lambda Z\right) \le \mathbb{E}Z + \sqrt{2L \log 2} B_{\underline{\mathbf{W}}} \sqrt{\sum_{i=1}^N \|\underline{\mathbf{x}}_i\|_{\mathrm{F}}^2}$$

$$\le (\sqrt{2L \log 2} + 1) B_{\underline{\mathbf{W}}} \sqrt{\sum_{i=1}^N \|\underline{\mathbf{x}}_i\|_{\mathrm{F}}^2}.$$

Further applying Lemma 29 completes the proof. $\qquad\square$

### C.2 Adversarial Generalization Bound for t-NNs

*Proof of Theorem 5.* According to Theorem 2 and Eq. (4) in Ref. [3], the adversarial generalization gap of $\ell \circ f$ for any $f \in \mathfrak{F}$ with $L_\ell$-Lipschitz continuous loss function $\ell$ satisfying Assumption 2 can be upper bounded by $L_\ell \hat{R}_S(\mathfrak{F}^{\mathrm{adv}})$, where $\hat{R}_S(\mathfrak{F}^{\mathrm{adv}})$ is the empirical Rademacher complexity of the adversarial version $\tilde{\mathfrak{F}}^{\mathrm{adv}}$ of the function set $\mathfrak{F}$ defined as follows

$$\mathfrak{F}^{\mathrm{adv}} := \{\tilde{f} : (\underline{\mathbf{x}}, y) \mapsto \min_{R_{\mathrm{a}}(\underline{\mathbf{x}} - \underline{\mathbf{x}}') \le \xi} y f(\underline{\mathbf{x}}') \mid f \in \mathfrak{F}\}. \tag{21}$$

To bound $\hat{R}_S(\mathfrak{F}^{\mathrm{adv}})$, we use the Dudley's inequality (Lemma 31) which requires to compute the covering number of $\mathfrak{F}^{\mathrm{adv}}$.

Let $\mathbf{C}_l$ be the $\delta_l$-covering of $\{\underline{\mathbf{W}}^{(l)} \mid \|\underline{\mathbf{W}}^{(l)}\|_{\mathrm{F}} \le B_l\}$, $\forall l = 1, \cdots, L$. Consider the following subset of $\mathfrak{F}$ whose t-matrix weights are all in $\mathbf{C}_l$:

$$\mathfrak{F}_c := \left\{ f_c : \underline{\mathbf{x}} \mapsto f_{\mathcal{W}_c}(\underline{\mathbf{x}}) \mid \mathcal{W}_c = (\mathbf{w}, \underline{\mathbf{W}}_c^{(1)}, \cdots, \underline{\mathbf{W}}_c^{(L)}), \ \underline{\mathbf{W}}_c^{(l)} \in \mathbf{C}_l \right\}$$

with adversarial version

$$\mathfrak{F}_c^{\mathrm{adv}} := \left\{ \tilde{f}_c : (\mathbf{x}, y) \mapsto \inf_{R_{\mathrm{a}}(\mathbf{x}-\mathbf{x}') \leq \xi} y f_c(\mathbf{x}') \mid f_c \in \mathfrak{F}_c \right\}.$$

For all $\tilde{f} \in \mathfrak{F}^{\mathrm{adv}}$, we need to find the smallest distance to $\mathfrak{F}_c^{\mathrm{adv}}$, i.e. we need to calculate

$$\sup_{\tilde{f} \in \mathfrak{F}^{\mathrm{adv}}} \inf_{\tilde{f}_c \in \mathfrak{F}_c^{\mathrm{adv}}} \left\| \tilde{f} - \tilde{f}_c \right\|_S.$$

For all $(\mathbf{x}_i, y_i) \in S$, given $\tilde{f}$ and $\tilde{f}_c$ with $\left\| \underline{\mathbf{W}}^{(l)} - \underline{\mathbf{W}}_c^{(l)} \right\|_{\mathrm{F}} \leq \delta_l$, $l = 1, \cdots, L$, consider

$$|\tilde{f}(\mathbf{x}_i, y_i) - \tilde{f}_c(\mathbf{x}_i, y_i)| = \left| \inf_{R_{\mathrm{a}}(\mathbf{x}-\mathbf{x}') \leq \xi} y_i f(\mathbf{x}_i') - \inf_{R_{\mathrm{a}}(\mathbf{x}-\mathbf{x}') \leq \xi} y_i f_c(\mathbf{x}_i') \right|.$$

Letting $\underline{\mathbf{x}}_i^f = \mathrm{arginf}_{R_{\mathrm{a}}(\mathbf{x}_i - \mathbf{x}_i') \leq \xi} y_i f(\mathbf{x}_i')$ and $\underline{\mathbf{x}}_i^c = \mathrm{arginf}_{R_{\mathrm{a}}(\mathbf{x}_i - \mathbf{x}_i') \leq \xi} y_i f_c(\mathbf{x}_i')$, we have

$$|\tilde{f}(\mathbf{x}_i, y_i) - \tilde{f}_c(\mathbf{x}_i, y_i)| = |y_i f(\mathbf{x}_i^f) - y_i f_c(\mathbf{x}_i^c)|.$$

Let

$$\underline{\mathbf{x}}_i^\xi = \begin{cases} \underline{\mathbf{x}}_i^c & \text{if} \quad y_i f(\mathbf{x}_i^f) \geq y_i f_c(\mathbf{x}_i^c) \\ \underline{\mathbf{x}}_i^f & \text{if} \quad y_i f(\mathbf{x}_i^c) < y_i f_c(\mathbf{x}_i^c) \end{cases}.$$

Then,

$$|\tilde{f}(\mathbf{x}_i, y_i) - \tilde{f}_c(\mathbf{x}_i, y_i)| = |y_i f(\mathbf{x}_i^f) - y_i f_c(\mathbf{x}_i^g)| \leq |y_i f(\mathbf{x}_i^\xi) - y_i f_c(\mathbf{x}_i^\xi)| = |f(\mathbf{x}_i^\xi) - f_c(\mathbf{x}_i^\xi)|.$$

Let $g_l(\underline{\mathbf{x}}^\delta) = \mathbf{w}^\top \mathrm{vec}\Big( \sigma(\underline{\mathbf{W}}_c^{(L)} *_M \sigma(\underline{\mathbf{W}}_c^{(L-1)} *_M \cdots *_M \sigma(\underline{\mathbf{W}}_c^{(l+1)} *_M \sigma(\underline{\mathbf{W}}^{(l)} *_M \cdots *_M \sigma(\underline{\mathbf{W}}^{(1)} *_M$

$\underline{\mathbf{x}}^\delta) \cdots ))) \Big)$ and $g_0(\underline{\mathbf{x}}^\delta) = f_c(\underline{\mathbf{x}}^\delta)$. Then, we have

$$|f(\mathbf{x}_i^\xi) - f_c(\mathbf{x}_i^\xi)| \leq \sum_{l=1}^{L} |g_l(\mathbf{x}_i^\xi) - g_{l-1}(\mathbf{x}_i^\xi)|.$$

We can see that

$$\left\| \sigma(\underline{\mathbf{W}}^{(l)} *_M \cdots *_M \sigma(\underline{\mathbf{W}}^{(1)} *_M \underline{\mathbf{x}}_i^\xi) \cdots)) \right\|_{\mathrm{F}} \leq \prod_{l'=1}^{l} \left\| \underline{\mathbf{W}}^{(l')} \right\|_{\mathrm{F}} \left\| \underline{\mathbf{x}}_i^\xi \right\|_{\mathrm{F}} \leq \prod_{l'=1}^{l} B_l B_{x, R_{\mathrm{a}}, \xi},$$

and

$$\left\| \sigma(\underline{\mathbf{W}}_c^{(l+1)} *_M \sigma(\underline{\mathbf{W}}^{(l)} *_M \cdots *_M \sigma(\underline{\mathbf{W}}^{(1)} *_M \underline{\mathbf{x}}_i^\xi) \cdots)) - \sigma(\underline{\mathbf{W}}^{(l+1)} *_M \sigma(\underline{\mathbf{W}}^{(l)} *_M \cdots *_M \sigma(\underline{\mathbf{W}}^{(1)} *_M \underline{\mathbf{x}}_i^\xi) \cdots)) \right\|_{\mathrm{F}}$$

$$\leq \left\| \underline{\mathbf{W}}_c^{(l+1)} *_M \sigma(\underline{\mathbf{W}}^{(l)} *_M \cdots *_M \sigma(\underline{\mathbf{W}}^{(1)} *_M \underline{\mathbf{x}}_i^\xi) \cdots) - \underline{\mathbf{W}}^{(l+1)} *_M \sigma(\underline{\mathbf{W}}^{(l)} *_M \cdots *_M \sigma(\underline{\mathbf{W}}^{(1)} *_M \underline{\mathbf{x}}_i^\xi) \cdots) \right\|_{\mathrm{F}}$$

$$\leq \left\| (\underline{\mathbf{W}}_c^{(l+1)} - \underline{\mathbf{W}}^{(l+1)}) *_M \sigma(\underline{\mathbf{W}}^{(l)} *_M \cdots *_M \sigma(\underline{\mathbf{W}}^{(1)} *_M \underline{\mathbf{x}}_i^\xi) \cdots) \right\|_{\mathrm{F}}$$

$$\leq \left\| \underline{\mathbf{W}}_c^{(l+1)} - \underline{\mathbf{W}}^{(l+1)} \right\|_{\mathrm{F}} \left\| \sigma(\underline{\mathbf{W}}^{(l)} *_M \cdots *_M \sigma(\underline{\mathbf{W}}^{(1)} *_M \underline{\mathbf{x}}_i^\xi) \cdots) \right\|_{\mathrm{F}}$$

$$\leq \delta_l \cdot \prod_{l'=1}^{l-1} B_{l'} B_{x, R_{\mathrm{a}}, \xi}.$$

(22)

Then, we have

$$|g_l(\mathbf{x}_i^\xi) - g_{l-1}(\mathbf{x}_i^\xi)| \leq \delta_l \cdot B_{\mathbf{w}} \prod_{l' \neq l} B_{l'} B_{x, R_{\mathrm{a}}, \xi} = \frac{\delta_l B_{\tilde{f}}}{B_l},$$

where $B_{\tilde{f}}$ is given in Lemma 39.

Letting $\frac{\delta_l B_{\tilde{f}}}{B_l} = \frac{\epsilon}{L}$ it gives

$$\sup_{\tilde{f} \in \mathfrak{F}^{\mathrm{adv}}} \inf_{\tilde{f}_c \in \mathfrak{F}_c^{\mathrm{adv}}} \left\| \tilde{f} - \tilde{f}_c \right\|_S \leq \sum_{l=1}^{L-1} |g_l(\mathbf{x}_i^\xi) - g_{l-1}(\mathbf{x}_i^\xi)| \leq \epsilon.$$

Then, $\mathfrak{F}_c^{\mathrm{adv}}$ is an $\epsilon$-covering of $\mathfrak{F}^{\mathrm{adv}}$. We further proceed by computing the $\epsilon$-covering number of $\mathfrak{F}^{\mathrm{adv}}$ as follows:

$$\mathsf{N}(\mathfrak{F}^{\mathrm{adv}}, \|\cdot\|_S, \epsilon) \leq |\mathfrak{F}_c^{\mathrm{adv}}| = \prod_{l=1}^{L} |\mathsf{C}_l| \leq \prod_{l=1}^{L} \left( \frac{3B_l}{\delta_l} \right)^{cd_{l-1}d_l} \leq \left( \frac{3LB_{\tilde{f}}}{\epsilon} \right)^{\sum_{l=1}^{L} cd_{l-1}d_l}.$$

where the second inequality is due to Lemma 32.

Then, we use Dudley's integral in Lemma 31 to upper bound $\hat{R}_S(\mathfrak{F}^{\mathrm{adv}})$ as follows

$$\hat{R}_S(\mathfrak{F}^{\mathrm{adv}}) \leq \inf_{\delta > 0} \left( 8\delta + \frac{12}{\sqrt{N}} \int_\delta^{D_{\tilde{f}}/2} \sqrt{\log \mathsf{N}(\mathfrak{F}^{\mathrm{adv}}, \|\cdot\|_S, \epsilon)} \mathrm{d}\epsilon \right)$$

$$\leq \inf_{\delta > 0} \left( 8\delta + \frac{12}{\sqrt{N}} \int_\delta^{D_{\tilde{f}}/2} \sqrt{\left( \sum_{l=1}^{L} cd_{l-1}d_l \right) \log \left( 3LB_{\tilde{f}}/(\epsilon) \right)} \mathrm{d}\epsilon \right)$$

$$= \inf_{\delta > 0} \left( 8\delta + \frac{12 D_{\tilde{f}} \sqrt{\sum_{l=1}^{L} cd_{l-1}d_l}}{\sqrt{N}} \int_{\delta/D_{\tilde{f}}}^{1/2} \sqrt{\log \left( 3L/(2t) \right)} \mathrm{d}t \right),$$

where the diameter $D_{\tilde{f}}$ of $\mathfrak{F}^{\mathrm{adv}}$ is given in Lemma 40 and we can find from Lemma 40 that $D_{\tilde{f}} = 2B_{\tilde{f}}$ in our setting.

Following Ref. [55], let $\delta \to 0$, and use integration by part, we obtain $\int_0^{1/2} \sqrt{\log \left( 3L/(2t) \right)} \mathrm{d}t \leq \sqrt{\log 3L}$. Hence, we have

$$\hat{R}_S(\mathfrak{F}^{\mathrm{adv}}) \leq \frac{24 B_{\mathbf{w}} \prod_{l=1}^{L} B_l \left( \sqrt{\sum_{l=1}^{L} cd_{l-1}d_l} \right) B_{x, R_a, \xi}}{\sqrt{N}},$$

and the proof can be completed by using Lemma 29. $\qquad\square$

### C.3 Generalization Bound under Exact Low-tubal-rank Parameterization

*Proof of Theorem 6.* The idea is similar to the proof of Theorem 5. According to Theorem 2 and Eq. (4) in Ref. [3], the adversarial generalization gap of $\ell \circ f$ for any $f \in \mathfrak{F}_\mathbf{r}$ with $L_\ell$-Lipschitz continuous loss function $\ell$ satisfying Assumption 2 can be upper bounded by $L_\ell \hat{R}_S(\mathfrak{F}_\mathbf{r}^{\mathrm{adv}})$, where $\hat{R}_S(\mathfrak{F}_\mathbf{r}^{\mathrm{adv}})$ is the empirical Rademacher complexity of the adversarial version $\mathfrak{F}_\mathbf{r}^{\mathrm{adv}}$ of function set $\mathfrak{F}_\mathbf{r}$ defined as follows

$$\mathfrak{F}_\mathbf{r}^{\mathrm{adv}} := \{ \tilde{f} : (\mathbf{x}, y) \mapsto \min_{R_a(\mathbf{x} - \mathbf{x}') \leq \xi} y f(\mathbf{x}') \mid f \in \mathfrak{F}_\mathbf{r} \}. \tag{23}$$

To bound $\hat{R}_S(\mathfrak{F}_\mathbf{r}^{\mathrm{adv}})$, we first use the Dudley's inequality (Lemma 31) and compute the covering number of $\mathfrak{F}_\mathbf{r}^{\mathrm{adv}}$.

Let $\mathsf{C}_l$ be the $\delta_l$-covering of $\{ \underline{\mathbf{W}}^{(l)} \mid \left\| \underline{\mathbf{W}}^{(l)} \right\|_{\mathrm{F}} \leq B_l \text{ and } r_{\mathrm{t}}(\underline{\mathbf{W}}^{(l)}) \leq r_l \}$, $\forall l = 1, \cdots, L$. Consider the following subset of $\mathfrak{F}_\mathbf{r}$ whose t-matrix weights are all in $\mathsf{C}_l$:

$$\mathfrak{F}_c := \left\{ f_c : \underline{\mathbf{x}} \mapsto f(\underline{\mathbf{x}}; \underline{\mathbf{W}}_c) \mid \underline{\mathbf{W}}_c = (\mathbf{w}, \underline{\mathbf{W}}_c^{(1)}, \cdots, \underline{\mathbf{W}}_c^{(L)}), \ \underline{\mathbf{W}}_c^{(l)} \in \mathsf{C}_l \right\}$$

with adversarial version

$$\mathfrak{F}_c^{\mathrm{adv}} := \left\{ \tilde{f}_c : (\mathbf{x}, y) \mapsto \inf_{R_a(\mathbf{x} - \mathbf{x}') \leq \xi} y f_c(\mathbf{x}') \mid f_c \in \mathfrak{F}_c \right\}.$$

For all $\tilde{f} \in \mathfrak{F}_{\mathbf{r}}^{\mathrm{adv}}$, we need to find the smallest distance to $\mathfrak{F}_c^{\mathrm{adv}}$, i.e. we need to calculate

$$\sup_{\tilde{f} \in \mathfrak{F}_{\mathbf{r}}^{\mathrm{adv}}} \inf_{\tilde{f}_c \in \mathfrak{F}_c^{\mathrm{adv}}} \left\| \tilde{f} - \tilde{f}_c \right\|_S .$$

For all $(\mathbf{x}_i, y_i) \in S$, given $\tilde{f}$ and $\tilde{f}_c$ with $\left\| \underline{\mathbf{W}}^{(l)} - \underline{\mathbf{W}}_c^{(l)} \right\|_{\mathrm{F}} \leq \delta_l$, $l = 1, \cdots, L$, consider

$$|\tilde{f}(\mathbf{x}_i, y_i) - \tilde{f}_c(\mathbf{x}_i, y_i)| = \left| \inf_{R_{\mathrm{a}}(\mathbf{x} - \mathbf{x}') \leq \xi} y_i f(\mathbf{x}_i') - \inf_{R_{\mathrm{a}}(\mathbf{x} - \mathbf{x}') \leq \xi} y_i f_c(\mathbf{x}_i') \right|.$$

Letting $\mathbf{x}_i^f = \mathrm{arginf}_{R_{\mathrm{a}}(\mathbf{x}_i - \mathbf{x}_i') \leq \xi} y_i f(\mathbf{x}_i')$ and $\mathbf{x}_i^c = \mathrm{arginf}_{R_{\mathrm{a}}(\mathbf{x}_i - \mathbf{x}_i') \leq \xi} y_i f_c(\mathbf{x}_i')$, we have

$$|\tilde{f}(\mathbf{x}_i, y_i) - \tilde{f}_c(\mathbf{x}_i, y_i)| = |y_i f(\mathbf{x}_i^f) - y_i f_c(\mathbf{x}_i^c)|.$$

Let

$$\mathbf{x}_i^{\xi} = \begin{cases} \mathbf{x}_i^c & \text{if} \quad y_i f(\mathbf{x}_i^f) \geq y_i f_c(\mathbf{x}_i^c) \\ \mathbf{x}_i^f & \text{if} \quad y_i f(\mathbf{x}_i^c) < y_i f_c(\mathbf{x}_i^c) \end{cases}$$

Then,

$$|\tilde{f}(\mathbf{x}_i, y_i) - \tilde{f}_c(\mathbf{x}_i, y_i)| = |y_i f(\mathbf{x}_i^f) - y_i f_c(\mathbf{x}_i^g)| \leq |y_i f(\mathbf{x}_i^{\xi}) - y_i f_c(\mathbf{x}_i^{\xi})| = |f(\mathbf{x}_i^{\xi}) - f_c(\mathbf{x}_i^{\xi})|.$$

Let $g_l(\mathbf{x}^{\delta}) = \mathbf{w}^{\top} \mathrm{vec}\left( \sigma(\underline{\mathbf{W}}_c^{(L)} *_M \sigma(\underline{\mathbf{W}}_c^{(L-1)} *_M \cdots *_M \sigma(\underline{\mathbf{W}}_c^{(l+1)} *_M \sigma(\underline{\mathbf{W}}^{(l)} *_M \cdots *_M \sigma(\underline{\mathbf{W}}^{(1)} *_M$

$\mathbf{x}^{\delta}) \cdots ))) \right)$ and $g_0(\mathbf{x}^{\delta}) = f_c(\mathbf{x}^{\delta})$. Then, we have

$$|f(\mathbf{x}_i^{\xi}) - f_c(\mathbf{x}_i^{\xi})| \leq \sum_{l=1}^L |g_l(\mathbf{x}_i^{\xi}) - g_{l-1}(\mathbf{x}_i^{\xi})|.$$

We can see that

$$\left\| \sigma(\underline{\mathbf{W}}^{(l)} *_M \cdots *_M \sigma(\underline{\mathbf{W}}^{(1)} *_M \mathbf{x}_i^{\xi}) \cdots )) \right\|_{\mathrm{F}} \leq \prod_{l'=1}^l \left\| \underline{\mathbf{W}}^{(l')} \right\|_{\mathrm{F}} \left\| \mathbf{x}_i^{\xi} \right\|_{\mathrm{F}} \leq \prod_{l'=1}^l B_l B_{x, R_{\mathrm{a}}, \xi}$$

and

$$\left\| \sigma(\underline{\mathbf{W}}_c^{(l+1)} *_M \sigma(\underline{\mathbf{W}}^{(l)} *_M \cdots *_M \sigma(\underline{\mathbf{W}}^{(1)} *_M \mathbf{x}_i^{\xi}) \cdots )) - \sigma(\underline{\mathbf{W}}^{(l+1)} *_M \sigma(\underline{\mathbf{W}}^{(l)} *_M \cdots *_M \sigma(\underline{\mathbf{W}}^{(1)} *_M \mathbf{x}_i^{\xi}) \cdots )) \right\|_{\mathrm{F}}$$

$$\leq \left\| \underline{\mathbf{W}}_c^{(l+1)} *_M \sigma(\underline{\mathbf{W}}^{(l)} *_M \cdots *_M \sigma(\underline{\mathbf{W}}^{(1)} *_M \mathbf{x}_i^{\xi}) \cdots ) - \underline{\mathbf{W}}^{(l+1)} *_M \sigma(\underline{\mathbf{W}}^{(l)} *_M \cdots *_M \sigma(\underline{\mathbf{W}}^{(1)} *_M \mathbf{x}_i^{\xi}) \cdots ) \right\|_{\mathrm{F}}$$

$$\leq \left\| (\underline{\mathbf{W}}_c^{(l+1)} - \underline{\mathbf{W}}^{(l+1)}) *_M \sigma(\underline{\mathbf{W}}^{(l)} *_M \cdots *_M \sigma(\underline{\mathbf{W}}^{(1)} *_M \mathbf{x}_i^{\xi}) \cdots ) \right\|_{\mathrm{F}}$$

$$\leq \left\| \underline{\mathbf{W}}_c^{(l+1)} - \underline{\mathbf{W}}^{(l+1)} \right\|_{\mathrm{F}} \left\| \sigma(\underline{\mathbf{W}}^{(l)} *_M \cdots *_M \sigma(\underline{\mathbf{W}}^{(1)} *_M \mathbf{x}_i^{\xi}) \cdots ) \right\|_{\mathrm{F}}$$

$$\leq \delta_l \cdot \prod_{l'=1}^{l-1} B_{l'} B_{x, R_{\mathrm{a}}, \xi}.$$

Then, we have

$$|g_l(\mathbf{x}_i^{\xi}) - g_{l-1}(\mathbf{x}_i^{\xi})| \leq \delta_l \cdot B_{\mathbf{w}} \prod_{l' \neq l} B_{l'} B_{x, R_{\mathrm{a}}, \xi} = \frac{\delta_l B_{\tilde{f}}}{B_l}.$$

Letting $\frac{\delta_l B_{\tilde{f}}}{B_l} = \frac{\epsilon}{L}$ gives

$$\sup_{\tilde{f} \in \mathfrak{F}_{\mathbf{r}}^{\mathrm{adv}}} \inf_{\tilde{f}_c \in \mathfrak{F}_c^{\mathrm{adv}}} \left\| \tilde{f} - \tilde{f}_c \right\|_S \leq \sum_{l=1}^{L-1} |g_l(\mathbf{x}_i^{\xi}) - g_{l-1}(\mathbf{x}_i^{\xi})| \leq \epsilon.$$

Then, $\mathfrak{F}_c^{\text{adv}}$ is an $\epsilon$-covering of $\mathfrak{F}_{\mathbf{r}}^{\text{adv}}$. We further proceed by computing the $\epsilon$-covering number of $\mathfrak{F}_{\mathbf{r}}^{\text{adv}}$ as follows:

$$
\mathsf{N}(\mathfrak{F}_{\mathbf{r}}^{\text{adv}}, \|\cdot\|_S, \epsilon) \leq |\mathfrak{F}_c^{\text{adv}}| = \prod_{l=1}^{L} |\mathsf{C}_l| \overset{(i}{\leq} \prod_{l=1}^{L} \left(\frac{9B_l}{\delta_l}\right)^{\mathsf{c}r_l(d_{l-1}+d_l+1)} \leq \left(\frac{9LB_{\tilde{f}}}{\epsilon}\right)^{\sum_{l=1}^{L} \mathsf{c}r_l(d_{l-1}+d_l+1)},
$$

where the inequality $(i)$ holds due to Lemma 33.

Then, we use Dudley's integral to upper bound $\hat{R}_S(\mathfrak{F}_{\mathbf{r}}^{\text{adv}})$ as follows

$$
\hat{R}_S(\mathfrak{F}_{\mathbf{r}}^{\text{adv}}) \leq \inf_{\delta > 0} \left(8\delta + \frac{12}{\sqrt{N}} \int_{\delta}^{D_{\tilde{f}}/2} \sqrt{\log \mathsf{N}(\mathfrak{F}_{\mathbf{r}}^{\text{adv}}, \|\cdot\|_S, \epsilon)} \mathrm{d}\epsilon\right)
$$

$$
\leq \inf_{\delta > 0} \left(8\delta + \frac{12}{\sqrt{N}} \int_{\delta}^{D_{\tilde{f}}/2} \sqrt{\sum_{l=1}^{L} \mathsf{c}r_l(d_{l-1}+d_l+1)\log\left(9LB_{\tilde{f}}/\epsilon\right)} \mathrm{d}\epsilon\right)
$$

$$
= \inf_{\delta > 0} \left(8\delta + \frac{12D_{\tilde{f}}\sqrt{\sum_{l=1}^{L} \mathsf{c}r_l(d_{l-1}+d_l+1)}}{\sqrt{N}} \int_{\delta/D_{\tilde{f}}}^{1/2} \sqrt{\log\left(9L/(2t)\right)} \mathrm{d}t\right),
$$

where the diameter $D_{\tilde{f}}$ of $\mathfrak{F}^{\text{adv}}$ is given in Lemma 40 and we have $D_{\tilde{f}} = 2B_{\tilde{f}}$. Following Ref. [55], let $\delta \to 0$, and use interation by part, we obtain $\int_0^{1/2} \sqrt{\log\left(9L/(2t)\right)} \mathrm{d}t \leq \sqrt{\log 9L}$. Further applying Lemma 29 completes the proof. $\qquad\square$

## D  Implicit bias towards low-rankness in the transformed domain

Recent research has shown that GF maximizes the margin of homogeneous networks during standard training, which leads to an implicit bias towards margin maximization [16, 30]. Moreover, it has been demonstrated that this implicit bias also extends to adversarial margin maximization during adversarial training of multi-homogeneous fully connected neural networks with exponential loss [29]. Our analysis builds on these findings by showing that this implicit bias also holds for adversarial training of t-NNs when the adversarial perturbation is scale invariant [29].

First, it is straightforward to see that any t-NN $f \in \mathfrak{F}$ is homogeneous as follows

$$
f(\underline{\mathbf{x}}; a\underline{\mathbf{W}}) = a^{L+1} f(\underline{\mathbf{x}}; \underline{\mathbf{W}}), \tag{24}
$$

for any positive constant $a$.

**Lemma 18** (Euler's theorem on t-NNs). *For any t-NN $f \in \mathfrak{F}$, we have*

$$
\left\langle \frac{\partial f(\underline{\mathbf{x}}; \underline{\mathbf{W}})}{\partial \underline{\mathbf{W}}}, \underline{\mathbf{W}} \right\rangle = (L+1)f(\underline{\mathbf{x}}; \underline{\mathbf{W}}). \tag{25}
$$

*Proof.* By taking derivatives with respect to $a$ on both sides of Eq. (24), we obtain

$$
\left\langle \frac{\partial f(\underline{\mathbf{x}}; a\underline{\mathbf{W}})}{\partial (a\underline{\mathbf{W}})}, \frac{\mathrm{d}(a\underline{\mathbf{W}})}{\mathrm{d}a} \right\rangle = (L+1)a^L f(\underline{\mathbf{x}}; \underline{\mathbf{W}}),
$$

which immediately results in Eq. (25). $\qquad\square$

In this section, we follow the setting of Ref. [29] where the adversarial perturbation $\underline{\boldsymbol{\delta}}_i$ is scale invariant. As Lemma 7 shows, $l_2$-FGM [34], FGSM [9], $l_2$-PGD and $l_\infty$-PGD [31] perturbations for the t-NNs are all scale invariant.

*Proof of Lemma 7.* Note that by taking derivatives with respect to $\underline{\mathbf{x}}$ on both sides of Eq. (24), we have

$$
\frac{\partial f(\underline{\mathbf{x}}; a\underline{\mathbf{W}})}{\partial \underline{\mathbf{x}}} = a^{L+1} \frac{\partial f(\underline{\mathbf{x}}; \underline{\mathbf{W}})}{\partial \underline{\mathbf{x}}}, \tag{26}
$$

Therefore, any $\frac{\partial f(\underline{\mathbf{x}}; \underline{\mathbf{W}})}{\partial \underline{\mathbf{x}}}$ is positive homogeneous. Then, for any non-zero $\underline{\mathbf{z}} = \frac{\partial f(\underline{\mathbf{x}}; \underline{\mathbf{W}})}{\partial \underline{\mathbf{x}}}$, we prove Lemma 7 in the following cases:

- $l_2$-FGM perturbtion [34]. The $l_2$-FGM perturbtion is defined as $\boldsymbol{\delta}_{\text{FGM}}(\underline{\mathbf{W}}) = \xi y \tilde{\ell}' \mathbf{z} \left\| y \tilde{\ell}' \mathbf{z} \right\|_{\text{F}}^{-1} = -\xi y \mathbf{z} \|\mathbf{z}\|_{\text{F}}^{-1}$, because $\tilde{\ell}' \leq 0$. Using Eq. (26), we have

$$\boldsymbol{\delta}_{\text{FGM}}(a\underline{\mathbf{W}}) = -\frac{\xi y \cdot a^{L+1} \mathbf{z}}{\|a^{L+1} \mathbf{z}\|_{\text{F}}} = \boldsymbol{\delta}_{\text{FGM}}(\underline{\mathbf{W}}).$$

- FGSM perturbations [9]. The FGSM perturbtion is taken as $\boldsymbol{\delta}_{\text{FSGM}}(\underline{\mathbf{W}}) = \xi \operatorname{sgn}(\xi y \tilde{\ell}' \mathbf{z})$. Using Eq. (26), we have

$$\boldsymbol{\delta}_{\text{FSGM}}(a\underline{\mathbf{W}}) = \xi \operatorname{sgn}(\xi y \tilde{\ell}' a^{L+1} \mathbf{z}) = \boldsymbol{\delta}_{\text{FSGM}}(\underline{\mathbf{W}}).$$

- $l_2$-PGD perturbation [31]. The $l_2$-PGD perturbtion is taken as

$$\boldsymbol{\delta}_{\text{PGD}}^{j+1}(\underline{\mathbf{W}}) = \mathfrak{P}_{\mathbb{B}_2(0,\xi)} \left[ \boldsymbol{\delta}_{\text{PGD}}^{j}(\underline{\mathbf{W}}) - \rho \frac{\xi y \mathbf{z}}{\|\mathbf{z}\|_{\text{F}}} \right], \tag{27}$$

where $j$ is the attack step, $\mathfrak{P}_{\mathbb{B}_2(\xi)}$ is the projector onto $l_2$-norm ball of radius $\xi$, and $\rho$ is the learning rate. We prove by induction. For $j = 0$, we have

$$\boldsymbol{\delta}_{\text{PGD}}^{1}(a\underline{\mathbf{W}}) = \mathfrak{P}_{\mathbb{B}_2(0,\xi)} \left[ -\rho \frac{\xi y a^{L+1} \mathbf{z}}{\|a^{L+1} \mathbf{z}\|_{\text{F}}} \right] = \boldsymbol{\delta}_{\text{PGD}}^{1}(\underline{\mathbf{W}}).$$

If we have $\boldsymbol{\delta}_{\text{PGD}}^{j}(a\underline{\mathbf{W}}) = \boldsymbol{\delta}_{\text{PGD}}^{j}(\underline{\mathbf{W}})$, then for $j + 1$, we have

$$\begin{aligned}
\boldsymbol{\delta}_{\text{PGD}}^{j+1}(a\underline{\mathbf{W}}) &= \mathfrak{P}_{\mathbb{B}_2(0,\xi)} \left[ \boldsymbol{\delta}_{\text{PGD}}^{j}(a\underline{\mathbf{W}}) - \rho \frac{\xi y a^{L+1} \mathbf{z}}{\|a^{L+1} \mathbf{z}\|_{\text{F}}} \right] \\
&= \mathfrak{P}_{\mathbb{B}_2(0,\xi)} \left[ \boldsymbol{\delta}_{\text{PGD}}^{j}(\underline{\mathbf{W}}) - \rho \frac{\xi y \mathbf{z}}{\|\mathbf{z}\|_{\text{F}}} \right] \\
&= \boldsymbol{\delta}_{\text{PGD}}^{j+1}(\underline{\mathbf{W}}).
\end{aligned} \tag{28}$$

- $l_\infty$-PGD perturbation [31]. Since the scale invariance of this pertubation can be proved very similarly to that of $l_2$-PGD perturbations, we just omit it.

$\square$

For an original example $\underline{\mathbf{x}}_i$, the margin for its adversarial example $\underline{\mathbf{x}}_i + \boldsymbol{\delta}_i(\underline{\mathbf{W}})$ is defined as $\tilde{q}_i(\underline{\mathbf{W}}) := y_i f(\underline{\mathbf{x}}_i + \boldsymbol{\delta}_i(\underline{\mathbf{W}}); \underline{\mathbf{W}})$; for sample $S = \{(\underline{\mathbf{x}}_i, y_i)\}_{i=1}^N$, the margin for the $N$ corresponding examples is denoted by $\tilde{q}_m(\underline{\mathbf{W}})$ where $m \in \operatorname{argmin}_{m=1,\cdots,N} y_i f(\underline{\mathbf{x}}_i + \boldsymbol{\delta}_i(\underline{\mathbf{W}}) : \underline{\mathbf{W}})$.

Let $\rho = \|\underline{\mathbf{W}}\|_{\text{F}}$ for simplicity. We use the normalized parameter $\underline{\hat{\mathbf{W}}} := \underline{\mathbf{W}}/\rho$ to denote the direction of the weights $\underline{\mathbf{W}}$. We introduce the normalized margin of $(\underline{\mathbf{x}}_i, y_i)$ as $\hat{q}_i(\underline{\mathbf{W}}) := \tilde{q}_i(\underline{\hat{\mathbf{W}}}) = \tilde{q}_i(\underline{\mathbf{W}}) \rho^{-(L+1)}$, and similarly define the normalized robust margin of the sample $S$ as $\hat{q}_m(\underline{\mathbf{W}}) := \tilde{q}_m(\underline{\hat{\mathbf{W}}}) = \tilde{q}_m(\underline{\mathbf{W}}) \rho^{-(L+1)}$.

Note that the adversarial empirical risk can be written as $\hat{\mathcal{L}}^{\text{adv}} = \frac{1}{N} \sum_{i=1}^N e^{-\mathfrak{f}(\tilde{q}_i(\underline{\mathbf{W}}))}$. Motivated by [30] which uses the LogSumExp function to smoothly approximate the normalized standard margin, we define the smoothed normalized margin $\tilde{\gamma}$ as follows.

**Definition 11** (Smoothed normalized robust margin). *For a loss function[9] $\ell$ satisfying Assumption 2, the smoothed normalized robust margin is defined as*

$$\tilde{\gamma}(\underline{\mathbf{W}}) := \frac{\ell^{-1}(N\hat{\mathcal{L}}^{\text{adv}})}{\rho^{L+1}} = \frac{\mathfrak{g}(\frac{1}{N\hat{\mathcal{L}}^{\text{adv}}})}{\rho^{L+1}} = \frac{\mathfrak{g}\left(-\log\left(\sum_{i=1}^N e^{-\mathfrak{f}(\tilde{q}_i(\underline{\mathbf{W}}))}\right)\right)}{\rho^{L+1}}. \tag{29}$$

---

[9]In this paper, the loss function satisfying Assumption 2 belongs to the class of *margin-based loss function* [42, Definition 2.24]. That is, although the loss function $\ell(f(\underline{\mathbf{x}}), y)$ is a binary function of $f(\underline{\mathbf{x}})$ and $y$, there is a unary function $\mathfrak{l}(\cdot)$, such that $\ell(f(\underline{\mathbf{x}}), y) = \mathfrak{l}(y f(\underline{\mathbf{x}}))$. With a slight abuse of notation, we simply use $\ell^{-1}(\cdot)$ to denote $\mathfrak{l}^{-1}(\cdot)$, i.e., if $z = \ell(f(\underline{\mathbf{x}}), y)$ then we have $\ell^{-1}(z) = y f(\underline{\mathbf{x}})$.

To better understand the relation between the normalized sample robust margin $\hat{q}_m$ and the smoothed normalized robust margin $\tilde{\gamma}$, we provide the following lemma.

**Lemma 19** (Adapted from Lemma A.5 of Ref. [30]). *Under Assumption 2, we have the following properties about the robust marin $\tilde{q}_m$:*

*(a)* $\mathfrak{f}(\tilde{q}_m) - \log N \leq \log \frac{1}{N\hat{\mathcal{L}}^{\mathrm{adv}}} \leq \mathfrak{f}(\tilde{q}_m)$.

*(b) If $\log \frac{1}{N\hat{\mathcal{L}}^{\mathrm{adv}}} > \mathfrak{f}(b_{\mathfrak{f}})$, then there exists $\xi \in (\mathfrak{f}(\tilde{q}_m) - \log N, \mathfrak{f}(\tilde{q}_m)) \cap (b_{\mathfrak{f}}, \infty)$ such that*

$$\hat{q}_m - \rho^{-(L+1)}\mathfrak{g}'(\xi) \log N \leq \tilde{\gamma} \leq \hat{q}_m,$$

*which shows the smoothed normalized margin $\tilde{\gamma}$ is a rough approximation of the normalized robust margin $\hat{q}_m$.*

*(c) For a sequence $\{\underline{\mathbf{W}}_s \mid s \in \mathbb{N}\}$, if $\hat{\mathcal{L}}^{\mathrm{adv}}(\underline{\mathbf{W}}_s) \to 0$, then $|\tilde{\gamma}(\underline{\mathbf{W}}_s) - \hat{q}_m(\underline{\mathbf{W}}_s)| \to 0$.*

### D.1 Convergence to KKT points of Euclidean norm minimization in direction

The KKT condition for the optimization problem Eq. (14) are

$$\underline{\mathbf{W}} - \sum_{i=1}^{N} \lambda_i \frac{\partial \hat{q}_i}{\partial \underline{\mathbf{W}}} = \mathbf{0}, \tag{30}$$

$$\lambda_i(\tilde{q}_i - 1) = 0, \quad i = 1, \cdots, N,$$

where the dual variables $\lambda_i \geq 0$. We define the approximate KKT point in a similar manner to Ref. [29] as follows.

**Definition 12** (Approximate KKT points). *The $(\kappa, \iota)$-approximate KKT points of the optimization problem are those feasible points which satisfy the following two conditions:*

$$\text{Condition (I):} \qquad \left\| \underline{\mathbf{W}} - \sum_{i=1}^{N} \lambda_i \frac{\partial \tilde{q}_i}{\partial \underline{\mathbf{W}}} \right\|_{\mathrm{F}} \leq \kappa, \tag{31}$$

$$\text{Condition (II):} \qquad \lambda_i(\tilde{q}_i - 1) \leq \iota, \ \ i = 1, \cdots, N,$$

*where $\kappa, \iota > 0$ and $\lambda_i \geq 0$.*

*Proof of Lemma 9.* Let $\check{\underline{\mathbf{W}}} := \underline{\mathbf{W}}/\tilde{q}_m^{\frac{1}{L+1}}$ denote the scaled version of $\underline{\mathbf{W}}(t)$ such that the sample robust margin $\tilde{q}_m = 1$. Thus we have $f(\mathbf{x}; \underline{\mathbf{W}}) = \tilde{q}_m f(\mathbf{x}; \check{\underline{\mathbf{W}}})$ by homogeneity of t-NNs. According to Lemma 18, we further have

$$(L+1)f(\underline{\mathbf{x}}; \underline{\mathbf{W}}) = \left\langle \underline{\mathbf{W}}, \frac{\partial f(\mathbf{x}; \underline{\mathbf{W}})}{\partial \underline{\mathbf{W}}} \right\rangle, \quad (L+1)f(\underline{\mathbf{x}}; \check{\underline{\mathbf{W}}}) = \left\langle \check{\underline{\mathbf{W}}}, \frac{\partial f(\underline{\mathbf{x}}; \check{\underline{\mathbf{W}}})}{\partial \check{\underline{\mathbf{W}}}} \right\rangle,$$

leading to

$$\left\langle \check{\underline{\mathbf{W}}}, \frac{\partial f(\mathbf{x}; \check{\underline{\mathbf{W}}})}{\partial \check{\underline{\mathbf{W}}}} \right\rangle = \frac{1}{\hat{q}_m^{L/(L+1)}} \frac{\partial f(\mathbf{x}; \underline{\mathbf{W}})}{\partial \underline{\mathbf{W}}}.$$

We will prove that $\check{\underline{\mathbf{W}}}$ is a $(\kappa, \iota)$-KKT point of Problem (14) with $(\kappa, \iota) \to \mathbf{0}$.

Let $\dot{\underline{\mathbf{W}}}(t) := \frac{d\mathbf{W}(t)}{dt}$ for simplicity. By the chain rule and GF update rule, we have

$$\dot{\underline{\mathbf{W}}} = \frac{1}{N} \sum_{i=1}^{N} e^{-\mathfrak{f}(\tilde{q}_i)} \mathfrak{f}'(\tilde{q}_i) \frac{\partial \tilde{q}_i}{\partial \underline{\mathbf{W}}}.$$

Using the homogeneity of t-NNs, we obtain

$$\frac{1}{2} \frac{d\|\mathbf{W}\|_{\mathrm{F}}^2}{dt} = \left\langle \dot{\underline{\mathbf{W}}}, \underline{\mathbf{W}} \right\rangle = \left\langle \frac{1}{N} \sum_{i=1}^{N} e^{-\mathfrak{f}(\tilde{q}_i)} \mathfrak{f}'(\tilde{q}_i) \frac{\partial \tilde{q}_i}{\partial \underline{\mathbf{W}}}, \underline{\mathbf{W}} \right\rangle = (L+1) \cdot \frac{1}{N} \sum_{i=1}^{N} e^{-\mathfrak{f}(\tilde{q}_i)} \mathfrak{f}'(\tilde{q}_i) \tilde{q}_i.$$

By letting $\nu(t) = \sum_{i=1}^{N} e^{-\mathfrak{f}(\tilde{q}_i)} \mathfrak{f}'(\tilde{q}_i) \tilde{q}_i$, we obtain $\left\langle \dot{\underline{\mathbf{W}}}, \underline{\mathbf{W}} \right\rangle = (L+1)\nu/N$.

We construct the dual variables $\lambda_i$ in Problem (31) in terms of $\dot{\underline{\mathbf{W}}}$ as follows

$$\lambda_i(t) := \frac{1}{N}\tilde{q}_m^{1-2/(L+1)} \cdot \rho e^{-\mathfrak{f}(\bar{q}_i)}\mathfrak{f}'(\tilde{q}_i)\left\|\dot{\underline{\mathbf{W}}}\right\|_{\mathrm{F}}^{-1}. \tag{32}$$

To prove $\underline{\check{\mathbf{W}}}$ is a $(\kappa, \iota)$-KKT point of Problem (14), we need to check the conditions in Problem (31).

**Step 1: Check Condition (I) of the $(\kappa, \iota)$-approximate KKT conditions**. We check the Condition (I) in Problem (31) for all $t > t_0$ as follows

$$\left\|\underline{\check{\mathbf{W}}} - \sum_{i=1}^{N}\lambda_i\frac{\partial\tilde{q}_i(\underline{\check{\mathbf{W}}})}{\partial\underline{\check{\mathbf{W}}}}\right\|_{\mathrm{F}}^2 \overset{(i)}{=} \left\|\frac{\mathbf{W}}{\tilde{q}_m^{1/(L+1)}} - \frac{\dot{\mathbf{W}}}{\left\|\dot{\mathbf{W}}\right\|_{\mathrm{F}}}\cdot\frac{\rho}{\tilde{q}_m^{1/(L+1)}}\right\|_{\mathrm{F}}^2$$

$$= \frac{\rho^2}{\tilde{q}_m^{2/(L+1)}}\left\|\frac{\mathbf{W}}{\|\mathbf{W}\|_{\mathrm{F}}} - \frac{\dot{\mathbf{W}}}{\left\|\dot{\mathbf{W}}\right\|_{\mathrm{F}}}\right\|_{\mathrm{F}}^2$$

$$\overset{(ii)}{=} \frac{1}{\tilde{\gamma}^{2/(L+1)}}\left(2 - 2\left\langle\frac{\mathbf{W}}{\|\mathbf{W}\|_{\mathrm{F}}}, \frac{\dot{\mathbf{W}}}{\left\|\dot{\mathbf{W}}\right\|_{\mathrm{F}}}\right\rangle\right) \tag{33}$$

$$\overset{(iii)}{\leq} \frac{2}{\tilde{\gamma}(t_0)^{2/(L+1)}}\left(1 - \left\langle\frac{\mathbf{W}}{\|\mathbf{W}\|_{\mathrm{F}}}, \frac{\dot{\mathbf{W}}}{\left\|\dot{\mathbf{W}}\right\|_{\mathrm{F}}}\right\rangle\right)$$

$$:= \kappa^2(t),$$

where equality $(i)$ is obtained by using the definition that $\underline{\check{\mathbf{W}}} = \mathbf{W}/\tilde{q}_m^{\frac{1}{L+1}}$, the fact $\tilde{q}_i(\mathbf{W}) = y_i f(\mathbf{x}_i + \boldsymbol{\delta}_i(\mathbf{W}); \mathbf{W}) = y_i \cdot \tilde{q}_m f(\mathbf{x}_i + \boldsymbol{\delta}_i(\underline{\check{\mathbf{W}}}); \underline{\check{\mathbf{W}}}) = \tilde{q}_i(\underline{\check{\mathbf{W}}})$ due to the scale invariance of the adversarial perturbation $\boldsymbol{\delta}_i(\mathbf{W})$ and the homogeneity of t-NN, and the chain rule in computing $\frac{\partial\tilde{q}_i}{\partial\underline{\mathbf{W}}}$ as follows

$$\sum_{i=1}^{N}\lambda_i\frac{\partial\tilde{q}_i(\underline{\check{\mathbf{W}}})}{\partial\underline{\check{\mathbf{W}}}} = \sum_{i=1}^{N}\frac{\lambda_i}{\hat{q}_m^{\frac{L}{L+1}}}\frac{\partial\tilde{q}_i}{\partial\mathbf{W}} = \left\|\dot{\mathbf{W}}\right\|_{\mathrm{F}}^{-1}\frac{\rho}{\tilde{q}_m^{\frac{L}{L+1}}}\cdot\frac{1}{N}\sum_{i=1}^{N}e^{-\mathfrak{f}(\bar{q}_i)}\mathfrak{f}'(\tilde{q}_i)\frac{\partial\tilde{q}_i}{\partial\mathbf{W}} = \frac{\dot{\mathbf{W}}}{\left\|\dot{\mathbf{W}}\right\|_{\mathrm{F}}}\frac{\rho}{\tilde{q}_m^{\frac{L}{L+1}}}.$$

In Eq. (33), equality $(ii)$ holds by property $(b)$ in Lemma 19; $(iii)$ holds by the non-decreasing property of $\tilde{\gamma}(t)$ for all $t \in [t_0, \infty)$ in Lemma 22.

Note that Eq. (33) indicates that $\kappa(t)$ is in terms of the cosine of the angle between $\mathbf{W}(t)$ and $\dot{\mathbf{W}}(t)$. We can further obtain that $\kappa(t) \to 0$ as $t \to \infty$ by showing the angle between $\mathbf{W}(t)$ and $\dot{\mathbf{W}}(t)$ approximates $0$ which was orignially observed by Ref. [30] for standard training[10] on a fixed training sample $S$.

**Lemma 20** (Adapted from Lemma C.12 in Ref. [30]). *Under Assumption 2 and Assumption 8 for t-NNs, the angle between $\underline{\mathbf{W}}(t)$ and $\dot{\underline{\mathbf{W}}}(t)$ approximates $0$ as $t \to \infty$ along the trajectory of adversarial training with scale invariant adversarial pertubations, i.e.,*

$$\lim_{t\to\infty}\left\langle\frac{\mathbf{W}(t)}{\|\mathbf{W}(t)\|_{\mathrm{F}}}, \frac{\dot{\mathbf{W}}(t)}{\left\|\dot{\mathbf{W}}(t)\right\|_{\mathrm{F}}}\right\rangle \to 1.$$

Note that according to Assumption 8 we have $\tilde{\gamma}(t_0) \geq b_{\mathfrak{f}}\rho(t_0)^{-(L+1)}$ which means $\tilde{\gamma}(t_0)$ cannot be arbitrarily close to $0$. Thus, by invoking Lemma 20, we obtain that

$$\lim_{t\to\infty}\kappa^2(t) = \lim_{t\to\infty}\frac{2}{\tilde{\gamma}(t_0)^{2/(L+1)}}\left(1 - \left\langle\frac{\mathbf{W}}{\|\mathbf{W}\|_{\mathrm{F}}}, \frac{\dot{\mathbf{W}}}{\left\|\dot{\mathbf{W}}\right\|_{\mathrm{F}}}\right\rangle\right) = 0. \tag{34}$$

---

[10]The Lemma C.12 in [30] which was intended for standard training, can be safely extended to the adversarial settings in this paper. This is because by our construction for adversarial training with scale invariant adversarial perturbations, the adversarial traing margin are locally Lipschitz and the prediction function $f(\mathbf{x} + \boldsymbol{\delta}(\mathbf{W}); \mathbf{W})$ is positively homogeneous with respect to $\underline{\mathbf{W}}$.

**Step 2: Check Condition (II) of the $(\kappa, \iota)$-approximate KKT conditions**. We check the Condition (II) in Problem (31) for all $t > t_0$ as follows

$$\sum_{i=1}^{N} \lambda_i(\tilde{q}_i(\check{\mathbf{W}}) - 1) \overset{(i)}{=} \sum_{i=1}^{N} \frac{1}{N} \tilde{q}_m^{1-2/(L+1)} \rho e^{-\mathfrak{f}(\tilde{q}_i)} \mathfrak{f}'(\tilde{q}_i) \left\| \dot{\underline{\mathbf{W}}} \right\|_{\mathrm{F}}^{-1} \cdot (\tilde{q}_i(\mathbf{W}) \tilde{q}_m^{-1} - 1)$$

$$= \rho \left\| \dot{\underline{\mathbf{W}}} \right\|_{\mathrm{F}}^{-1} \tilde{q}_m^{-2/(L+1)} \cdot \frac{1}{N} \sum_{i=1}^{N} e^{-\mathfrak{f}(\tilde{q}_i)} \mathfrak{f}'(\tilde{q}_i)(\tilde{q}_i - \tilde{q}_m),$$

(35)

where $(i)$ is due to the definition that $\check{\underline{\mathbf{W}}} = \underline{\mathbf{W}}/\tilde{q}_m^{\frac{1}{L+1}}$ and the fact that $\tilde{q}_i(\mathbf{W}) = y_i f(\mathbf{x}_i + \underline{\boldsymbol{\delta}}_i(\mathbf{W}); \mathbf{W}) = y_i \cdot \tilde{q}_m f(\mathbf{x}_i + \underline{\boldsymbol{\delta}}_i(\check{\mathbf{W}}); \check{\mathbf{W}}) = \tilde{q}_i(\check{\mathbf{W}})$ due to the scale invariance of the adversarial pertabation $\underline{\boldsymbol{\delta}}_i(\mathbf{W})$ and the homogenouty of t-NN.

To upper bound Eq. (35), first note that $\left\| \dot{\underline{\mathbf{W}}} \right\|_{\mathrm{F}} \geq \left\langle \dot{\underline{\mathbf{W}}}, \hat{\underline{\mathbf{W}}} \right\rangle = \left\langle \frac{1}{N} \sum_{i=1}^{N} e^{-\mathfrak{f}(\tilde{q}_i)} \mathfrak{f}'(\tilde{q}_i) \frac{\partial \tilde{q}_i}{\partial \underline{\mathbf{W}}}, \hat{\underline{\mathbf{W}}} \right\rangle = \rho^{-1}(L+1)\nu/N$, in which $\nu$ can be further lower bounded as

$$\nu \overset{(i)}{\geq} \frac{\mathfrak{g}(\log \frac{1}{N\hat{\mathcal{L}}^{\mathrm{adv}}})}{\mathfrak{g}'(\log \frac{1}{N\hat{\mathcal{L}}^{\mathrm{adv}}})} N\hat{\mathcal{L}}^{\mathrm{adv}} \overset{(ii)}{\geq} \frac{1}{2K} \log \frac{1}{N\hat{\mathcal{L}}^{\mathrm{adv}}} \cdot N\hat{\mathcal{L}}^{\mathrm{adv}} \overset{(iii)}{\geq} \frac{1}{2K} e^{-\mathfrak{f}(\hat{q}_m)} \log \frac{1}{N\hat{\mathcal{L}}^{\mathrm{adv}}}, \quad (36)$$

where $(i)$ is due to Lemma 23, $(ii)$ holds because of Lemma 24, and $(iii)$ uses $N\hat{\mathcal{L}}^{\mathrm{adv}} = \sum_i e^{-\mathfrak{f}(\tilde{q}_i)} \geq e^{-\mathfrak{f}(\tilde{q}_m)}$. Combing Eq. (35) and Eq. (36) yields

$$\sum_i \lambda_i(\tilde{q}_i(\check{\mathbf{W}}) - 1) \leq \frac{2K\tilde{q}_m^{-2/(L+1)}\rho^2}{(L+1)\log \frac{1}{N\hat{\mathcal{L}}^{\mathrm{adv}}}} \sum_{i=1}^{N} e^{\mathfrak{f}(\tilde{q}_m)-\mathfrak{f}(\tilde{q}_i)} \mathfrak{f}'(\tilde{q}_i)(\tilde{q}_i - \tilde{q}_m)$$

$$\overset{(i)}{\geq} \frac{2K\tilde{\gamma}^{-2/(L+1)}}{(L+1)\log \frac{1}{N\hat{\mathcal{L}}^{\mathrm{adv}}}} \sum_{i=1}^{N} e^{\mathfrak{f}(\tilde{q}_m)-\mathfrak{f}(\tilde{q}_i)} \mathfrak{f}'(\tilde{q}_i)(\tilde{q}_i - \tilde{q}_m),$$

(37)

where $(i)$ uses $\tilde{q}_m^{-2/(L+1)}\rho^2 \leq \tilde{\gamma}^{-2/(L+1)}$ by Lemma 19.

In Eq. (37), if $\tilde{q}_i > \tilde{q}_m$, then there exists an $\xi_i \in (\tilde{q}_m, \tilde{q}_i)$ such that $\mathfrak{f}(\tilde{q}_m) - \mathfrak{f}(\tilde{q}_i) = \mathfrak{f}'(\xi_i)(\tilde{q}_i - \tilde{q}_m)$ by the mean value theorem. Further, we know that $\mathfrak{f}'(\tilde{q}_i) \leq K^{\lceil \log_2(\tilde{q}_i/\xi_i) \rceil} \mathfrak{f}'(\xi_i)$ by Assumption 2. Note that $\lceil \log_2(\tilde{q}_i/\xi_i) \rceil \leq \log_2(2B_0\rho^{L+1}/\tilde{q}_m) \leq \log_2(2B_0/\tilde{\gamma})$, where

$$B_0(t) := \sup \left\{ \tilde{q}_i \rho^{-(L+1)} \mid \underline{\mathbf{W}} \neq \mathbf{0} \right\} = \sup \left\{ \tilde{q}_i \mid \|\underline{\mathbf{W}}\|_{\mathrm{F}} = 1 \right\}.$$

Then, for all $t > t_0$, we have

$$\sum_i \lambda_i(\tilde{q}_i(\check{\underline{\mathbf{W}}}) - 1) \leq \frac{2K\tilde{\gamma}^{-2/(L+1)}\rho^2}{(L+1)\log \frac{1}{N\hat{\mathcal{L}}^{\mathrm{adv}}}} K^{\log_2(2B_0/\tilde{\gamma})} \sum_{i:\tilde{q}_i \neq \tilde{q}_m} e^{\mathfrak{f}'(\xi_i)(\tilde{q}_i - \tilde{q}_m)} \mathfrak{f}'(\xi_i)(\tilde{q}_i - \tilde{q}_m)$$

$$\overset{(i)}{\leq} \frac{2K\tilde{\gamma}^{-2/(L+1)}}{(L+1)\log \frac{1}{N\hat{\mathcal{L}}^{\mathrm{adv}}}} K^{\log_2(2B_0/\tilde{\gamma})} \cdot Ne$$

$$\overset{(ii)}{\leq} \frac{2KNe\tilde{\gamma}^{-2/(L+1)}}{(L+1)\log \frac{1}{N\hat{\mathcal{L}}^{\mathrm{adv}}}} \left( \frac{B_0}{\tilde{\gamma}} \right)^{\log_2(2K)}$$

(38)

$$\overset{(iii)}{\leq} \frac{2KNe}{(L+1)\tilde{\gamma}(t_0)^{-2/(L+1)}} \left( \frac{B_0}{\tilde{\gamma}(t_0)} \right)^{\log_2(2K)} \cdot \log \frac{1}{N\hat{\mathcal{L}}^{\mathrm{adv}}}$$

$$:= \iota(t),$$

where $(i)$ holds because the function $x \mapsto e^{-z}z$ on $(0, \infty)$ has maximum $e$ at $z = 1$; $(ii)$ is due to $a^{\log_c b} = b^{\log_c a}$; $(iii)$ holds by the non-decreasing property of $\tilde{\gamma}(t)$ for all $t \in [t_0, \infty)$ in Lemma 22. Note that by Lemma 22, we have $\lim_{t \to \infty} \hat{\mathcal{L}}^{\mathrm{adv}}(t) = 0$, which further yields

$$\lim_{t \to \infty} \iota(t) = 0. \quad (39)$$

**Step 3: Check the condition for convergence to KKT point**. According to Eq. (34) and Eq. (39), the limit point of $\check{\underline{\mathbf{W}}}(t)$ satisfy the $(\kappa, \iota)$-approximate KKT conditions of Problem 14 along the

trajectory of adversarial training of t-NN with scale invariant adversarial perturbations where $\lim_{t\to\infty}(\kappa(t), \iota(t)) = \mathbf{0}$. Then, we need to check the condition between $(\kappa, \iota)$-approximate points and KKT points.

According to Ref. [30], the KKT condition becomes a necessary condition for global optimality of Problem (14) when the Mangasarian-Fromovitz Constraint Qualification (MFCQ) [33] is satisfied. It is straightforward to see that Problem (14) satisfies the MFCQ condition, i.e.,

$$\left\langle \frac{\partial \tilde{q}_i}{\partial \underline{\mathbf{W}}}, \underline{\mathbf{W}} \right\rangle = (L+1)\tilde{q}_i \geq 0.$$

at every feasible point $\underline{\mathbf{W}}$. Then restating the theorem in Ref. [7] regarding the relation between $(\kappa, \iota)$-approximate KKT point and KKT point in our setting yields the following result.

**Theorem 21** (Theorem 3.6 in Ref. [7] and Theorem C.4 in Ref. [30])**.** *Let $\{\underline{\check{\mathbf{W}}}(j) \mid j \in \mathbb{N}\}$ be a sequence of feasible point of Problem (14), and $\underline{\check{\mathbf{W}}}(j)$ is a $(\kappa(j), \iota(j))$-approximate KKT point for all $j$ with two sequences[11] $\{\kappa(j) > 0 \mid j \in \mathbb{N}\}$ and $\{\iota(j) > 0 \mid j \in \mathbb{N}\}$ and $\lim_{j\to\infty}(\kappa(j), \iota(j)) = \mathbf{0}$. If $\lim_{j\to\infty} \underline{\check{\mathbf{W}}}(j) = \underline{\check{\mathbf{W}}}^*$, and MFCQ holds at $\underline{\check{\mathbf{W}}}^*$, then $\underline{\check{\mathbf{W}}}^*$ is a KKT point of Problem (14).*

Recall that $\underline{\check{\mathbf{W}}} = \underline{\mathbf{W}}/\tilde{q}_m^{\frac{1}{L+1}}$. Then, it can be concluded that the limit point of $\{\underline{\mathbf{W}}(t)/\|\underline{\mathbf{W}}(t)\|_F : t > 0\}$ of GF for empirical adversarial risk $\hat{\mathcal{L}}^{\mathrm{adv}}(\underline{\mathbf{W}}) := \frac{1}{N}\sum_{i=1}^N e^{-\mathfrak{f}(y_i f(\mathbf{x}_i + \underline{\boldsymbol{\delta}}_i(\underline{\mathbf{W}}); \underline{\mathbf{W}}))}$ with scale invariant perturbations $\underline{\boldsymbol{\delta}}_i$ is aligned with the direction of a KKT point of Problem (14). $\qquad\square$

### D.2 Technical Lemmas for Proving Lemma 9

**Lemma 22.** *Under Assumption 2, we have the following statements for GF-based adversarial training in Eq. (13) with scale invariant perturbations:*

(**I**). *For a.e. $t \in (t_0, \infty)$, the smoothed normalized robust margin $\tilde{\gamma}(\underline{\mathbf{W}}(t))$ defined in Eq. (29) is non-decreasing, i.e.,*

$$\frac{\mathrm{d}\tilde{\gamma}(\underline{\mathbf{W}}(t))}{\mathrm{d}t} \geq 0.$$

(**II**). *The adversarial objective $\hat{\mathcal{L}}^{\mathrm{adv}}(\underline{\mathbf{W}}) := \frac{1}{N}\sum_{i=1}^N e^{-\mathfrak{f}(y_i f(\mathbf{x}_i + \underline{\boldsymbol{\delta}}_i(\underline{\mathbf{W}}); \underline{\mathbf{W}}))}$ with scale invariant pertubations $\underline{\boldsymbol{\delta}}_i$ converges to zero as $t \to \infty$, i.e.,*

$$\lim_{t\to\infty} \hat{\mathcal{L}}^{\mathrm{adv}}(\underline{\mathbf{W}}(t)) = 0, \tag{40}$$

*and the Euclidean norm of the t-NN weights diverges, i.e.,*

$$\lim_{t\to\infty} \|\underline{\mathbf{W}}(t)\|_F = \infty. \tag{41}$$

*Proof of Lemma 22.* We follow the idea of Ref. [30] to prove Lemma 22 as follows.

**Step 1: Prove Part (I)**. We prove **(I)** by showing the following results for all $t \geq t_0$,

$$\frac{\mathrm{d}\log\rho}{\mathrm{d}t} > 0 \quad \text{and} \quad \frac{\mathrm{d}\log\tilde{\gamma}}{\mathrm{d}t} \geq (L+1)\left(\frac{\mathrm{d}\log\rho}{\mathrm{d}t}\right)^{-1}\left\|\frac{\mathrm{d}\hat{\underline{\mathbf{W}}}}{\mathrm{d}t}\right\|_F.$$

Recalling the quantity $\nu(t) = \sum_{i=1}^N e^{-\mathfrak{f}(\tilde{q}_i)}\mathfrak{f}(\tilde{q}_i)\tilde{q}_i$, by chain rule we obtain

$$\frac{\mathrm{d}\log\rho}{\mathrm{d}t} = \frac{1}{2\rho^2}\frac{\mathrm{d}\rho^2}{\mathrm{d}t} = \frac{1}{\rho^2}\left\langle \frac{1}{N}\sum_{i=1}^N e^{-\mathfrak{f}(\tilde{q}_i)}\mathfrak{f}'(\tilde{q}_i)\frac{\partial \tilde{q}_i}{\partial \underline{\mathbf{W}}}, \underline{\mathbf{W}} \right\rangle = \frac{\nu(L+1)}{\rho^2 N} \overset{(i)}{\geq} 0,$$

where $(i)$ holds due to Lemma 23.

---

[11]Using the same method to [30, Lemma C.12], we can construct the two sequences, i.e., $\{\kappa(j) > 0 \mid j \in \mathbb{N}\}$ and $\{\iota(j) > 0 \mid j \in \mathbb{N}\}$, based on Eq. (33) and Eq. (35).

By the chaining rule, we also have

$$
\begin{aligned}
\frac{\mathrm{d}\log\tilde{\gamma}}{\mathrm{d}t} &= \frac{\mathrm{d}}{\mathrm{d}t}\left(\log\left(\log\frac{1}{N\hat{\mathcal{L}}^{\mathrm{adv}}}\right) - (L+1)\log\rho\right) \\
&= \frac{\mathfrak{g}'(\log\frac{1}{N\hat{\mathcal{L}}^{\mathrm{adv}}})}{\mathfrak{g}(\log\frac{1}{N\hat{\mathcal{L}}^{\mathrm{adv}}})}\cdot\frac{1}{N\hat{\mathcal{L}}^{\mathrm{adv}}}\cdot\left(-\frac{\mathrm{d}N\hat{\mathcal{L}}^{\mathrm{adv}}}{\mathrm{d}t}\right) - (L+1)^2\cdot\frac{\nu(t)}{\rho^2} \\
&= \frac{1}{\nu(t)}\cdot\left(-\frac{\mathrm{d}N\hat{\mathcal{L}}^{\mathrm{adv}}}{\mathrm{d}t}\right) - (L+1)^2\cdot\frac{\nu(t)}{\rho^2} \\
&= \frac{1}{\nu(t)}\cdot\left(-\frac{\mathrm{d}N\hat{\mathcal{L}}^{\mathrm{adv}}}{\mathrm{d}t} - (L+1)^2\cdot\frac{\nu(t)^2}{\rho^2}\right).
\end{aligned}
$$

Note that according to Eq. (13), we have

$$
\frac{\mathrm{d}\hat{\mathcal{L}}^{\mathrm{adv}}}{\mathrm{d}t} = \left\langle\frac{\partial\hat{\mathcal{L}}^{\mathrm{adv}}}{\partial\underline{\mathbf{W}}},\frac{\mathrm{d}\underline{\mathbf{W}}}{\mathrm{d}t}\right\rangle = -\left\|\frac{\mathrm{d}\underline{\mathbf{W}}}{\mathrm{d}t}\right\|_{\mathrm{F}}^2,
$$

for $t \geq 0$ almost everywhere. One the other hand, we have

$$
L\nu(t) = \left\langle\underline{\mathbf{W}},\frac{\mathrm{d}\underline{\mathbf{W}}}{\mathrm{d}t}\right\rangle.
$$

Thus, we obtain

$$
\frac{\mathrm{d}\log\tilde{\gamma}}{\mathrm{d}t} \geq \frac{1}{N\nu}\left(\left\|\frac{\mathrm{d}\underline{\mathbf{W}}}{\mathrm{d}t}\right\|_{\mathrm{F}}^2 - \left\langle\hat{\underline{\mathbf{W}}},\frac{\mathrm{d}\underline{\mathbf{W}}}{\mathrm{d}t}\right\rangle^2\right) \geq \frac{1}{N\nu(t)}\left\|(\mathbf{I}-\mathrm{vec}(\hat{\underline{\mathbf{W}}})\mathrm{vec}(\hat{\underline{\mathbf{W}}})^\top)\mathrm{vec}(\frac{\mathrm{d}\underline{\mathbf{W}}}{\mathrm{d}t})\right\|^2.
$$

By the chain rule,

$$
\frac{\mathrm{d}\hat{\underline{\mathbf{W}}}}{\mathrm{d}t} = \frac{1}{\rho}(\mathbf{I}-\mathrm{vec}(\hat{\underline{\mathbf{W}}})\mathrm{vec}(\hat{\underline{\mathbf{W}}})^\top)\mathrm{vec}(\frac{\mathrm{d}\underline{\mathbf{W}}}{\mathrm{d}t}),
$$

for $t > 0$ allmost everywhere. So, we have

$$
\frac{\mathrm{d}}{\mathrm{d}t}\log\tilde{\gamma} \geq \frac{\rho^2}{N\nu(t)}\left\|\frac{\mathrm{d}\hat{\underline{\mathbf{W}}}}{\mathrm{d}t}\right\|_{\mathrm{F}}^2 = (L+1)\left(\frac{\mathrm{d}}{\mathrm{d}t}\log\rho\right)^{-1}\left\|\frac{\mathrm{d}\hat{\underline{\mathbf{W}}}}{\mathrm{d}t}\right\|_{\mathrm{F}}^2 \geq 0.
$$

**Step 2: Prove Part (II)**. Motivated by [30, Lemma B.8], we prove **(II)** as follows. First, note that

$$
-\frac{\mathrm{d}\hat{\mathcal{L}}^{\mathrm{adv}}}{\mathrm{d}t} = \left\|\frac{\mathrm{d}\underline{\mathbf{W}}}{\mathrm{d}t}\right\|_{\mathrm{F}}^2 \geq \left\langle\hat{\underline{\mathbf{W}}},\dot{\underline{\mathbf{W}}}\right\rangle^2 = \rho^{-2}N^{-2}(L+1)^2\nu^2.
$$

By lower bounding $\nu$ with Lemma 23 and replacing $\rho$ with $(\mathfrak{g}(\log\frac{1}{N\hat{\mathcal{L}}^{\mathrm{adv}}})/\tilde{\gamma})^{1/(L+1)}$ by the definition smoothed normalized robust margin of $\tilde{\gamma}$ in Eq. (29), we obtain

$$
\begin{aligned}
-\frac{\mathrm{d}\hat{\mathcal{L}}^{\mathrm{adv}}}{\mathrm{d}t} &\geq (L+1)^2\left(\frac{\mathfrak{g}(\log\frac{1}{N\hat{\mathcal{L}}^{\mathrm{adv}}})}{\mathfrak{g}'(\log\frac{1}{N\hat{\mathcal{L}}^{\mathrm{adv}}})}N\hat{\mathcal{L}}^{\mathrm{adv}}\right)^2\cdot\left(\frac{\tilde{\gamma}}{\mathfrak{g}(\log\frac{1}{N\hat{\mathcal{L}}^{\mathrm{adv}}})}\right)^{2/(L+1)} \\
&\geq (L+1)^2\tilde{\gamma}(t_0)^{2/(L+1)}\cdot\frac{\mathfrak{g}(\log\frac{1}{N\hat{\mathcal{L}}^{\mathrm{adv}}})^{2-2/(L+1)}}{\mathfrak{g}'(\log\frac{1}{N\hat{\mathcal{L}}^{\mathrm{adv}}})^2}\cdot(N\hat{\mathcal{L}}^{\mathrm{adv}})^2,
\end{aligned}
$$

where the last inequality holds due to the non-decreasing property of $\tilde{\gamma}$ in Part **(I)**. Then, we obtain

$$
\frac{\mathfrak{g}'(\log\frac{1}{N\hat{\mathcal{L}}^{\mathrm{adv}}})^2}{\mathfrak{g}(\log\frac{1}{N\hat{\mathcal{L}}^{\mathrm{adv}}})^{2-2/(L+1)}}\cdot\frac{\mathrm{d}}{\mathrm{d}t}\frac{1}{N\hat{\mathcal{L}}^{\mathrm{adv}}} \geq (L+1)^2\tilde{\gamma}(t_0)^{2/(L+1)}.
$$

By integrating on both sides from $t_0$ to $t$, we obtain

$$
G\left(\frac{1}{N\hat{\mathcal{L}}^{\mathrm{adv}}}\right) \geq (L+1)^2\tilde{\gamma}(t_0)^{2/(L+1)}(t-t_0), \tag{42}
$$

where

$$G(z) := \int_{1/(N\hat{\mathcal{L}}^{\mathrm{adv}}(t_0))}^{z} \frac{\mathfrak{g}'(\log u)^2}{g(\log u)^{2-2/(L+1)}} \mathrm{d}u.$$

We use proof by contradiction to show the empirical training risk $\hat{\mathcal{L}}^{\mathrm{adv}}$ converges to zero. Note that $\frac{1}{(N\hat{\mathcal{L}}^{\mathrm{adv}})}$ is non-decreasing. If $\frac{1}{(N\hat{\mathcal{L}}^{\mathrm{adv}})}$ does not grow to $\infty$, then neither does $G(\frac{1}{(N\hat{\mathcal{L}}^{\mathrm{adv}})})$. But the RHS of Eq. (42) grows to $\infty$, which is a contradiction. Therefore, $\lim_{t\to\infty} N\hat{\mathcal{L}}^{\mathrm{adv}} = 0$. Hence $\lim_{t\to\infty} \hat{\mathcal{L}}^{\mathrm{adv}}(t) = 0$ and $\lim_{t\to\infty} \rho(t) = \infty$. $\square$

**Lemma 23** (Adapted from Lemma B.5 of Ref. [30]). *The quantity $\nu = \sum_i e^{-\mathfrak{f}(\tilde{q}_i)}\mathfrak{f}(\tilde{q}_i)\tilde{q}_i$ has a lower bound for all $t \in (t_0, \infty)$,*

$$\nu(t) \geq \frac{\mathfrak{g}(\log \frac{1}{N\hat{\mathcal{L}}^{\mathrm{adv}}})}{\mathfrak{g}'(\log \frac{1}{N\hat{\mathcal{L}}^{\mathrm{adv}}})} N\hat{\mathcal{L}}^{\mathrm{adv}}. \tag{43}$$

**Lemma 24** (Lemma D.1 of Ref. [30]). *For $\mathfrak{f}(\cdot)$ and $\mathfrak{g}(\cdot)$ in Assumption 2, we have*

$$\frac{\mathfrak{g}(x)}{\mathfrak{g}'(x)} \in \left[\frac{1}{2K}x, 2Kx\right], \forall x \in [b_{\mathfrak{g}}, \infty) \quad \text{and} \quad \frac{\mathfrak{f}(q)}{\mathfrak{f}'(q)} \in \left[\frac{1}{2K}q, 2Kq\right], \forall q \in [\mathfrak{g}(b_{\mathfrak{g}}), \infty).$$

### D.3  Proof of Theorem 10

Note that the t-NN $g(\underline{\mathbf{x}}; \underline{\mathbf{V}})$ with weights $\underline{\mathbf{V}} = (\underline{\mathbf{V}}^{(1)}, \cdots, \underline{\mathbf{V}}^{(J)}, \mathbf{v})$ in Theorem 6 has the following structure

$$g(\underline{\mathbf{x}}; \underline{\mathbf{V}}) = \mathbf{v}^{\top}\mathrm{vec}(\mathbf{g}(\underline{\mathbf{x}}))$$
$$\mathbf{g}(\underline{\mathbf{x}}) = \mathbf{g}^{(J)}(\underline{\mathbf{x}}) \in \mathbb{R}^{m_J \times 1 \times \mathsf{c}}$$
$$\mathbf{g}^{(j)}(\underline{\mathbf{x}}) = \sigma(\underline{\mathbf{V}}^{(j)} *_M \mathbf{g}^{(j-1)}(\underline{\mathbf{x}})) \in \mathbb{R}^{m_j \times 1 \times \mathsf{c}}, \ \forall j = 1, \cdots, J$$
$$\mathbf{g}^{(0)}(\underline{\mathbf{x}}) = \underline{\mathbf{x}}.$$

Let $\alpha = (\frac{1}{B_v})^{\frac{L-J}{L+1}}$ and $\beta = (\frac{1}{B_v})^{-\frac{J+1}{L+1}}$, where $L$ is a sufficiently large integer greater than $J$. We then construct a t-NN $h(\underline{\mathbf{x}}; \underline{\mathbf{H}})$ of $L$ t-product layers which perfectly realizes $g(\underline{\mathbf{x}}; \underline{\mathbf{V}})$. Specifically, we construct $h$ with weights $\underline{\mathbf{H}} = (\underline{\mathbf{H}}^{(1)}, \cdots, \underline{\mathbf{H}}^{(L)}, \mathbf{h})$ satisfying the following equation

$$\underline{\mathbf{H}} = (\underbrace{\alpha\underline{\mathbf{V}}^{(1)}, \cdots, \alpha\underline{\mathbf{V}}^{(J)}}_{\text{first } J \text{ t-product layers}}, \underbrace{\beta\underline{\mathbf{I}}, \cdots, \beta\underline{\mathbf{I}}}_{\text{last } (L-J) \text{ t-product layers}}, \mathbf{v}),$$

or more clearly

$$h(\underline{\mathbf{x}}; \underline{\mathbf{V}}) = \mathbf{v}^{\top}\mathrm{vec}(\mathbf{h}(\underline{\mathbf{x}}))$$
$$\mathbf{h}(\underline{\mathbf{x}}) = \mathbf{h}^{(L)}(\underline{\mathbf{x}}) \in \mathbb{R}^{m_J \times 1 \times \mathsf{c}}$$
$$\mathbf{h}^{(l)}(\underline{\mathbf{x}}) = \sigma(\beta\underline{\mathbf{I}} *_M \mathbf{h}^{(l-1)}(\underline{\mathbf{x}})) \in \mathbb{R}^{m_J \times 1 \times \mathsf{c}}, \ \forall l = J+1, \cdots, L$$
$$\mathbf{h}^{(j)}(\underline{\mathbf{x}}) = \sigma(\alpha\underline{\mathbf{V}}^{(j)} *_M \mathbf{h}^{(j-1)}(\underline{\mathbf{x}})) \in \mathbb{R}^{m_j \times 1 \times \mathsf{c}}, \ \forall j = 1, \cdots, J$$
$$\mathbf{h}^{(0)}(\underline{\mathbf{x}}) = \underline{\mathbf{x}},$$

where $\underline{\mathbf{I}}$ is the t-identity tensor.

It is easy to prove that for any input $\underline{\mathbf{x}}$, the input and output of $g(\underline{\mathbf{x}}; \underline{\mathbf{V}}) = h(\underline{\mathbf{x}}; \underline{\mathbf{H}})$, it can also be proved that

$$\min_{R_{\mathrm{a}}(\underline{\mathbf{x}}-\underline{\mathbf{x}}') \leq \xi} yg(\underline{\mathbf{x}}'; \underline{\mathbf{V}}) = \min_{R_{\mathrm{a}}(\underline{\mathbf{x}}-\underline{\mathbf{x}}') \leq \xi} yh(\underline{\mathbf{x}}'; \underline{\mathbf{H}}).$$

Therefore, $h(\underline{\mathbf{x}}; \underline{\mathbf{H}})$ can also robustly classify $(\underline{\mathbf{x}}_i, y_i)_{i=1}^{N}$ because $g(\underline{\mathbf{x}}; \underline{\mathbf{V}})$ can robustly classify $(\underline{\mathbf{x}}_i, y_i)_{i=1}^{N}$.

Then, we consider the class of over-parameterized t-NNs $\mathfrak{F} = \{f(\underline{\mathbf{x}}; \underline{\mathbf{W}})\}$ defined in Eq. (14) with dimensionality of weight $\underline{\mathbf{W}}^{(l)} \in \mathbb{R}^{d_l \times d_{l-1} \times \mathsf{c}}$ safisfying $d_l \gg \max_{j \leq J}\{m_j\}$ for all $l = 1, \cdots, L$. Specifically, we construct $f$ with weights

$$\underline{\mathbf{W}} = (\underline{\mathbf{W}}^{(1)}, \cdots, \underline{\mathbf{W}}^{(L)}, \mathbf{w}),$$

and structure

$$f(\mathbf{x}; \underline{\mathbf{W}}) = \mathbf{w}^\top \text{vec}(\mathbf{f}(\mathbf{x}))$$
$$\mathbf{f}(\mathbf{x}) = \mathbf{f}^{(L)}(\mathbf{x}) \in \mathbb{R}^{d_L \times 1 \times \mathsf{c}}$$
$$\mathbf{f}^{(l)}(\mathbf{x}) = \sigma(\underline{\mathbf{W}}^{(l)} *_M \mathbf{f}^{(l-1)}(\mathbf{x})) \in \mathbb{R}^{d_l \times 1 \times \mathsf{c}}, \forall l = 1, \cdots, L$$
$$\mathbf{f}^{(0)}(\mathbf{x}) = \mathbf{x}.$$

Note that according to our construction there is a function $f(\mathbf{x}; \underline{\mathbf{W}}_h) \in \mathfrak{F}$ with weights $\underline{\mathbf{W}}_h = (\underline{\mathbf{W}}_h^{(1)}, \cdots, \underline{\mathbf{W}}_h^{(L)}, \mathbf{w}_h)$ satisfying

$$(\underline{\mathbf{W}}_h^{(l)})_{i_1,i_2,i_3} = \begin{cases} (\underline{\mathbf{H}}^{(l)})_{i_1,i_2,i_3} & \text{if } i_1 \le m_l, i_2 \le m_{l-1}, l \le J \\ (\underline{\mathbf{H}}^{(l)})_{i_1,i_2,i_3} & \text{if } i_1 \le m_J, i_2 \le m_J, l = J+1, \cdots, L \\ 0 & \text{otherwise} \end{cases}$$

and

$$\mathbf{w}_i = \begin{cases} \mathbf{h}_i & \text{if } i \le \mathsf{c}m_J \\ 0 & \text{otherwise} \end{cases}.$$

We can also see that $h'(\mathbf{x} + \underline{\boldsymbol{\delta}}) = h(\mathbf{x} + \underline{\boldsymbol{\delta}})$ for any $\mathbf{x} \in \mathbb{R}^{d \times 1 \times \mathsf{c}}$ and any $\underline{\boldsymbol{\delta}}$ satisfying $R_\mathrm{a}(\underline{\boldsymbol{\delta}}) \le \xi$. Thus, we can say that the weight $\underline{\mathbf{W}}_h$ of $f(\mathbf{x}; \underline{\mathbf{W}}_h)$ is a feasible solution to Problem (14), i.e.,

$$\min_{R_\mathrm{a}(\underline{\boldsymbol{\delta}}_i) \le \xi} y_i f(\mathbf{x}_i + \underline{\boldsymbol{\delta}}_i; \underline{\mathbf{W}}_h) \ge 1, \forall i = 1, \cdots, N.$$

Now consider the optimal solution $\underline{\mathbf{W}}^* = (\underline{\mathbf{W}}^{*(1)}, \cdots, \underline{\mathbf{W}}^{*(L)}, \mathbf{w}^*)$ to Problem (14). Then according to the optimility of $\underline{\mathbf{W}}^*$ and the feasibility of $\underline{\mathbf{W}}_h$ to Problem 14, we have

$$\|\underline{\mathbf{W}}^*\|_\mathrm{F}^2 \le \|\underline{\mathbf{W}}_h\|_\mathrm{F}^2 \le \alpha^2 \cdot B_v^2 \cdot (J+1) + \beta^2 \cdot (\mathsf{c}m_J) \cdot (L-J)$$
$$= B_v^{\frac{2(J+1)}{L+1}} (K + 1 + (\mathsf{c}m_J)(L-J)) \tag{44}$$

and

$$\min_{R_\mathrm{a}(\underline{\boldsymbol{\delta}}_i) \le \xi} y_i f(\mathbf{x}_i + \underline{\boldsymbol{\delta}}_i; \underline{\mathbf{W}}^*) \ge 1, \forall i = 1, \cdots, N. \tag{45}$$

As there is an example $(\mathbf{x}^*, y^*)$ satisfying $\|\mathbf{x}^*\|_\mathrm{F} \le 1$ in the training set $S = \{(\mathbf{x}_i, y_i)\}_{i=1}^N \subseteq \mathbb{R}^{d \times 1 \times \mathsf{c}} \times \{\pm 1\}$. Then according to Eq. (45), we have

$$y^* f(\mathbf{x}^*; \underline{\mathbf{W}}^*) = y^* f(\mathbf{x}^* + \mathbf{0}; \underline{\mathbf{W}}^*) \ge \min_{R_\mathrm{a}(\underline{\boldsymbol{\delta}}) \le \xi} y^* f(\mathbf{x}^* + \underline{\boldsymbol{\delta}}; \underline{\mathbf{W}}^*) \ge 1,$$

which means

$$1 \le f(\mathbf{x}^*) \le \|\mathbf{x}^*\|_\mathrm{F} \|\mathbf{w}^*\|_2 \prod_{l=1}^L \left\|\underline{\mathbf{W}}^{*(l)}\right\|_\mathrm{sp} \le \|\mathbf{w}^*\|_2 \prod_{l=1}^L \left\|\underline{\mathbf{W}}^{*(l)}\right\|_\mathrm{sp}$$
$$\le \left(\frac{1}{L+1} \left(\|\mathbf{w}^*\|_2 + \sum_{l=1}^L \left\|\underline{\mathbf{W}}^{*(l)}\right\|_\mathrm{sp}\right)\right)^{1/(L+1)}$$

indicating that

$$\frac{1}{L+1} \left(\|\mathbf{w}^*\|_2 + \sum_{l=1}^L \left\|\underline{\mathbf{W}}^{*(l)}\right\|_\mathrm{sp}\right) > 1.$$

On the other hand, according to Eq. (44) and Lemma 25, we have

$$\left\|\underline{\mathbf{W}}^{*(1)}\right\|_\mathrm{F} = \cdots = \left\|\underline{\mathbf{W}}^{*(L)}\right\|_\mathrm{F} = \|\mathbf{w}^*\|_2 \le \left(\frac{1}{(L+1)} B_v^{\frac{2(J+1)}{L+1}} (K + 1 + (\mathsf{c}m_J)(L-J))\right)^{1/2}.$$

Therefore, we obtain

$$\frac{1}{L+1} \left(\frac{\|\mathbf{w}^*\|_2}{\|\mathbf{w}^*\|_2} + \sum_{l=1}^L \frac{\left\|\underline{\mathbf{W}}^{*(l)}\right\|_\mathrm{sp}}{\left\|\underline{\mathbf{W}}^{*(l)}\right\|_\mathrm{F}}\right) \ge \left(\frac{1}{B_v}\right)^{\frac{J+1}{L+1}} \sqrt{\frac{L+1}{(J+1) + (\mathsf{c}m_J)(L-J)}}.$$

Note that

$$\left\|\mathbf{W}^{(l)}\right\|_{\mathrm{F}} = \left\|\widetilde{\mathbf{W}}_M^{*(l)}\right\|_{\mathrm{F}} \quad \text{and} \quad \left\|\mathbf{W}^{(l)}\right\|_{\mathrm{sp}} = \left\|\widetilde{\mathbf{w}}_M^{*(l)}\right\|,$$

where $\widetilde{\mathbf{W}}_M^{*(l)}$ denotes the $M$-block-diagonal matrix of weight tensor $\underline{\mathbf{W}}^{*(l)}$. Then, we have

$$\frac{1}{L} \sum_{l=1}^{L} \frac{\left\|\widetilde{\mathbf{W}}_M^{*(l)}\right\|}{\left\|\widetilde{\mathbf{W}}_M^{*(l)}\right\|_{\mathrm{F}}} \geq \left(1 + \frac{1}{L}\right) \left(\frac{1}{B_v}\right)^{\frac{J+1}{L+1}} \sqrt{\frac{L+1}{(J+1) + (\mathsf{c}m_J)(L-J)}} - \frac{1}{L}.$$

Taking the reciprocal of both sides gives

$$\frac{L}{\sum_{l=1}^{L} \left(r_{\mathrm{stb}}(\widetilde{\mathbf{W}}_M^{*(l)})\right)^{-1/2}} \leq \frac{1}{\left(1 + \frac{1}{L}\right) \left(\frac{1}{B_v}\right)^{\frac{J+1}{L+1}} \sqrt{\frac{L+1}{(J+1)+(\mathsf{c}m_J)(L-J)}} - \frac{1}{L}}.$$

**Lemma 25.** *For every $1 \leq i, j \leq L$, we have $\left\|\underline{\mathbf{W}}^{*(i)}\right\|_{\mathrm{F}} = \left\|\underline{\mathbf{W}}^{*(j)}\right\|_{\mathrm{F}} = \|\mathbf{w}^*\|_2$.*

*Proof.* Let $1 \leq i < j \leq L$. For $\mu > 0$, we define a t-NN $f_\mu(\underline{\mathbf{x}}; \underline{\mathbf{V}})$ with weights $\underline{\mathbf{V}} = (\underline{\mathbf{V}}^{(1)}, \cdots, \underline{\mathbf{V}}^L, \mathbf{v})$ which are constructed from $f(\underline{\mathbf{x}}; \underline{\mathbf{W}}^*)$ whose weights $\underline{\mathbf{W}}^* = (\underline{\mathbf{W}}^{*(1)}, \cdots, \underline{\mathbf{W}}^{*(L)}, \underline{\mathbf{w}}^*)$ is an optimal solution to Problem (14). Specifically, the construction of $\underline{\mathbf{V}} = (\underline{\mathbf{V}}^{(1)}, \cdots, \underline{\mathbf{V}}^L, \mathbf{v})$ is given as follows:

$$\forall l = 1, \cdots, L, \quad \underline{\mathbf{V}}^{(l)} = \begin{cases} \underline{\mathbf{W}}^{*(l)} & \text{if } l \neq i \text{ and } 1 \neq j \\ \mu \underline{\mathbf{W}}^{*(i)} & \text{if } l = i \\ \mu^{-1} \underline{\mathbf{W}}^{*(j)} & \text{if } 1 = j \end{cases}.$$

Note that for every input example $\underline{\mathbf{x}}$ and perturbation $\underline{\boldsymbol{\delta}}$, $f_\mu(\underline{\mathbf{x}} + \underline{\boldsymbol{\delta}}; \underline{\mathbf{V}}) = f(\underline{\mathbf{x}} + \underline{\boldsymbol{\delta}}; \underline{\mathbf{W}}^*)$, then $\underline{\mathbf{V}}$ is also feasible to Problem (14). Note that we have

$$\frac{\mathrm{d}}{\mathrm{d}\mu} \left(\left\|\mu \underline{\mathbf{W}}^{*(i)}\right\|_{\mathrm{F}}^2 + \left\|\mu^{-1} \underline{\mathbf{W}}^{*(j)}\right\|_{\mathrm{F}}^2\right) = 2\mu \left\|\underline{\mathbf{W}}^{*(i)}\right\|_{\mathrm{F}}^2 - 2\mu^{-3} \left\|\underline{\mathbf{W}}^{*(j)}\right\|_{\mathrm{F}}^2.$$

When $\mu = 1$ the above expression equals $2\left\|\underline{\mathbf{W}}^{*(i)}\right\|_{\mathrm{F}}^2 - 2\left\|\underline{\mathbf{W}}^{*(j)}\right\|_{\mathrm{F}}^2$. Hence, if $\left\|\underline{\mathbf{W}}^{*(i)}\right\|_{\mathrm{F}} \neq \left\|\underline{\mathbf{W}}^{*(j)}\right\|_{\mathrm{F}}$, then the derivative at $\mu$ is non-zero, which leads to a contradiction to the optimality of $\underline{\mathbf{W}}^*$ to Problem (14). Note that if we consider changing norms of $\underline{\mathbf{W}}^{*(i)}$ and $\mathbf{w}$ instead of $\underline{\mathbf{W}}^{*(i)}$ and $\underline{\mathbf{W}}^{*(j)}$, the same conclusion also holds. Thus, the optimality of $\underline{\mathbf{W}}^*$ strictly leads to $\left\|\underline{\mathbf{W}}^{*(i)}\right\|_{\mathrm{F}} = \left\|\underline{\mathbf{W}}^{*(j)}\right\|_{\mathrm{F}} = \|\mathbf{w}^*\|_2$. $\qquad\square$

## E   Generalization bound of approximately low-tubal-rank t-NNs

*Proof of Theorem 12.* Given a $(\delta, \mathbf{r})$-approximately low-tubal-rank parameterized t-NN $f(\underline{\mathbf{x}}; \underline{\mathbf{W}}) \in \mathfrak{F}_{\delta,\mathbf{r}} \subset \mathfrak{F}_{\delta,\mathbf{r}}$, let $g(\underline{\mathbf{x}}; \underline{\mathbf{W}}_\mathbf{r}) \in \mathfrak{F}_\mathbf{r}$ be its compressed version whose t-product layer weight tensors have tubal-ranks upper bounded by $\mathbf{r}$.

**Step 1: Upper bound the adversarial empirical $L_2$-distance between $f$ and $g$.** Consider function $g(\underline{\mathbf{x}}) = g(\underline{\mathbf{x}}; \underline{\mathbf{W}}_\mathbf{r})$ parameterized by $\underline{\mathbf{W}}_\mathbf{r} = (\underline{\mathbf{W}}_{r_1}^{(1)}, \cdots, \underline{\mathbf{W}}_{r_L}^{(L)}, \mathbf{w})$ as the function whose t-product layer weights are low-tubal-rank approximations of $f(\underline{\mathbf{x}}; \underline{\mathbf{W}})$. Let $\tilde{f}(\underline{\mathbf{x}}, y) = \inf_{R_a(\underline{\mathbf{x}}-\underline{\mathbf{x}}') \leq \xi} yf(\underline{\mathbf{x}}')$ and $\tilde{g}(\underline{\mathbf{x}}, y) = \inf_{R_a(\underline{\mathbf{x}}-\underline{\mathbf{x}}') \leq \xi} yg(\underline{\mathbf{x}}')$ denote the adversarial versions of $f$ and $g$, respectively.

We first bound the adversarial empirical $L_2$-distance between $f$ and $g$ as follows.

$$|\tilde{f}(\underline{\mathbf{x}}_i, y_i) - \tilde{g}(\underline{\mathbf{x}}_i, y_i)| = |\inf_{R_a(\underline{\mathbf{x}}_i-\underline{\mathbf{x}}_i') \leq \xi} y_i f(\underline{\mathbf{x}}_i') - \inf_{R_a(\underline{\mathbf{x}}_i-\underline{\mathbf{x}}_i') \leq \xi} y_i g(\underline{\mathbf{x}}_i')|.$$

Letting $\underline{\mathbf{x}}_i^f = \arginf_{R_a(\underline{\mathbf{x}}_i - \underline{\mathbf{x}}_i') \leq \xi} y_i f(\underline{\mathbf{x}}_i')$ and $\underline{\mathbf{x}}_i^g = \arginf_{R_a(\underline{\mathbf{x}}_i - \underline{\mathbf{x}}_i') \leq \xi} y_i g(\underline{\mathbf{x}}_i')$, we have $|\tilde{f}(\underline{\mathbf{x}}_i, y_i) - \tilde{g}(\underline{\mathbf{x}}_i, y_i)| = |y_i f(\underline{\mathbf{x}}_i^f) - y_i g(\underline{\mathbf{x}}_i^g)|$. By letting

$$\underline{\mathbf{x}}_i^\xi = \begin{cases} \underline{\mathbf{x}}_i^g & \text{if} \quad y_i f(\underline{\mathbf{x}}_i^f) \geq y_i g(\underline{\mathbf{x}}_i^g) \\ \underline{\mathbf{x}}_i^f & \text{otherwise} \end{cases},$$

we obtain

$$|\tilde{f}(\underline{\mathbf{x}}_i, y_i) - \tilde{g}(\underline{\mathbf{x}}_i, y_i)| = |y_i f(\underline{\mathbf{x}}_i^f) - y_i g(\underline{\mathbf{x}}_i^g)| \leq |y_i f(\underline{\mathbf{x}}_i^\xi) - y_i g(\underline{\mathbf{x}}_i^\xi)| = |f(\underline{\mathbf{x}}_i^\xi) - g(\underline{\mathbf{x}}_i^\xi)|.$$

Let $h_l(\underline{\mathbf{x}}^\xi) = \mathbf{w}^\top \mathrm{vec}\left( \sigma(\underline{\mathbf{W}}_{r_l}^{(L)} *_M \sigma(\underline{\mathbf{W}}_{r-1}^{(L-1)} *_M \cdots *_M \sigma(\underline{\mathbf{W}}_r^{(l+1)} *_M \sigma(\underline{\mathbf{W}}^{(l)} *_M \cdots *_M \sigma(\underline{\mathbf{W}}^{(1)} *_M \right.$

$\left. \underline{\mathbf{x}}^\xi) \cdots))) \right)$ and $h_0(\underline{\mathbf{x}}^\xi) = g(\underline{\mathbf{x}}^\xi)$. Then, we have

$$|f(\underline{\mathbf{x}}_i^\xi) - g(\underline{\mathbf{x}}_i^\xi)| \leq \sum_{l=1}^{L} |h_l(\underline{\mathbf{x}}_i^\xi) - h_{l-1}(\underline{\mathbf{x}}_i^\xi)|.$$

We can see that for any $l = 1, \cdots, L$:

$$\left\| \sigma(\underline{\mathbf{W}}^{(l-1)} *_M \cdots *_M \sigma(\underline{\mathbf{W}}^{(1)} *_M \underline{\mathbf{x}}_i^\xi) \cdots)) \right\|_F \leq \prod_{l'=1}^{l-1} \left\| \underline{\mathbf{W}}^{(l')} \right\|_F \left\| \underline{\mathbf{x}}_i^\xi \right\|_F \leq \prod_{l'=1}^{l-1} \left\| \underline{\mathbf{W}}^{(l')} \right\|_F B_{x, R_a, \xi},$$

and

$$\left\| \sigma(\underline{\mathbf{W}}_{r_l}^{(l)} *_M \sigma(\underline{\mathbf{W}}^{(l-1)} *_M \cdots *_M \sigma(\underline{\mathbf{W}}^{(1-1)} *_M \underline{\mathbf{x}}_i^\xi) \cdots)) - \sigma(\underline{\mathbf{W}}^{(l)} *_M \sigma(\underline{\mathbf{W}}^{(l-1)} *_M \cdots *_M \sigma(\underline{\mathbf{W}}^{(1)} *_M \underline{\mathbf{x}}_i^\xi) \cdots)) \right\|_F$$

$$\leq \left\| \underline{\mathbf{W}}_{r_l}^{(l)} *_M \sigma(\underline{\mathbf{W}}^{(l-1)} *_M \cdots *_M \sigma(\underline{\mathbf{W}}^{(1)} *_M \underline{\mathbf{x}}_i^\xi) \cdots) - \underline{\mathbf{W}}^{(l)} *_M \sigma(\underline{\mathbf{W}}^{(l-1)} *_M \cdots *_M \sigma(\underline{\mathbf{W}}^{(1)} *_M \underline{\mathbf{x}}_i^\xi) \cdots) \right\|_F$$

$$= \left\| (\underline{\mathbf{W}}_{r_l}^{(l)} - \underline{\mathbf{W}}^{(l)}) *_M \sigma(\underline{\mathbf{W}}^{(l-1)} *_M \cdots *_M \sigma(\underline{\mathbf{W}}^{(1)} *_M \underline{\mathbf{x}}_i^\xi) \cdots) \right\|_F$$

$$\leq \left\| \underline{\mathbf{W}}_{r_l}^{(l)} - \underline{\mathbf{W}}^{(l)} \right\|_F \left\| \sigma(\underline{\mathbf{W}}^{(l-1)} *_M \cdots *_M \sigma(\underline{\mathbf{W}}^{(1)} *_M \underline{\mathbf{x}}_i^\xi) \cdots) \right\|_F$$

$$\leq \delta_l \prod_{l'=1}^{l} \left\| \underline{\mathbf{W}}^{(l')} \right\|_F B_{x, R_a, \xi}.$$

Thus, we have

$$|h_l(\underline{\mathbf{x}}_i^\xi) - h_{l-1}(\underline{\mathbf{x}}_i^\xi)| \leq B_{\mathbf{w}} \prod_{j \neq l} B_j \delta B_{x, R_a, \xi} = \frac{\delta B_{\tilde{f}}}{B_l}.$$

This gives

$$|f(\underline{\mathbf{x}}_i^\xi) - g(\underline{\mathbf{x}}_i^\xi)| \leq \sum_{l=1}^{L} |h_l(\underline{\mathbf{x}}_i^\xi) - h_{l-1}(\underline{\mathbf{x}}_i^\xi)| \leq \sum_{l=1}^{L} \frac{\delta B_{\tilde{f}}}{B_l}.$$

Then, we can set

$$\hat{\mathfrak{r}} = \delta B_{\tilde{f}} \sum_{l=1}^{L} B_l^{-1}. \tag{46}$$

**Step 2: Divide and conquer the adversarial gap.** To upper bound the adversarial gap $\mathcal{L}^{\mathrm{adv}}(f) - \hat{\mathcal{L}}^{\mathrm{adv}}(f)$ of $f$ by using the properties of its compressed version $g$, we first decompose the adversarial gap into three terms as follows

$$\mathcal{L}^{\mathrm{adv}}(f) - \hat{\mathcal{L}}^{\mathrm{adv}}(f)$$
$$= \underbrace{\left[ (\mathcal{L}^{\mathrm{adv}}(f) - \mathcal{L}^{\mathrm{adv}}(g)) - (\hat{\mathcal{L}}^{\mathrm{adv}}(f) - \hat{\mathcal{L}}^{\mathrm{adv}}(g)) \right]}_{\mathbf{I}} + \underbrace{\left( \mathcal{L}^{\mathrm{adv}}(g) - \hat{\mathcal{L}}^{\mathrm{adv}}(g) \right)}_{\mathbf{II}}. \tag{47}$$

**Step 2.1**: *Upper bound* **II**. We first consider the event $\mathcal{E}_1$ in which term **II** is upper bounded with high probability. As $g \in \mathfrak{F}_{\mathbf{r}}$, the term **II** has already been upper bounded according to Theorem 6 as

$$\mathbf{II} \leq \frac{C'L_\ell B_{\tilde{f}}}{\sqrt{N}} \sqrt{\mathsf{c}\sum_{l=1}^{L} r_l(d_{l-1}+d_l)\log(9(L+1))} + 3B\sqrt{\frac{t}{2N}}, \tag{48}$$

with high probability $1 - 2e^{-t}$.

**Step 2.2**: *Upper bound* **I**. Note that term **I** can be written as

$$\mathcal{L}^{\text{adv}}(f) - \mathcal{L}^{\text{adv}}(g) - (\hat{\mathcal{L}}^{\text{adv}}(f) - \hat{\mathcal{L}}^{\text{adv}}(g))$$
$$= \frac{1}{N}\sum_{i=1}^{N}\left(\ell(\tilde{f}(\mathbf{x}),y) - \ell(\tilde{g}(\mathbf{x}),y) - \mathbb{E}[\ell(\tilde{f}(\mathbf{x}),y) - \ell(\tilde{g}(\mathbf{x}),y)]\right). \tag{49}$$

**Step (2.2.1)**: *Characterize the concentration behavior of* $\ell \circ \tilde{f} - \ell \circ \tilde{g}$. Given a constant $\mathfrak{r} > 0$, consider the event $\mathcal{E}_2(\mathfrak{r})$ in which $\left\|\tilde{f} - \tilde{g}\right\|_{L_2} \leq \mathfrak{r}$ already holds with high probability. Then, conditioned on Event $\mathcal{E}_2(\mathfrak{r})$, by using the $L_\ell$-Lipschitz continuity of the loss function $\ell(\cdot,\cdot)$ derived from Assumption 2, it can be proved that $\ell(\tilde{f}(\mathbf{x}),y) - \ell(\tilde{g}(\mathbf{x}),y)$ also has a small population $L_2$-norm with high probability.

Regarding Eq. (49) , it is natural to characterize the concentration behavior of centered random variable $\ell(\tilde{f}(\mathbf{x}),y) - \ell(\tilde{g}(\mathbf{x}),y) - \mathbb{E}[\ell(\tilde{f}(\mathbf{x}),y) - \ell(\tilde{g}(\mathbf{x}),y)]$.

- First, its variance under event $\mathcal{E}_2$ can be upper bounded by

$$\sup_{\|\tilde{f}-\tilde{g}\|_{L_2}\leq\mathfrak{r}} \texttt{Var}\left(\ell(\tilde{f}(\mathbf{x}),y) - \ell(\tilde{g}(\mathbf{x}),y) - \mathbb{E}[\ell(\tilde{f}(\mathbf{x}),y) - \ell(\tilde{g}(\mathbf{x}),y)]\right)$$

$$= \sup_{\|\tilde{f}-\tilde{g}\|_{L_2}\leq\mathfrak{r}} \texttt{Var}\left(\ell(\tilde{f}(\mathbf{x}),y) - \ell(\tilde{g}(\mathbf{x}),y)\right)$$

$$= \sup_{\|\tilde{f}-\tilde{g}\|_{L_2}\leq\mathfrak{r}} \mathbb{E}_{(\mathbf{x},y)}\left[(\ell(\tilde{f}(\mathbf{x},y)) - \ell(\tilde{g}(\mathbf{x},y)))^2 - \mathbb{E}_{(\mathbf{x},y)}[\ell(\tilde{f}(\mathbf{x},y)) - \ell(\tilde{g}(\mathbf{x},y))]\right]$$

$$\leq \sup_{\|\tilde{f}-\tilde{g}\|_{L_2}\leq\mathfrak{r}} \mathbb{E}_{(\mathbf{x},y)}\left[\ell(\tilde{f}(\mathbf{x},y)) - \ell(\tilde{g}(\mathbf{x},y))\right]^2$$

$$\leq \sup_{\|\tilde{f}-\tilde{g}\|_{L_2}\leq\mathfrak{r}} L_\ell^2 \mathbb{E}_{(\mathbf{x},y)}\left[\tilde{f}(\mathbf{x},y) - \tilde{g}(\mathbf{x},y)\right]^2$$

$$= \sup_{\|\tilde{f}-\tilde{g}\|_{L_2}\leq\mathfrak{r}} L_\ell^2 \left\|\tilde{f} - \tilde{g}\right\|_{L_2}^2$$

$$\leq L_\ell^2\mathfrak{r}^2$$

- Second, we upper bound its $L_\infty$-norm. First, Lemma 39 indicates that for any $h \in \mathfrak{F}$ with adversarial version $\tilde{h} \in \mathfrak{F}^{\text{adv}}$, we have $\left\|\tilde{h}\right\|_{L_\infty} \leq B_{\tilde{f}} := B_{\underline{\mathbf{W}}}B_{x,R_{\mathrm{a}},\xi}$. Then, by $\mathfrak{F}_{\mathbf{r}} \subset \mathfrak{F}_{\delta,\mathbf{r}} \subset \mathfrak{F}$, we have $\left\|\tilde{f}\right\|_{L_\infty} \leq B_{\tilde{f}}$ and $\|\tilde{g}\|_{L_\infty} \leq B_{\tilde{f}}$. Therefore, we can upper bound

the $L_\infty$-norm of $\ell(\tilde{f}(\mathbf{x}), y) - \ell(\tilde{g}(\mathbf{x}), y) - \mathbb{E}[\ell(\tilde{f}(\mathbf{x}), y) - \ell(\tilde{g}(\mathbf{x}), y)]$ as follows

$$\sup_{\|\tilde{f}-\tilde{g}\|_{L_2} \leq \mathfrak{r}} \left\| \ell(\tilde{f}(\mathbf{x}), y) - \ell(\tilde{g}(\mathbf{x}), y) - \mathbb{E}[\ell(\tilde{f}(\mathbf{x}), y) - \ell(\tilde{g}(\mathbf{x}), y)] \right\|_{L_\infty}$$

$$\leq \sup_{\|\tilde{f}-\tilde{g}\|_{L_2} \leq \mathfrak{r}} \left\| \ell(\tilde{f}(\mathbf{x}), y) - \ell(\tilde{g}(\mathbf{x}), y) \right\|_{L_\infty} + \left\| \mathbb{E}[\ell(\tilde{f}(\mathbf{x}), y) - \ell(\tilde{g}(\mathbf{x}), y)] \right\|_{L_\infty}$$

$$\leq \sup_{\|\tilde{f}-\tilde{g}\|_{L_2} \leq \mathfrak{r}} \left\| \ell(\tilde{f}(\mathbf{x}), y) - \ell(\tilde{g}(\mathbf{x}), y) \right\|_{L_\infty} + \mathbb{E}\left[ \left\| \ell(\tilde{f}(\mathbf{x}), y) - \ell(\tilde{g}(\mathbf{x}), y) \right\|_{L_\infty} \right]$$

$$\leq \sup_{\|\tilde{f}-\tilde{g}\|_{L_2} \leq \mathfrak{r}} L_\ell \sup_{(\mathbf{x}, y)} |y\tilde{f}(\mathbf{x})) - y\tilde{g}(\mathbf{x})| + \mathbb{E}[L_\ell \sup_{(\mathbf{x}, y)} |y\tilde{f}(\mathbf{x}) - y\tilde{g}(\mathbf{x})|]$$

$$\leq \sup_{\|\tilde{f}-\tilde{g}\|_{L_2} \leq \mathfrak{r}} 4 L_\ell B_{\tilde{f}}.$$

Then, the Talagrand's concentration inequality (Lemma 35) yields that with probability at least $1 - e^{-t}$:

$$\sup_{\|\tilde{f}-\tilde{g}\|_{L_2} \leq \mathfrak{r}} (\mathcal{L}^{\mathrm{adv}}(f) - \mathcal{L}^{\mathrm{adv}}(g)) - (\hat{\mathcal{L}}^{\mathrm{adv}}(f) - \hat{\mathcal{L}}^{\mathrm{adv}}(g))$$

$$\leq 2 \underbrace{\mathbb{E}\left[ \sup_{\|\tilde{f}-\tilde{g}\|_{L_2} \leq \mathfrak{r}} (\mathcal{L}^{\mathrm{adv}}(f) - \mathcal{L}^{\mathrm{adv}}(g)) - (\hat{\mathcal{L}}^{\mathrm{adv}}(f) - \hat{\mathcal{L}}^{\mathrm{adv}}(g)) \right]}_{(\mathbf{I})} + \frac{\sqrt{2t}L_\ell \mathfrak{r}}{\sqrt{N}} + \frac{8t L_\ell B_{\tilde{f}}}{N}.$$

Then, by the the standard symmetrization argument [47], we obtain an upper bound on term $(\mathbf{I})$ as follows:

$$\mathbb{E}\left[ \sup_{\|\tilde{f}-\tilde{g}\|_{L_2} \leq \mathfrak{r}} (\mathcal{L}^{\mathrm{adv}}(f) - \mathcal{L}^{\mathrm{adv}}(g)) - (\hat{\mathcal{L}}^{\mathrm{adv}}(f) - \hat{\mathcal{L}}^{\mathrm{adv}}(g)) \right]$$

$$\leq 2 \mathbb{E}_{(\mathbf{x}_i, y_i)_{i=1}^N} \mathbb{E}_{(\varepsilon_i)_{i=1}^N} \sup_{\|\tilde{f}-\tilde{g}\|_{L_2} \leq \mathfrak{r}} \left[ \frac{1}{N} \varepsilon_i \left( \ell(\tilde{f}(\mathbf{x}_i, y_i)) - \ell(\tilde{g}(\mathbf{x}_i, y_i)) \right) \right] \qquad (50)$$

$$= 2\Phi(\mathfrak{r}),$$

where $\Phi(\mathfrak{r})$ is defined as

$$\Phi(\mathfrak{r}) := \bar{R}_N \left( \{\ell \circ \tilde{f} - \ell \circ \tilde{g} \mid \tilde{f} \in \mathfrak{F}_{\delta, \mathbf{r}}^{\mathrm{adv}}, \tilde{g} \in \mathfrak{F}_{\mathbf{r}}^{\mathrm{adv}}, \left\| \tilde{f} - \tilde{g} \right\|_{L_2} \leq \mathfrak{r} \} \right).$$

Thus, there is a constant $C > 0$ such that for any $\tilde{f} - \tilde{g} \in (\mathfrak{F}_{\delta, \mathbf{r}}^{\mathrm{adv}} - \mathfrak{F}_{\mathbf{r}}^{\mathrm{adv}})$ satisfying $\left\| \tilde{f} - \tilde{g} \right\|_{L_2} \leq \mathfrak{r}$, it holds with probability at least $1 - e^{-t}$ that

$$(\mathcal{L}^{\mathrm{adv}}(f) - \mathcal{L}^{\mathrm{adv}}(g)) - (\hat{\mathcal{L}}^{\mathrm{adv}}(f) - \hat{\mathcal{L}}^{\mathrm{adv}}(g)) \leq C \left( \Phi(\mathfrak{r}) + L_\ell \mathfrak{r} \sqrt{\frac{t}{N}} + \frac{t L_\ell B_{\tilde{f}}}{N} \right). \qquad (51)$$

We denote the above event by $\mathcal{E}_3(\mathfrak{r})$. Note that Event $\mathcal{E}_3(\mathfrak{r})$ is conditioned on Event $\mathcal{E}_2(\mathfrak{r})$.

***Step (2.2.2):*** *Upper bound the probability of Event* $\mathcal{E}_2(\mathfrak{r}) := \left\{ \left\| \tilde{f} - \tilde{g} \right\|_{L_2} \leq \mathfrak{r} \right\}$. We further bound the probability of Event $\mathcal{E}_2(\mathfrak{r})$ in which $\left\| \tilde{f} - \tilde{g} \right\|_{L_2} \leq \mathfrak{r}$ holds. Generally speaking, $\tilde{f}$ and $\tilde{g}$ are date dependent and we can only bound the empirical $L_2$-distance between them. However, the local Rademacher complexity is characterized by the population $L_2$-norm. Thus, we need to bound the population $L_2$-distance between $\tilde{f}$ and $\tilde{g}$. Motivated by [43], we use the ratio type empirical process to bound the bound the population $L_2$-distance.

According to Assumption 11, their exists a function $\phi : [0, \infty) \to [0, \infty)$ such that

$$\dot{R}_{\mathfrak{r}}(\mathfrak{F}_{\delta,\mathbf{r}}^{\mathrm{adv}} - \mathfrak{F}_{\mathbf{r}}^{\mathrm{adv}}) \leq \phi(\mathfrak{r}) \text{ and } \phi(2\mathfrak{r}) \leq 2\phi(\mathfrak{r}), \ (\forall \mathfrak{r} > 0).$$

Define the quantity $\Gamma(\mathfrak{r}) := \mathbb{E}\left[\sup_h \left(\frac{1}{N}\sum_{i=1}^{N}\varepsilon_i h^2(\mathbf{x}_i, y_i) \mid h \in (\mathfrak{F}_{\delta,\mathbf{r}}^{\mathrm{adv}} - \mathfrak{F}_{\mathbf{r}}^{\mathrm{adv}}) : \|h\|_{L_2} \leq \mathfrak{r}\right)\right]$.
Then, we have

$$\Gamma(\mathfrak{r}) = \mathbb{E}\left[\sup_h \left(\frac{1}{N}\sum_{i=1}^{N}\varepsilon_i h^2(\mathbf{x}_i, y_i) \mid h \in (\mathfrak{F}_{\delta,\mathbf{r}}^{\mathrm{adv}} - \mathfrak{F}_{\mathbf{r}}^{\mathrm{adv}}) : \|h\|_{L_2} \leq \mathfrak{r}\right)\right]$$

$$\overset{(i)}{\leq} 2B_{\tilde{f}}\mathbb{E}\left[\sup_h \left(\frac{1}{N}\sum_{i=1}^{N}\varepsilon_i h(\mathbf{x}_i, y_i) \mid h \in (\mathfrak{F}_{\delta,\mathbf{r}}^{\mathrm{adv}} - \mathfrak{F}_{\mathbf{r}}^{\mathrm{adv}}) : \|h\|_{L_2} \leq \mathfrak{r}\right)\right]$$

$$\leq 2B_{\tilde{f}}\dot{R}_{\mathfrak{r}}(\mathfrak{F}_{\delta,\mathbf{r}}^{\mathrm{adv}} - \mathfrak{F}_{\mathbf{r}}^{\mathrm{adv}}) \overset{(ii)}{\leq} 2B_{\tilde{f}}\phi(\mathfrak{r}),$$

where $(i)$ is by the Talagrand's contraction lemma (Lemma 30).

We can verify that the square $h^2(\cdot)$ of any function $h \in \mathfrak{F}_{\delta,\mathbf{r}}^{\mathrm{adv}} - \mathfrak{F}_{\mathbf{r}}^{\mathrm{adv}}$ satisfies

(i) its $L_\infty$-norm is upper bouned by $B_{\tilde{f}}^2$, i.e., $\left\|h^2\right\|_{L_\infty} = \sup_{(\mathbf{x},y)}|h^2(\mathbf{x}, y)| \leq B_{\tilde{f}}^2$.

(ii) its second-order moment satisfies $\mathbb{E}_{\mathbf{x},y}\left[(h^2(\mathbf{x}, y)^2)\right] \leq \mathbb{E}_{\mathbf{x},y}\left[B_{\tilde{f}}^2(h^2(\mathbf{x}, y))\right] = B_{\tilde{f}}^2\mathbb{E}_{\mathbf{x},y}\left[h^2(\mathbf{x}, y)\right]$.

Thus, $h^2$ satisfy the conditions in Eq. (7.6) and Eq. (7.7) of [42] with parameters $B = B_{\tilde{f}}^2, V = B_{\tilde{f}}^2$ and $\vartheta = 1$. Noting that we have upper bounded $\Gamma(\mathfrak{r})$ by $2B_{\tilde{f}}\phi(\mathfrak{r})$, then by the peeling trick [42, Eq. (7.17)], we can show for any $\mathfrak{r} > \inf\{\sqrt{\mathbb{E}[h^2]} : h \in (\mathfrak{F}_{\delta,\mathbf{r}}^{\mathrm{adv}} - \mathfrak{F}_{\mathbf{r}}^{\mathrm{adv}}) : \|h\|_{L_2} \leq \mathfrak{r}\}$ and $t > 0$ that

$$\mathbb{P}\left[\sup_{h \in \mathfrak{F}_{\delta,\mathbf{r}}^{\mathrm{adv}} - \mathfrak{F}_{\mathbf{r}}^{\mathrm{adv}}} \frac{\|h\|_{L_2}^2 - \|h\|_S^2}{\|h\|_{L_2}^2 + \mathfrak{r}^2} \geq 8\frac{2B_{\tilde{f}}\phi(\mathfrak{r})}{\mathfrak{r}^2} + B_{\tilde{f}}\sqrt{\frac{2t}{\mathfrak{r}^2 N}} + B_{\tilde{f}}^2\frac{2t}{\mathfrak{r}^2 N}\right] \leq e^{-t}.$$

We further define a function $\mathfrak{r}_* = \mathfrak{r}_*(t)$ as

$$\mathfrak{r}_*(t) := \inf\left\{\mathfrak{r} > 0 \ \middle| \ \frac{16B_{\tilde{f}}\phi(\mathfrak{r})}{\mathfrak{r}^2} + B_{\tilde{f}}\sqrt{\frac{2t}{\mathfrak{r}^2 N}} + B_{\tilde{f}}^2\frac{2t}{\mathfrak{r}^2 N} \leq \frac{1}{2}\right\}, \tag{52}$$

which is useful to bound the ratio of the empirical $L_2$-norm and the population $L_2$-norm of an elements $h \in \mathfrak{F}_{\delta,\mathbf{r}}^{\mathrm{adv}} - \mathfrak{F}_{\mathbf{r}}^{\mathrm{adv}}$ with probability at least $1 - e^t$:

$$\frac{\|h\|_{L_2}^2 - \|h\|_S^2}{\|h\|_{L_2}^2 + \mathfrak{r}_*^2} \leq \frac{1}{2} \quad \Rightarrow \quad \|h\|_{L_2}^2 \leq 2(\|h\|_S^2 + \mathfrak{r}_*^2).$$

Recalling that $\left\|\tilde{f} - \tilde{g}\right\|_S \leq \hat{\mathfrak{r}}$, we obtain that the probability of Event $\mathcal{E}_2(\dot{\mathfrak{r}})$ with $\dot{\mathfrak{r}} = \sqrt{2(\hat{\mathfrak{r}}^2 + \mathfrak{r}_*^2(t))}$ is at least $1 - e^{-t}$.

**Step 2.3**: *Combining Events $\mathcal{E}_1, \mathcal{E}_2(\dot{\mathfrak{r}}), \mathcal{E}_3(\dot{\mathfrak{r}})$.* By combining Eqs. (46), (47), (51) and (48) along with their underlying events $\mathcal{E}_1, \mathcal{E}_2(\dot{\mathfrak{r}}), \mathcal{E}_3(\dot{\mathfrak{r}})$, we obtain

$$\mathcal{L}^{\mathrm{adv}}(f) - \hat{\mathcal{L}}^{\mathrm{adv}}(f) \leq \frac{C_1 L_\ell B_{\tilde{f}}}{\sqrt{N}}\sqrt{\mathsf{c}\sum_{l=1}^{L}r_l(d_{l-1} + d_l)\log(9(L+1)) + B\sqrt{\frac{t}{2N}}}$$

$$+ C_2\left(\Phi(\dot{\mathfrak{r}}) + L_\ell\dot{\mathfrak{r}}\sqrt{\frac{t}{N}} + \frac{tL_\ell B_{\tilde{f}}}{N}\right),$$

with probability at least $1 - 4e^{-t}$. $\qquad\square$

## E.1 Several Useful Results

According to Theorem 12, it remains to upper bound $\Phi(\mathfrak{r})$. In this subsection, we derive upper bounds on $\Phi(\mathfrak{r})$ in terms of covering numbers of the considered function sets $\mathfrak{F}_{\mathbf{r}}^{\mathrm{adv}}$ and $\mathfrak{F}_{\delta,\mathbf{r}}^{\mathrm{adv}}$.

Consider the supremum of the empirical $L_2$-norm of any function $(\tilde{f} - \tilde{g}) \in (\mathfrak{F}_{\delta,\mathbf{r}}^{\mathrm{adv}} - \mathfrak{F}_{\mathbf{r}}^{\mathrm{adv}})$ on sample $S = \{(\mathbf{x}_i, y_i)\}_{i=1}^{N}$ when the population $L_2$-norm is bounded by a given radius $\mathfrak{r} > 0$ as follows

$$\beta_S = \beta_S(\mathfrak{r}) = \sup \left\{ \left\| \tilde{f} - \tilde{g} \right\|_S \; \Big| \; \left\| \tilde{f} - \tilde{g} \right\|_{L_2} \leq \mathfrak{r}, \tilde{f} \in \mathfrak{F}_{\delta,\mathbf{r}}^{\mathrm{adv}}, \tilde{g} \in \mathfrak{F}_{\mathbf{r}}^{\mathrm{adv}} \right\}. \tag{53}$$

Recall that we have assumed $\dot{R}_{\mathfrak{r}}(\mathfrak{F}_{\delta,\mathbf{r}}^{\mathrm{adv}} - \mathfrak{F}_{\mathbf{r}}^{\mathrm{adv}}) \leq \phi(\mathfrak{r})$. We now give an explict example of $\phi(\mathfrak{r})$ in terms of the sum of covering entropy of $\mathfrak{F}_{\delta,\mathbf{r}}^{\mathrm{adv}}$ and $\mathfrak{F}_{\mathbf{r}}^{\mathrm{adv}}$

$$\dot{R}_{\mathfrak{r}}(\mathfrak{F}_{\delta,\mathbf{r}}^{\mathrm{adv}} - \mathfrak{F}_{\mathbf{r}}^{\mathrm{adv}})$$

$$\stackrel{(i)}{\leq} \mathbb{E}_S \hat{R}_{\mathfrak{r}} \left( \left\{ (\tilde{f} - \tilde{g}) \in (\mathfrak{F}_{\delta,\mathbf{r}}^{\mathrm{adv}} - \mathfrak{F}_{\mathbf{r}}^{\mathrm{adv}}), \left\| \tilde{f} - \tilde{g} \right\|_{L_2} \leq \mathfrak{r} \right\} \right)$$

$$\stackrel{(ii)}{\leq} \inf_{a} \left[ a + \mathbb{E}_S \int_a^{\beta_S} \sqrt{ \frac{ \log \mathsf{N} \left( \left\{ (\tilde{f} - \tilde{g}) \in (\mathfrak{F}_{\delta,\mathbf{r}}^{\mathrm{adv}} - \mathfrak{F}_{\mathbf{r}}^{\mathrm{adv}}), \left\| \tilde{f} - \tilde{g} \right\|_{L_2} \leq \mathfrak{r} \right\}, \left\| \cdot \right\|_S, \epsilon \right) }{ N } } \, d\epsilon \right]$$

$$\stackrel{(iii)}{\leq} \frac{1}{N} + \mathbb{E}_S \int_{1/N}^{\beta_S} \sqrt{ \frac{ \log \mathsf{N} \left( \left\{ (\tilde{f} - \tilde{g}) \in (\mathfrak{F}_{\delta,\mathbf{r}}^{\mathrm{adv}} - \mathfrak{F}_{\mathbf{r}}^{\mathrm{adv}}), \left\| \tilde{f} - \tilde{g} \right\|_{L_2} \leq \mathfrak{r} \right\}, \left\| \cdot \right\|_S, \epsilon \right) }{ N } } \, d\epsilon$$

$$\stackrel{(iv)}{\leq} \frac{1}{N} + \mathbb{E}_S \int_{1/N}^{\beta_S} \sqrt{ \frac{ \log \mathsf{N}(\mathfrak{F}_{\delta,\mathbf{r}}^{\mathrm{adv}}, \left\| \cdot \right\|_S, \epsilon/2) + \log \mathsf{N}(\mathfrak{F}_{\mathbf{r}}^{\mathrm{adv}}, \left\| \cdot \right\|_S, \epsilon/2) }{ N } } \, d\epsilon$$

$$=: \phi(\mathfrak{r}),$$

$$\tag{54}$$

where $(i)$ is the definition of localized Rademacher complexity; $(ii)$ is due to Dudley's inequality (Lemma 31); $(iii)$ is obtained by letting $a = 1/N$; $(iv)$ holds by Lemma 34.

**Lemma 26.** *We can upper bound $\Phi(\mathfrak{r})$ by using $\phi(\mathfrak{r})$ as follows*

$$\Phi(\mathfrak{r}) \leq C L_\ell \phi(\mathfrak{r}). \tag{55}$$

*Proof.* Recall that $\Phi(\mathfrak{r})$ is the average Rademacher complexity of the function set

$$\left\{ \ell \circ \tilde{f} - \ell \circ \tilde{g} \; \Big| \; \left\| \tilde{f} - \tilde{g} \right\|_{L_2} \leq \mathfrak{r}, \; \tilde{f} \in \mathfrak{F}_{\delta,\mathbf{r}}^{\mathrm{adv}}, \tilde{g} \in \mathfrak{F}_{\mathbf{r}}^{\mathrm{adv}} \right\}.$$

As $\ell(\cdot)$ is $L_\ell$-Lipschitz, we have for any functions $\tilde{f} \in \mathfrak{F}_{\delta,\mathbf{r}}^{\mathrm{adv}}, \tilde{g} \in \mathfrak{F}_{\mathbf{r}}^{\mathrm{adv}}$ satisfying $\left\| \tilde{f} - \tilde{g} \right\|_{L_2} \leq \mathfrak{r}$

$$\sup_{\tilde{f},\tilde{g}} \left\| \ell \circ \tilde{f} - \ell \circ \tilde{g} \right\|_S = \sup_{\tilde{f},\tilde{g}} \sqrt{ \sum_{i=1}^{N} \frac{1}{N} \left( \ell(\tilde{f}(\mathbf{x}_i, y_i)) - \ell(\tilde{g}(\mathbf{x}_i, y_i)) \right)^2 }$$

$$\stackrel{(i)}{\leq} \sup_{\tilde{f},\tilde{g}} \sqrt{ \sum_{i=1}^{N} \frac{1}{N} \left( L_\ell (\tilde{f}(\mathbf{x}_i, y_i) - \tilde{g}(\mathbf{x}_i, y_i)) \right)^2 }$$

$$= \sup_{\tilde{f},\tilde{g}} L_\ell \sqrt{ \sum_{i=1}^{N} \frac{1}{N} \left( \tilde{f}(\mathbf{x}_i, y_i) - \tilde{g}(\mathbf{x}_i, y_i) \right)^2 }$$

$$= L_\ell \sup_{\tilde{f},\tilde{g}} \left\| \tilde{f} - \tilde{g} \right\|_S$$

$$= L_\ell \beta_S,$$

where $(i)$ holds because the loss function $\ell$ is $L_\ell$-Lischitz continous.

To bound $\Phi(\mathfrak{r})$, we first bound the its empirical version using the Dudley's inequlity (Lemma 31) up to a constant as follows

$$\inf_{a>0}\left[a+\int_a^{L_\ell\beta_S}\sqrt{\frac{\log\mathsf{N}\left(\{\ell\circ\tilde{f}-\ell\circ\tilde{g}\mid\tilde{f}\in\mathfrak{F}_{\delta,\mathbf{r}}^{\mathrm{adv}},\tilde{g}\in\mathfrak{F}_{\mathbf{r}}^{\mathrm{adv}},\left\|\tilde{f}-\tilde{g}\right\|_{L_2}\le\mathfrak{r}\}\right),\|\cdot\|_S,\epsilon)}{N}}\,\mathrm{d}\epsilon\right]$$

$$\overset{(i)}{\le}\frac{L_\ell}{N}+\int_{L_\ell/N}^{L_\ell\beta_S}\sqrt{\frac{\log\mathsf{N}\left(\{\ell\circ\tilde{f}-\ell\circ\tilde{g}\mid\tilde{f}\in\mathfrak{F}_{\delta,\mathbf{r}}^{\mathrm{adv}},\tilde{g}\in\mathfrak{F}_{\mathbf{r}}^{\mathrm{adv}},\left\|\tilde{f}-\tilde{g}\right\|_{L_2}\le\mathfrak{r}\}\right),\|\cdot\|_S,\epsilon)}{N}}\,\mathrm{d}\epsilon$$

$$\overset{(ii)}{\le}\frac{L_\ell}{N}+\int_{L_\ell/N}^{L_\ell\beta_S}\sqrt{\frac{\log\mathsf{N}\left(\{\tilde{f}-\tilde{g}\mid\tilde{f}\in\mathfrak{F}_{\delta,\mathbf{r}}^{\mathrm{adv}},\tilde{g}\in\mathfrak{F}_{\mathbf{r}}^{\mathrm{adv}},\left\|\tilde{f}-\tilde{g}\right\|_{L_2}\le\mathfrak{r}\}\right),\|\cdot\|_S,\epsilon/L_\ell)}{N}}\,\mathrm{d}\epsilon$$

$$\overset{(iii)}{\le}\frac{L_\ell}{N}+\int_{1/N}^{\beta_S}\sqrt{\frac{\log\mathsf{N}\left(\{\tilde{f}-\tilde{g}\mid\tilde{f}\in\mathfrak{F}_{\delta,\mathbf{r}}^{\mathrm{adv}},\tilde{g}\in\mathfrak{F}_{\mathbf{r}}^{\mathrm{adv}},\left\|\tilde{f}-\tilde{g}\right\|_{L_2}\le\mathfrak{r}\}\right),\|\cdot\|_S,t)}{N}}\,L_\ell\mathrm{d}t$$

$$=L_\ell\phi(\mathfrak{r}),$$

$$(56)$$

where in $(i)$ we let $a=L_\ell/N$; $(ii)$ holds by the Lipschitzness of $\ell$ and the definition of covering number; we use change of variable $t=\epsilon/L_\ell$ in $(iii)$.

By taking expectations on the RHS of Eq. (56) with respect to the sample $S$, we obtain Eq. (55).

$\square$

To determine an appropriate radius of the population $L_2$-norm $\dot{\mathfrak{r}}=2\sqrt{\hat{\mathfrak{r}}^2+\mathfrak{r}_*^2}$, we need to compute the value of $\mathfrak{r}_*$ of satisfying Eq. (52). Using Eq. (54), we show how to compute $\phi(\mathfrak{r})$ when the covering numbers of $\mathfrak{F}_{\delta,\mathbf{r}}^{\mathrm{adv}}$ and $\mathfrak{F}_{\mathbf{r}}^{\mathrm{adv}}$ satisfy a special bound.

**Lemma 27** (Adapted from Lemma 3 in Ref. [43]). *Suppose that the covering numbers of $\mathfrak{F}_{\delta,\mathbf{r}}^{\mathrm{adv}}$ and $\mathfrak{F}_{\mathbf{r}}^{\mathrm{adv}}$ satisfy*

$$\sup_S\log\mathsf{N}(\mathfrak{F}_{\delta,\mathbf{r}}^{\mathrm{adv}},\|\cdot\|_S,\epsilon/2)+\sup_S\log\mathsf{N}(\mathfrak{F}_{\mathbf{r}}^{\mathrm{adv}},\|\cdot\|_S,\epsilon/2)\le a_1+a_2\log(\epsilon^{-1})+a_3\epsilon^{-2q}\quad(57)$$

*for some $q\le 1$. Then, it holds that*

(I) *The bound $\phi(\mathfrak{r})$ of the local Rademacher complexity $\dot{R}_{\mathfrak{r}}(\mathfrak{F}_{\delta,\mathbf{r}}^{\mathrm{adv}}-\mathfrak{F}_{\mathbf{r}}^{\mathrm{adv}})$ of radius $\mathfrak{r}$ can be upper bounded as*

$$\phi(\mathfrak{r})\le C\max\left\{\frac{1}{N}+B_{\tilde{f}}\frac{a_1+a_2\log N}{N}+\mathfrak{r}\sqrt{\frac{a_1+a_2\log N}{N}},\right.$$
$$\left.C_q\left[\frac{1}{N}+(\frac{a_3B_{\tilde{f}}^{1-q}}{N})^{\frac{1}{1+q}}+\mathfrak{r}^{1-q}\sqrt{\frac{a_3}{N}}\right]\right\},$$

$$(58)$$

*for a universal constant $C>0$ and a constant $C_q>0$ which only depends on $q\le 1$.*

(II) *In particular, the quantity $\mathfrak{r}_*(t)$ satisying Eq. (52) can be upper bounded as*

$$\mathfrak{r}_*^2(t)\le C\left[B_{\tilde{f}}\frac{a_1+a_2\log N}{N}+(\frac{a_3}{N})^{\frac{1}{1+q}}\left(B_{\tilde{f}}^{\frac{1-q}{1+q}}+1\right)+\frac{1+tB_{\tilde{f}}}{N}\right].$$

*Proof of Lemma 27.* According to Eq. (54) which gives $\phi(\mathfrak{r})$ and the definition of $\beta_S$ in Eq. (53), we need to upper bound

$$\mathbb{E}_S \int_{1/N}^{\beta_S} \sqrt{\frac{\log \mathsf{N}(\mathfrak{F}_{\delta,\mathbf{r}}^{\text{adv}}, \|\cdot\|_S, \epsilon/2) + \log \mathsf{N}(\mathfrak{F}_{\mathbf{r}}^{\text{adv}}, \|\cdot\|_S, \epsilon/2)}{N}} \, d\epsilon$$

$$\leq \mathbb{E}_S \int_{1/N}^{\beta_S} \sqrt{\frac{a_1 + a_2 \log(\epsilon^{-1}) + a_3 \epsilon^{-2q}}{N}} \, d\epsilon$$

$$\leq \mathbb{E}_S \int_{1/N}^{\beta_S} \sqrt{\frac{a_1 + a_2 \log(\epsilon^{-1})}{N}} \, d\epsilon + \mathbb{E}_S \int_{1/N}^{\beta_S} \sqrt{\frac{a_3 \epsilon^{-2q}}{N}} \, d\epsilon$$

$$\leq \sqrt{a_1 + a_2 \log N} \sqrt{\mathbb{E}_S \beta_S} + \frac{\sqrt{a_3}}{1-q} \mathbb{E}_S \beta_S^{1-q}$$

$$\leq \underbrace{\sqrt{a_1 + a_2 \log N} \sqrt{2 B_{\tilde{f}} \phi(\mathfrak{r}) + \mathfrak{r}^2}}_{\mathbf{I}} + \underbrace{\frac{\sqrt{a_3}}{1-q} \left( 2 B_{\tilde{f}} \phi(\mathfrak{r}) + \mathfrak{r}^2 \right)^{\frac{1-q}{2}}}_{\mathbf{II}}.$$

Hence, if $\mathbf{I} \geq \mathbf{II}$, then

$$\phi(\mathfrak{r}) \leq C \left( \frac{1}{N} + \sqrt{\frac{a_1 + a_2 \log N}{N}} + \sqrt{2 B_{\tilde{f}} \phi(\mathfrak{r}) + \mathfrak{r}^2} \right)$$

$$\leq \frac{C}{N} + C^2 B_{\tilde{f}} \frac{a_1 + a_2 \log N}{N} + C\mathfrak{r} \sqrt{\frac{a_1 + a_2 \log N}{N}} + \frac{\phi(\mathfrak{r})}{2},$$

which leads to

$$\phi(\mathfrak{r}) \leq \frac{2C}{N} + 2C^2 B_{\tilde{f}} \frac{a_1 + a_2 \log N}{N} + 2C\mathfrak{r} \sqrt{\frac{a_1 + a_2 \log N}{N}}.$$

If $\mathbf{I} < \mathbf{II}$, then by using Young's inequality we obtain

$$\phi(\mathfrak{r}) \leq C \left( \frac{1}{N} + \frac{\sqrt{a_3}}{1-q} \left( 2 B_{\tilde{f}} \phi(\mathfrak{r}) + \mathfrak{r}^2 \right)^{\frac{1-q}{2}} \right)$$

$$\leq \frac{C}{N} + C \left( q \left( \frac{c_1^{1-q} C^2 a_3}{N(1-q)^2} \right)^{\frac{1}{1+q}} + (1-q) \frac{2 B_{\tilde{f}} \phi(\mathfrak{r})}{c_1} + \sqrt{\frac{a_3}{N(1-q)}} \mathfrak{r}^{2(1-q)} \right),$$

for any $c_1 > 0$. Thus, by taking $c_1 = 4C(1-q)B_{\tilde{f}}$, we obtain

$$\phi(\mathfrak{r}) \leq \frac{2C}{N} + 2qC \left( \frac{(4C(1-q)B_{\tilde{f}})^{1-q} C^2 a_3}{N(1-q)^2} \right)^{\frac{1}{1+q}} + 2C \sqrt{\frac{a_3}{N(1-q)}} \mathfrak{r}^{2(1-q)}$$

$$\leq \frac{2C}{N} + \frac{2qC^2 \cdot 4^{\frac{1-q}{1+q}}}{1-q} \left( \frac{B_{\tilde{f}}^{1-q} a_3}{N} \right)^{\frac{1}{1+q}} + \frac{2C}{(1-q)} \mathfrak{r}^{(1-q)} \sqrt{\frac{a_3}{N}}$$

$$\leq \frac{C_q}{N} + C_q \left( \frac{B_{\tilde{f}}^{1-q} a_3}{N} \right)^{\frac{1}{1+q}} + C_q \mathfrak{r}^{(1-q)} \sqrt{\frac{a_3}{N}},$$

where $C_q > 0$ is a universal constant only depending on $q$.

Then, we obtain the bound on $\phi(\mathfrak{r})$ in Eq. (58). The bound on $\mathfrak{r}_*^2$ can be obtained by simple calculations based on Eqs. (52) and (58). □

**Lemma 28.** *When the population $L_2$-norm of any function $h = \tilde{f} - \tilde{g} \in (\mathfrak{F}_{\delta,\mathbf{r}}^{\text{adv}} - \mathfrak{F}_{\mathbf{r}}^{\text{adv}})$ is upper bounded by $\mathfrak{r}$, its squared empirical $L_2$-norm can be upper bouned as follows:*

$$\mathbb{E}_S \left[ \sup_h \left( \frac{1}{N} \sum_{i=1}^N h(\underline{\mathbf{x}}_i, y_i)^2 \,\bigg|\, h \in (\mathfrak{F}_{\delta,\mathbf{r}}^{\text{adv}} - \mathfrak{F}_{\mathbf{r}}^{\text{adv}}) \text{ and } \|h\|_{L_2} \leq \mathfrak{r} \right) \right] \leq 2 B_{\tilde{f}} \dot{R}_{\mathfrak{r}}(\mathfrak{F}_{\delta,\mathbf{r}}^{\text{adv}} - \mathfrak{F}_{\mathbf{r}}^{\text{adv}}) + \mathfrak{r}^2.$$

$$(59)$$

*Proof.* The squared empirical $L_2$-norm can be upper bouned based on the population $L_2$-norm as follows

$$
\mathbb{E}_S \left[ \sup \left( \frac{1}{N} \sum_{i=1}^{N} h(\underline{\mathbf{x}}_i, y_i)^2 \mid h \in (\mathfrak{F}_{\delta,\mathbf{r}}^{\mathrm{adv}} - \mathfrak{F}_{\mathbf{r}}^{\mathrm{adv}}) \text{ and } \|h\|_{L_2} \leq \mathfrak{r} \right) \right]
$$

$$
\leq \mathbb{E} \left[ \sup \left( \frac{1}{N} \sum_{i=1}^{N} h(\underline{\mathbf{x}}_i, y_i)^2 - \mathbb{E}_{(\underline{\mathbf{x}}_i, y_i)}[h(\underline{\mathbf{x}}, y)^2] \mid h \in (\mathfrak{F}_{\delta,\mathbf{r}}^{\mathrm{adv}} - \mathfrak{F}_{\mathbf{r}}^{\mathrm{adv}}) \text{ and } \|h\|_{L_2} \leq \mathfrak{r} \right) \right] + \mathfrak{r}^2
$$

$$
\overset{(i)}{\leq} 2\mathbb{E}_{S,\boldsymbol{\varepsilon}} \left[ \sup \left( \frac{1}{N} \sum_{i=1}^{N} \varepsilon_i h(\underline{\mathbf{x}}_i, y_i)^2 \mid h \in (\mathfrak{F}_{\delta,\mathbf{r}}^{\mathrm{adv}} - \mathfrak{F}_{\mathbf{r}}^{\mathrm{adv}}) \text{ and } \|h\|_{L_2} \leq \mathfrak{r} \right) \right] + \mathfrak{r}^2
$$

$$
\overset{(ii)}{\leq} 2B_{\tilde{f}} \mathbb{E}_{S,\boldsymbol{\varepsilon}} \left[ \sup \left( \frac{1}{N} \sum_{i=1}^{N} \varepsilon_i h(\underline{\mathbf{x}}_i, y_i) \mid h \in (\mathfrak{F}_{\delta,\mathbf{r}}^{\mathrm{adv}} - \mathfrak{F}_{\mathbf{r}}^{\mathrm{adv}}) : \|h\|_{L_2} \leq \mathfrak{r} \right) \right] + \mathfrak{r}^2
$$

$$
= 2B_{\tilde{f}} \dot{R}_{\mathfrak{r}}(\mathfrak{F}_{\delta,\mathbf{r}}^{\mathrm{adv}} - \mathfrak{F}_{\mathbf{r}}^{\mathrm{adv}}) + \mathfrak{r}^2
$$

$$
\leq 2B_{\tilde{f}} \phi(\mathfrak{r}) + \mathfrak{r}^2,
$$

(60)

where $(i)$ is due to the symmertrization argument [47] and $\boldsymbol{\varepsilon} = \{\varepsilon_i\}_{i=1}^{N}$ are *i.i.d.* Rademacher variables, and $(ii)$ holds because of the contraction inequality (Lemma 30). □

## E.2 Adversarial Generalization Gap under Assumption 13

*Proof of Theorem 14.* In this situation, we can see that for any $1 \leq r_l \leq \min\{d_l, d_{l-1}\}$, we can approximate $\underline{\mathbf{W}}^{(l)}$ with its optimal tubal-rank-$r_l$ approximation tensor $\underline{\mathbf{W}}_{r_l}^{(l)}$ according to [20, Theorem 3.7] and achieve the following approximation error bound on F-norm

$$
\left\| \underline{\mathbf{W}}^{(l)} - \underline{\mathbf{W}}_{r_l}^{(l)} \right\|_{\mathrm{F}} = \left\| \underline{\mathbf{W}}^{(l)} \times_3 \mathbf{M} - \underline{\mathbf{W}}_{r_l}^{(l)} \times_3 \mathbf{M} \right\|_{\mathrm{F}}
$$

$$
\leq \sqrt{ \sum_{k=1}^{\mathsf{c}} \sum_{j=r_l+1}^{\min\{d_l, d_{l-1}\}} \sigma_j^2 (\underline{\mathbf{W}}^{(l)} \times_3 \mathbf{M})_{:,:,k} }
$$

$$
\leq \sqrt{ \sum_{k=1}^{\mathsf{c}} \sum_{j=r_l+1}^{\min\{d_l, d_{l-1}\}} (V_0 \cdot j^{-\alpha})^2 }
$$

$$
\leq \sqrt{ \sum_{k=1}^{\mathsf{c}} \frac{1}{2\alpha - 1} V_0^2 (r_l - 1)^{1-2\alpha} }
$$

$$
\overset{(i)}{\leq} \sqrt{ \frac{\mathsf{c}}{2\alpha - 1} } V_0 r_l^{(1-2\alpha)/2} := \delta_l^{\mathrm{F}},
$$

where $(i)$ holds because of Lemma 41.

We also have a specral norm bound for $\underline{\mathbf{W}}^{(l)} - \underline{\mathbf{W}}_{r_l}^{(l)}$ according to [20, Theorem 3.7] under Assumption 13 as follows

$$
\left\| \underline{\mathbf{W}}^{(l)} - \underline{\mathbf{W}}_{r_l}^{(l)} \right\|_{\mathrm{sp}} \leq V_0 (r_l + 1)^{-\alpha} := \delta_l^{\mathrm{sp}},
$$

and we also have

$$
\sqrt{\mathsf{c}} \delta_l^{\mathrm{sp}} \leq \delta_l^{\mathrm{F}}.
\tag{61}
$$

Consider function $g(\underline{\mathbf{x}}) = g(\underline{\mathbf{x}}; \underline{\mathbf{W}}_{\mathbf{r}})$ parameterized by $\underline{\mathbf{W}}_{\mathbf{r}} = (\underline{\mathbf{W}}_{r_1}^{(1)}, \cdots, \underline{\mathbf{W}}_{r_L}^{(L)}, \mathbf{w})$ as the function whose t-product layer weights are low-tubal-rank approximations of $f(\underline{\mathbf{x}}; \underline{\mathbf{W}})$. Let $\tilde{f}(\underline{\mathbf{x}}, y) = \inf_{R_{\mathrm{a}}(\underline{\mathbf{x}} - \underline{\mathbf{x}}') \leq \xi} y f(\underline{\mathbf{x}}')$ and $\tilde{g}(\underline{\mathbf{x}}, y) = \inf_{R_{\mathrm{a}}(\underline{\mathbf{x}} - \underline{\mathbf{x}}') \leq \xi} y g(\underline{\mathbf{x}}')$ denote the adversarial versions of $f$ and $g$, respectively.

This helps us bounding $\hat{\mathfrak{r}}$ as follows

$$
|\tilde{f}(\underline{\mathbf{x}}_i, y_i) - \tilde{g}(\underline{\mathbf{x}}_i, y_i)| = | \inf_{R_{\mathrm{a}}(\underline{\mathbf{x}}_i - \underline{\mathbf{x}}_i') \leq \xi} y_i f(\underline{\mathbf{x}}_i') - \inf_{R_{\mathrm{a}}(\underline{\mathbf{x}}_i - \underline{\mathbf{x}}_i') \leq \xi} y_i g(\underline{\mathbf{x}}_i')|.
$$

Letting $\mathbf{x}_i^f = \arginf_{R_a(\mathbf{x}_i - \mathbf{x}_i') \leq \xi} y_i f(\mathbf{x}_i')$ and $\mathbf{x}_i^g = \arginf_{R_a(\mathbf{x}_i - \mathbf{x}_i') \leq \xi} y_i g(\mathbf{x}_i')$, we have $|\tilde{f}(\mathbf{x}_i, y_i) - \tilde{g}(\mathbf{x}_i, y_i)| = |y_i f(\mathbf{x}_i^f) - y_i g(\mathbf{x}_i^g)|$. By letting

$$
\mathbf{x}_i^\xi = \begin{cases} \mathbf{x}_i^g & \text{if} \quad y_i f(\mathbf{x}_i^f) \geq y_i g(\mathbf{x}_i^g) \\ \mathbf{x}_i^f & \text{otherwise} \end{cases},
$$

we obtain

$$
|\tilde{f}(\mathbf{x}_i, y_i) - \tilde{g}(\mathbf{x}_i, y_i)| = |y_i f(\mathbf{x}_i^f) - y_i g(\mathbf{x}_i^g)| \leq |y_i f(\mathbf{x}_i^\xi) - y_i g(\mathbf{x}_i^\xi)| = |f(\mathbf{x}_i^\xi) - g(\mathbf{x}_i^\xi)|.
$$

Let $h_l(\mathbf{x}^\xi) = \mathbf{w}^\top \mathrm{vec}\left( \sigma(\underline{\mathbf{W}}_{r_l}^{(L)} *_M \sigma(\underline{\mathbf{W}}_{r-1}^{(L-1)} *_M \cdots *_M \sigma(\underline{\mathbf{W}}_r^{(l+1)} *_M \sigma(\underline{\mathbf{W}}^{(l)} *_M \cdots *_M \sigma(\underline{\mathbf{W}}^{(1)} *_M \mathbf{x}^\xi) \cdots))) \right)$ and $h_0(\mathbf{x}^\xi) = g(\mathbf{x}^\xi)$. Then, we have

$$
|f(\mathbf{x}_i^\xi) - g(\mathbf{x}_i^\xi)| \leq \sum_{l=1}^{L} |h_l(\mathbf{x}_i^\xi) - h_{l-1}(\mathbf{x}_i^\xi)|.
$$

We can see that for any $l = 1, \cdots, L$:

$$
\left\| \sigma(\underline{\mathbf{W}}^{(l-1)} *_M \cdots *_M \sigma(\underline{\mathbf{W}}^{(1)} *_M \mathbf{x}_i^\xi) \cdots)) \right\|_F \leq \prod_{l'=1}^{l-1} \left\| \underline{\mathbf{W}}^{(l')} \right\|_F \left\| \mathbf{x}_i^\xi \right\|_F \leq \prod_{l'=1}^{l-1} \left\| \underline{\mathbf{W}}^{(l')} \right\|_F B_{x, R_a, \xi},
$$

and

$$
\left\| \sigma(\underline{\mathbf{W}}_{r_l}^{(l)} *_M \sigma(\underline{\mathbf{W}}^{(l-1)} *_M \cdots *_M \sigma(\underline{\mathbf{W}}^{(1-1)} *_M \mathbf{x}_i^\xi) \cdots)) - \sigma(\underline{\mathbf{W}}^{(l)} *_M \sigma(\underline{\mathbf{W}}^{(l-1)} *_M \cdots *_M \sigma(\underline{\mathbf{W}}^{(1)} *_M \mathbf{x}_i^\xi) \cdots)) \right\|_F
$$

$$
\leq \left\| \underline{\mathbf{W}}_{r_l}^{(l)} *_M \sigma(\underline{\mathbf{W}}^{(l-1)} *_M \cdots *_M \sigma(\underline{\mathbf{W}}^{(1)} *_M \mathbf{x}_i^\xi) \cdots) - \underline{\mathbf{W}}^{(l)} *_M \sigma(\underline{\mathbf{W}}^{(l-1)} *_M \cdots *_M \sigma(\underline{\mathbf{W}}^{(1)} *_M \mathbf{x}_i^\xi) \cdots) \right\|_F
$$

$$
= \left\| (\underline{\mathbf{W}}_{r_l}^{(l)} - \underline{\mathbf{W}}^{(l)}) *_M \sigma(\underline{\mathbf{W}}^{(l-1)} *_M \cdots *_M \sigma(\underline{\mathbf{W}}^{(1)} *_M \mathbf{x}_i^\xi) \cdots) \right\|_F
$$

$$
\overset{(i)}{\leq} \left\| \underline{\mathbf{W}}_{r_l}^{(l)} - \underline{\mathbf{W}}^{(l)} \right\|_{sp} \left\| \sigma(\underline{\mathbf{W}}^{(l-1)} *_M \cdots *_M \sigma(\underline{\mathbf{W}}^{(1)} *_M \mathbf{x}_i^\xi) \cdots) \right\|_F
$$

$$
\leq \delta_l^{sp} \prod_{l'=1}^{l} \left\| \underline{\mathbf{W}}^{(l')} \right\|_F B_{x, R_a, \xi},
$$

where in $(i)$ we relax the inequality by using the tensor spectral norm of insted of the F-norm of $\underline{\mathbf{W}}_{r_l}^{(l)} - \underline{\mathbf{W}}^{(l)}$ which leads to a smaller upper bound. Thus, we have

$$
|h_l(\mathbf{x}_i^\xi) - h_{l-1}(\mathbf{x}_i^\xi)| \leq B_{\mathbf{w}} \prod_{j \neq l} B_j \delta_l^{sp} B_{x, R_a, \xi} = \frac{\delta_l^{sp} B_{\tilde{f}}}{B_l}.
$$

This gives

$$
|f(\mathbf{x}_i^\xi) - g(\mathbf{x}_i^\xi)| \leq \sum_{l=1}^{L} |h_l(\mathbf{x}_i^\xi) - h_{l-1}(\mathbf{x}_i^\xi)| \leq \sum_{l=1}^{L} \frac{\delta_l^{sp} B_{\tilde{f}}}{B_l}.
$$

Then, we can set

$$
\hat{\mathfrak{r}} = \sum_{l=1}^{L} \frac{\delta_l^{sp} B_{\tilde{f}}}{B_l} = V_0 B_{\tilde{f}} \sum_{l=1}^{L} \frac{(r_l + 1)^{-\alpha}}{B_l}. \tag{62}
$$

Then we can construct an $\epsilon$-cover of $\mathfrak{F}_{\delta, \mathbf{r}}^{adv}$ composed of $\tilde{g}(\mathbf{x}; \mathbf{W_r})$ by carefully setting the value of rank parameter $\mathbf{r} = (r_1, \cdots, r_L)^\top$ according to the covering accuracy $\epsilon$. Directly setting $\frac{\delta_l^{sp} D_{\tilde{f}}}{2B_l} = \epsilon / L$, we obtain

$$
r_l = \min\left\{ \left\lceil A_l \cdot \epsilon^{-\frac{1}{\alpha}} \right\rceil - 1, d_l, d_{l-1} \right\} \leq \min\left\{ A_l \cdot \epsilon^{-\frac{1}{\alpha}}, d_l, d_{l-1} \right\},
$$

where

$$
A_l = \left( \frac{B_l}{LV_0 B_{\tilde{f}}} \right)^{-\frac{1}{\alpha}}.
$$

Then, we obtain the covering entropy of $\mathfrak{F}^{\text{adv}}_{\delta,\mathbf{r}}$ by

$$
\begin{aligned}
\log \mathsf{N}(\mathfrak{F}^{\text{adv}}_{\delta,\mathbf{r}}, \|\cdot\|_S, \epsilon) &\leq \log \prod_{l=1}^{L} \left( \frac{9B_l}{\delta_l^{\text{F}}} \right)^{r_l(d_l + d_{l-1} + 1)\mathsf{c}} \\
&\overset{(i)}{\leq} \log \prod_{l=1}^{L} \left( \frac{9B_l}{\sqrt{\mathsf{c}}\delta_l^{\text{sp}}} \right)^{r_l(d_l + d_{l-1} + 1)\mathsf{c}} \\
&\leq \sum_{l=1}^{L} r_l(d_l + d_{l-1} + 1)\mathsf{c}\log(9LB_{\tilde{f}}/(\sqrt{\mathsf{c}}\epsilon)) \\
&\leq \sum_{l}(A_l\epsilon^{-\frac{1}{\alpha}})(d_l + d_{l-1} + 1)\mathsf{c}\left( \log(\epsilon^{-1}) + \log(9LB_{\tilde{f}}/(\sqrt{\mathsf{c}})) \right),
\end{aligned}
$$

(63)

where $(ii)$ holds by Eq. (61).

Since $\mathfrak{F}^{\text{adv}}_{\mathbf{r}} \subset \mathfrak{F}^{\text{adv}}_{\delta,\mathbf{r}}$, we have

$$
\log \mathsf{N}(\mathfrak{F}^{\text{adv}}_{\mathbf{r}}, \|\cdot\|_S, \epsilon) \leq \log \mathsf{N}(\mathfrak{F}^{\text{adv}}_{\delta,\mathbf{r}}, \|\cdot\|_S, \epsilon) \leq \sum_{l=1}^{L} r_l(d_l + d_{l-1} + 1)\mathsf{c}\log(9LB_{\tilde{f}}/(\sqrt{\mathsf{c}}\epsilon))
$$

To use Lemma 27, we bound $\log \mathsf{N}(\mathfrak{F}^{\text{adv}}_{\delta,\mathbf{r}}, \|\cdot\|_S, \epsilon/2) + \mathsf{N}(\mathfrak{F}^{\text{adv}}_{\mathbf{r}}, \|\cdot\|_S, \epsilon/2)$ when $\epsilon \geq 2/N$ as follows:

$$
\log \mathsf{N}(\mathfrak{F}^{\text{adv}}_{\delta,\mathbf{r}}, \|\cdot\|_S, \epsilon/(2)) + \mathsf{N}(\mathfrak{F}^{\text{adv}}_{\mathbf{r}}, \|\cdot\|_S, \epsilon/2) \leq a_1 + a_2\log(\epsilon^{-1}) + a_3\epsilon^{-\frac{1}{\alpha}},
$$

where

$$
\begin{aligned}
a_1 &= \log(9LB_{\tilde{f}}/\sqrt{\mathsf{c}})a_2, \\
a_2 &= \mathsf{c}\sum_{l=1}^{L} r_l(d_l + d_{l-1} + 1), \\
a_3 &= \left( \log N + \log(9LB_{\tilde{f}}/\sqrt{\mathsf{c}}) \right)\mathsf{c}\sum_{l=1}^{L} A_l(d_l + d_{l-1} + 1).
\end{aligned}
$$

For simplicity, further let

$$
\begin{aligned}
E_1 &= \frac{a_1 + a_2\log N}{N} = \frac{\mathsf{c}\sum_{l=1}^{L} r_l(d_l + d_{l-1} + 1)}{N}\log(9NLB_{\tilde{f}}/\sqrt{\mathsf{c}}), \\
E_2 &= \frac{a_3}{N} = \frac{\mathsf{c}\sum_{l=1}^{L} A_l(d_l + d_{l-1} + 1)}{N}\log(9NLB_{\tilde{f}}/\sqrt{\mathsf{c}}).
\end{aligned}
$$

(64)

Then, according to Lemma 27, we have

$$
\mathfrak{r}^2_*(t) \leq C\left\{ B_{\tilde{f}}E_1 + \frac{1 + tB_{\tilde{f}}}{N}, \quad E_2^{\frac{2\alpha}{2\alpha+1}}\left( B_{\tilde{f}}^{\frac{2\alpha-1}{2\alpha+1}} + 1 \right) \right\},
$$

(65)

which further leads to

$$
\begin{aligned}
&\Phi(\dot{\mathfrak{r}}) + L_\ell\dot{\mathfrak{r}}\sqrt{\frac{t}{N}} + \frac{tL_\ell B_{\tilde{f}}}{N} \\
&\leq 2L_\ell\phi(\dot{\mathfrak{r}}) + 2L_\ell(\hat{\mathfrak{r}} + \mathfrak{r}_*)\sqrt{\frac{t}{N}} + \frac{tL_\ell B_{\tilde{f}}}{N} \\
&\leq C_q L_\ell\max\left\{ \frac{1}{N} + B_{\tilde{f}}E_1 + (\hat{\mathfrak{r}} + \mathfrak{r}_*)\sqrt{E_1}, \ E_2^{\frac{2\alpha}{2\alpha+1}}B_{\tilde{f}}^{\frac{2\alpha-1}{2\alpha+1}} + (\hat{\mathfrak{r}} + \mathfrak{r}_*)^{\frac{2\alpha}{2\alpha+1}}\sqrt{E_2} \right\} \\
&\quad + 2L_\ell(\hat{\mathfrak{r}} + \mathfrak{r}_*)\sqrt{\frac{t}{N}} + \frac{tL_\ell B_{\tilde{f}}}{N}.
\end{aligned}
$$

(66)

Note that

$$
(\hat{\mathfrak{r}} + \mathfrak{r}_*)\sqrt{E_1} = \hat{\mathfrak{r}}\sqrt{E_1} + \mathfrak{r}_*\sqrt{E_1} \leq \hat{\mathfrak{r}}\sqrt{E_1} + \frac{1}{2}\mathfrak{r}_*^2 + \frac{1}{2}E_1,
$$

$$(\hat{\mathfrak{r}} + \mathfrak{r}_*)^{\frac{2\alpha}{2\alpha+1}} \sqrt{E_2} \leq \hat{\mathfrak{r}}^{\frac{2\alpha}{2\alpha+1}} \sqrt{E_2} + \mathfrak{r}_*^{\frac{2\alpha}{2\alpha+1}} \sqrt{E_2},$$

$$(\hat{\mathfrak{r}} + \mathfrak{r}_*)\sqrt{\frac{t}{N}} \leq \hat{\mathfrak{r}}\sqrt{\frac{t}{N}} + \frac{1}{2}(\mathfrak{r}_*^2 + \frac{t}{N}).$$

Then by simple calculation, we have

$$\Phi(\dot{\mathfrak{r}}) + L_\ell \dot{\mathfrak{r}}\sqrt{\frac{t}{N}} + \frac{tL_\ell B_{\tilde{f}}}{N}$$

$$\overset{(i)}{\leq} C_\alpha L_\ell \left\{ B_{\tilde{f}} E_1 + \hat{\mathfrak{r}}\sqrt{E_1} + E_2^{\frac{2\alpha}{2\alpha+1}} \left( B_{\tilde{f}}^{\frac{2\alpha-1}{2\alpha+1}} + 1 \right) + \hat{\mathfrak{r}}^{\frac{2\alpha}{2\alpha+1}} \sqrt{E_2} + \hat{\mathfrak{r}}\sqrt{\frac{t}{N}} + \frac{1 + tB_{\tilde{f}}}{N} \right\}.$$

$\square$

*Proof of Corollary 15.* The bound in Corollary 15 can be directly obtained if we choose the parameter **r** of tubal ranks in $\mathfrak{F}_{\mathbf{r}}$ by $r_l = \min\{\lceil (LV_0 B_{\tilde{f}} B_l^{-1})^{1/\alpha}\rceil, d_l, d_{l-1}\}$. $\square$

# F  Useful Notions and Lemmas

In this section, we provide several notions and lemmas which are used in the previous analysis.

## F.1  Tools for Analyzing General DNNs

We briefly list the tools used in this paper for analyzing the generalization error of general DNNs, including Rademacher complexity, covering number, and concentration inequalities, *etc.*.

**Definition 13** (Rademacher complexity). *Given an i.i.d. sample $S := \{(\mathbf{x}_i, y_i)\}_{i=1}^N$ of size $N$ and a function class $\mathcal{H}$, the empirical Rademacher complexity of $\mathcal{H}$ is defined as*

$$\hat{R}_S(\mathcal{H}) := \mathbb{E}_{\varepsilon_1, \cdots, \varepsilon_N} \left[ \sup_{h \in \mathcal{H}} \frac{1}{N} \varepsilon_i h(\mathbf{x}_i, y_i) \right],$$

*where $\varepsilon_1, \cdots, \varepsilon_N$ are i.i.d. Rademacher variables, i.e., $\varepsilon_i$ equals to $1$ or $-1$ with equal probability. The average Rademacher complexity is further defined as*

$$\bar{R}_N = \mathbb{E}_S \hat{R}_S(\mathcal{H}).$$

**Lemma 29** ([4]). *Given an i.i.d. sample $S := \{(\mathbf{x}_i, y_i)\}_{i=1}^N$ of size $N$, a loss function $\ell(h(\cdot), y)$ taking values in $[0, B]$, the generalization error of any function $f$ in hypothesis set $\mathcal{F}$ satisfies*

$$\mathcal{L}(f) \leq \hat{\mathcal{L}}(f) + 2\hat{R}_S(\ell \circ \mathcal{F}) + 3B\sqrt{\frac{t}{2N}}, \tag{67}$$

*with probability at least $1 - e^{-t}$ for all $t \geq 0$.*

**Lemma 30** (Talagrand's contraction lemma [44]). *Given function set $\mathcal{F}$ and $L_\ell$-Lipschtz function $\ell$, for a function sets defined as $l_\mathcal{F} := \{\ell \circ f \mid f \in \mathcal{F}\}$, we have*

$$\hat{R}_S(\mathcal{F}) \leq L_\ell \hat{R}_S(\mathcal{F}).$$

**Definition 14** ($\epsilon$-covering net). *Let $\epsilon > 0$ and $(\mathcal{X}, d(\cdot, \cdot))$ be a metric space, where $d(\cdot, \cdot)$ is a (pseudo)-metric. We say $\mathcal{Z} \subset \mathcal{X}$ is an $\epsilon$-covering net of $\mathcal{X}$, if for any $x \in \mathcal{X}$, there exists $z \in \mathcal{Z}$ such that $d(x, z) \leq \epsilon$. Define the smallest $|\mathcal{Z}|$ as the $\epsilon$-covering number of $\mathcal{X}$ and denote as $\mathsf{N}(\mathcal{X}, d(\cdot, \cdot), \epsilon)$.*

Given a traning dataset $S = \{\mathbf{x}_i, y_i\}_{i=1}^N$ and a function set $\mathcal{F}$. Consider the output space of the space of $\mathcal{F}$ restricted on $S$, i.e., $\mathcal{F}|_S = \{(f(\mathbf{x}_1, y_1), \cdots, f(\mathbf{x}_N, y_N))^\top \mid f \in \mathcal{F}\}$. Then, define a pseudo-norm of $\mathcal{F}|_S$ as:$\|f\|_S := N^{-1}\sqrt{\sum_{i=1}^N f(\mathbf{x}_i, y_i)^2}$. Then, the Rademacher complexity of $\mathcal{F}$ could be upper bounded by the $\epsilon$-covering number of $\mathcal{F}$ under the empirical $l_2$-pseudo-metric by the Dudley's inequality as follows:

**Lemma 31** (Dudley's integral inequality [47]). *The Rademacher complexity $\hat{R}_S(\mathcal{F})$ satisfies*

$$\hat{R}_S(\mathcal{F}) \leq \inf_{\delta > 0} \left[ 8\delta + \frac{12}{\sqrt{N}} \int_{\delta}^{\max_{f \in \mathcal{F}} \|f\|_S} \sqrt{\log \mathsf{N}(\mathcal{F}, \|\cdot\|_S, \epsilon)} d\epsilon \right]. \tag{68}$$

**Lemma 32** (Covering number of norm balls [47]). *Let $\mathbf{B}$ be a $l_p$-norm ball with radius $W$. Let $d(\mathbf{x}_1, \mathbf{x}_2) = \|\mathbf{x}_1 - \mathbf{x}_2\|_p$. Define the $\epsilon$-covering number of $\mathbf{B}$ as $\mathsf{N}(\mathbf{B}, d(\cdot, \cdot), \epsilon)$, we have*

$$\mathsf{N}(\mathbf{B}, d(\cdot, \cdot), \epsilon) \leq \left( 1 + \frac{2W}{\epsilon} \right)^d. \tag{69}$$

**Lemma 33** (Covering number of low-tubal-rank tensors). *For the set of tensors $\mathbf{T}_r := \{\underline{\mathbf{T}} \in \mathbb{R}^{m \times n \times \mathsf{c}} \mid r_\mathsf{t}(\underline{\mathbf{T}}) \leq r, \|\underline{\mathbf{T}}\|_\mathrm{F} \leq 1\}$ with $r \leq \min\{m, n\}$, its $\epsilon$-covering number can be upper bounded by*

$$\mathsf{N}(\mathbf{T}_r, \|\cdot\|_\mathrm{F}, \epsilon) \leq \left( \frac{9}{\epsilon} \right)^{(m+n+1)r\mathsf{c}}. \tag{70}$$

*Proof.* Consider the reduced t-SVD [20] of a tensor $\underline{\mathbf{X}} = \underline{\mathbf{U}} *_M \underline{\mathbf{S}} *_M \underline{\mathbf{V}}^\top \in \mathbf{T}_r$, where $*_M$ denotes the t-product induced by the linear transform $M(\cdot)$ defined in Eq. (1) , $\underline{\mathbf{U}} \in \mathbb{R}^{m \times r \times \mathsf{c}}$ and $\underline{\mathbf{V}} \in \mathbb{R}^{n \times r \times \mathsf{c}}$ are (semi)-t-orthogonal tensors, and $\underline{\mathbf{S}} \in \mathbb{R}^{r \times r \times \mathsf{c}}$ is an f-diagonal tensor. As $\underline{\mathbf{T}} \in \mathbf{T}_r$, we have $\|\underline{\mathbf{S}}\|_\mathrm{F} = \|\underline{\mathbf{X}}\|_\mathrm{F} \leq 1$. The idea is to cover $\mathbf{T}_r$ by covering the set of factor tensors $\underline{\mathbf{U}}, \underline{\mathbf{V}}$ and $\underline{\mathbf{S}}$.

Let $\mathbf{D} \subset \mathbb{R}^{r \times r \times \mathsf{c}}$ be the set of f-diagonal tensors with F-norm equal to one. We take $\mathbf{D}^c$ to be an $\epsilon/3$-covering net for $\mathbf{D}$. Then by Lemma 32, we have $|\mathbf{D}^c| \leq (9/\epsilon)^{r\mathsf{c}}$. Next, let $\mathbf{O}_{m,r} = \{\underline{\mathbf{U}} \in \mathbb{R}^{m \times r \times \mathsf{c}} \mid \underline{\mathbf{U}} *_M \underline{\mathbf{U}}^\top = \underline{\mathbf{I}}\}$. To cover $\mathbf{O}_{m,r}$, we consider the $\|\cdot\|_{\infty,2,2}$-norm defined as

$$\|\underline{\mathbf{X}}\|_{\infty,2,2} := \max_i \left\| \underline{\mathbf{X}}_{:,i,:} \right\|_\mathrm{F}.$$

Let $\mathbf{Q}_{m,r} := \{\underline{\mathbf{X}} \in \mathbb{R}^{m \times r \times \mathsf{c}} \mid \|\underline{\mathbf{X}}\|_{\infty,2,2} \leq 1\}$. Then, we have $\mathbf{O}_{m,r} \subset \mathbf{Q}_{m,r}$ by the definition of t-orthogonal tensors. Letting $\mathbf{Q}_{m,r}^c$ be an $\epsilon/3$-covering net of $\mathbf{Q}_{m,r}$, then we obtain $|\mathbf{Q}_{m,r}^c| \leq (9/\epsilon)^{mr\mathsf{c}}$ by Theorem 32. Similarly, an $\epsilon/3$-covering net of $\mathbf{Q}_{n,r}$ satisfies $|\mathbf{Q}_{n,r}^c| \leq (9/\epsilon)^{nr\mathsf{c}}$.

Now construct a set $\mathbf{T}_r^c = \{\underline{\mathbf{U}}^c *_M \underline{\mathbf{S}}^c *_M (\underline{\mathbf{V}}^c)^\top \mid \underline{\mathbf{U}}^c \in \mathbf{Q}_{m,r}^c, \underline{\mathbf{S}}^c \in \mathbf{D}^c, \underline{\mathbf{V}}^c \in \mathbf{Q}_{n,r}^c\}$. Then, we have

$$|\mathbf{T}_r^c| = |\mathbf{Q}_{m,r}^c| \cdot |\underline{\mathbf{S}}^c| \cdot |\mathbf{Q}_{n,r}^c| \leq (9/\epsilon)^{(m+n+1)r\mathsf{c}}.$$

Net, we will show that $\mathbf{T}_r^c$ is an $\epsilon$-covering net of $\mathbf{T}_r$, i.e., for any $\underline{\mathbf{X}} \in \mathbf{T}_r$, there is an $\underline{\mathbf{X}}^c \in \mathbf{T}_r^c$ satisfying $\|\underline{\mathbf{X}} - \underline{\mathbf{X}}^c\|_\mathrm{F} \leq \epsilon$.

Given $\underline{\mathbf{X}} \in \mathbf{T}_r$, consider its reduced t-SVD as $\underline{\mathbf{X}} = \underline{\mathbf{U}} *_M \underline{\mathbf{S}} *_M \underline{\mathbf{V}}^\top \in \mathbf{T}_r$. Then, there exists $\underline{\mathbf{X}}^c = \underline{\mathbf{U}}^c *_M \underline{\mathbf{S}}^c *_M (\underline{\mathbf{V}}^c)^\top$ with $\underline{\mathbf{U}}^c \in \mathbf{Q}_{m,r}^c, \underline{\mathbf{S}}^c \in \mathbf{D}^c, \underline{\mathbf{V}}^c \in \mathbf{Q}_{n,r}^c$ satisying $\|\underline{\mathbf{U}} - \underline{\mathbf{U}}^c\|_{\infty,2,2} \leq \epsilon/3$, $\|\underline{\mathbf{S}} - \underline{\mathbf{S}}^c\|_F \leq \epsilon/3$, and $\|\underline{\mathbf{V}} - \underline{\mathbf{V}}^c\|_{\infty,2,2} \leq \epsilon/3$. This gives

$$\|\underline{\mathbf{X}} - \underline{\mathbf{X}}^c\|_\mathrm{F}$$

$$= \left\| \underline{\mathbf{U}} *_M \underline{\mathbf{S}} *_M \underline{\mathbf{V}}^\top - \underline{\mathbf{U}}^c *_M \underline{\mathbf{S}}^c *_M (\underline{\mathbf{V}}^c)^\top \right\|_\mathrm{F}$$

$$= \left\| \underline{\mathbf{U}} *_M \underline{\mathbf{S}} *_M \underline{\mathbf{V}}^\top - \underline{\mathbf{U}}^c *_M \underline{\mathbf{S}} *_M \underline{\mathbf{V}}^\top + \underline{\mathbf{U}}^c *_M \underline{\mathbf{S}} *_M \underline{\mathbf{V}}^\top - \underline{\mathbf{U}}^c *_M \underline{\mathbf{S}}^c *_M \underline{\mathbf{V}}^\top \right.$$

$$\left. + \underline{\mathbf{U}}^c *_M \underline{\mathbf{S}}^c *_M \underline{\mathbf{V}}^\top - \underline{\mathbf{U}}^c *_M \underline{\mathbf{S}}^c *_M (\underline{\mathbf{V}}^c)^\top \right\|_\mathrm{F}$$

$$\leq \left\| \underline{\mathbf{U}} *_M \underline{\mathbf{S}} *_M \underline{\mathbf{V}}^\top - \underline{\mathbf{U}}^c *_M \underline{\mathbf{S}} *_M \underline{\mathbf{V}}^\top \right\|_\mathrm{F} + \left\| \underline{\mathbf{U}}^c *_M \underline{\mathbf{S}} *_M \underline{\mathbf{V}}^\top - \underline{\mathbf{U}}^c *_M \underline{\mathbf{S}}^c *_M \underline{\mathbf{V}}^\top \right\|_\mathrm{F}$$

$$+ \left\| \underline{\mathbf{U}}^c *_M \underline{\mathbf{S}}^c *_M \underline{\mathbf{V}}^\top - \underline{\mathbf{U}}^c *_M \underline{\mathbf{S}}^c *_M (\underline{\mathbf{V}}^c)^\top \right\|_\mathrm{F}$$

$$\leq \left\| \underline{\mathbf{U}} *_M \underline{\mathbf{S}} *_M \underline{\mathbf{V}}^\top - \underline{\mathbf{U}}^c *_M \underline{\mathbf{S}} *_M \underline{\mathbf{V}}^\top \right\|_\mathrm{F} + \left\| \underline{\mathbf{U}}^c *_M \underline{\mathbf{S}} *_M \underline{\mathbf{V}}^\top - \underline{\mathbf{U}}^c *_M \underline{\mathbf{S}}^c *_M \underline{\mathbf{V}}^\top \right\|_\mathrm{F}$$

$$+ \left\| \underline{\mathbf{U}}^c *_M \underline{\mathbf{S}}^c *_M \underline{\mathbf{V}}^\top - \underline{\mathbf{U}}^c *_M \underline{\mathbf{S}}^c *_M (\underline{\mathbf{V}}^c)^\top \right\|_\mathrm{F}$$

$$\leq \left\| (\underline{\mathbf{U}} - \underline{\mathbf{U}}^c) *_M \underline{\mathbf{S}} *_M \underline{\mathbf{V}}^\top \right\|_\mathrm{F} + \left\| \underline{\mathbf{U}}^c *_M (\underline{\mathbf{S}} - \underline{\mathbf{S}}^c) *_M \underline{\mathbf{V}}^\top \right\|_\mathrm{F} + \left\| \underline{\mathbf{U}}^c *_M \underline{\mathbf{S}}^c *_M (\underline{\mathbf{V}} - \underline{\mathbf{V}}^c)^\top \right\|_\mathrm{F}.$$

For the first term, note that since $\underline{\mathbf{V}}$ is a t-orthogonal tensor,

$$\left\|(\underline{\mathbf{U}} - \underline{\mathbf{U}}^c) *_M \underline{\mathbf{S}} *_M \underline{\mathbf{V}}^\top\right\|_{\mathrm{F}} = \|(\underline{\mathbf{U}} - \underline{\mathbf{U}}^c) *_M \underline{\mathbf{S}}\|_{\mathrm{F}}.$$

and

$$\|(\underline{\mathbf{U}} - \underline{\mathbf{U}}^c) *_M \underline{\mathbf{S}}\|_{\mathrm{F}}^2 = \sum_{i=1}^r \left\|(\underline{\mathbf{U}} - \underline{\mathbf{U}}^c)_{:,i,:} *_M \underline{\mathbf{S}}_{i,i,:}\right\|_{\mathrm{F}}^2$$

$$= \sum_{i=1}^r \|(\underline{\mathbf{U}} - \underline{\mathbf{U}}^c)_{:,i,:}\|_{\mathrm{F}}^2 \|\underline{\mathbf{S}}_{i,i,:}\|_{\mathrm{F}}^2$$

$$\leq (\sum_{i=1}^r \|\underline{\mathbf{S}}_{i,i,:}\|_{\mathrm{F}}^2) \max_i \|(\underline{\mathbf{U}} - \underline{\mathbf{U}}^c)_{:,i,:}\|_{\mathrm{F}}^2$$

$$\leq \|\underline{\mathbf{S}}\|_{\mathrm{F}}^2 \|\underline{\mathbf{U}} - \underline{\mathbf{U}}^c\|_{\infty,2,2}$$

$$\leq (\epsilon/3)^2.$$

Hence, $\left\|(\underline{\mathbf{U}} - \underline{\mathbf{U}}^c) *_M \underline{\mathbf{S}} *_M \underline{\mathbf{V}}^\top\right\|_{\mathrm{F}} \leq \epsilon/3$. Similarly, we have $\left\|\underline{\mathbf{U}}^c *_M \underline{\mathbf{S}}^c *_M (\underline{\mathbf{V}} - \underline{\mathbf{V}}^c)^\top\right\|_{\mathrm{F}} \leq \epsilon/3$. The middle term can be bounded $\left\|\underline{\mathbf{U}}^c *_M (\underline{\mathbf{S}} - \underline{\mathbf{S}}^c) *_M \underline{\mathbf{V}}^\top\right\|_{\mathrm{F}} \leq \|\underline{\mathbf{S}} - \underline{\mathbf{S}}^c\|_{\mathrm{F}} \leq \epsilon/3$ due to the property of t-orthognal tensors $\underline{\mathbf{U}}^c$ and $\underline{\mathbf{V}}^c$ [20]. Therefore, for any $\underline{\mathbf{X}} \in \mathbf{T}_r$, there is an $\underline{\mathbf{X}}^c \in \mathbf{T}_r^c$ satisfying $\|\underline{\mathbf{X}} - \underline{\mathbf{X}}^c\|_{\mathrm{F}} \leq \epsilon$. $\square$

**Lemma 34** (Covering number bounds for composition and addtion [10])**.** *Let $\mathcal{F}_1$ and $\mathcal{F}_2$ be classes of functions on normed space $(\mathcal{X}, \|\cdot\|_{\mathcal{X}}) \to (\mathcal{Y}, \|\cdot\|_{\mathcal{Y}})$ and let $\mathcal{F}$ be a class of c-Lipschitz functions $(\mathcal{Y}, \|\cdot\|_{\mathcal{Y}}) \to (\mathcal{Z}, \|\cdot\|_{\mathcal{Z}})$. Then for any $X \in \mathcal{X}^N$ and $\epsilon_{\mathcal{F}_1}, \epsilon_{\mathcal{F}_2}, \epsilon_{\mathcal{F}} > 0$, it holds that*

$$\mathsf{N}(\{f_1 + f_2 \mid f_1 \in \mathcal{F}_1, f_2 \in \mathcal{F}_2\}, \epsilon_{\mathcal{F}_1} + \epsilon_{\mathcal{F}_2}, \|\cdot\|_X) \leq \mathsf{N}(\mathcal{F}_1, \epsilon_{\mathcal{F}_1}, \|\cdot\|_X)\mathsf{N}(\mathcal{F}_2, \epsilon_{\mathcal{F}_2}, \|\cdot\|_X).$$

*and*

$$\mathsf{N}(\{f \circ f_1 \mid f \in \mathcal{F}, f_1 \in \mathcal{F}_1\}, \epsilon_{\mathcal{F}} + c\epsilon_{\mathcal{F}_1}, \|\cdot\|_X) \leq \mathsf{N}(\mathcal{F}_1, \epsilon_{\mathcal{F}_1}, \|\cdot\|_X) \sup_{f_1 \in \mathcal{F}_1} \mathsf{N}(\mathcal{F}, \epsilon_{\mathcal{F}}, \|\cdot\|_{f_1(X)}).$$

*Specifically, if $\mathcal{F} = \{f\}$ is a singleton, we have*

$$\mathsf{N}(\{f \circ f_1 \mid f \in \mathcal{F}, f_1 \in \mathcal{F}_1\}, c\epsilon_{\mathcal{F}_1}, \|\cdot\|_X) \leq \mathsf{N}(\mathcal{F}_1, \epsilon_{\mathcal{F}_1}, \|\cdot\|_X).$$

**Lemma 35** (Simplifided Talagrand's concentration inequality [42])**.** *Let $\mathcal{F}$ be a function class on $\mathcal{X}$ that is separable with respect to $L_\infty$-norm, and $\{\mathbf{x}_i\}_{i=1}^N$ be i.i.d. random variables in $\mathcal{X}$. Furthermore, suppose there exist constants $V \geq$ and $U \geq 0$ such that $V = \sup_{f \in \mathcal{F}} \mathbb{E}[(f - \mathbb{E}[f])^2]$ and $U = \sup_{f \in \mathcal{F}} \|f\|_{L_\infty}$. Letting $Z := \sup_{f \in \mathcal{F}} |N^{-1} \sum_{i=1}^N f(\mathbf{x}_i) - \mathbb{E}[f]|$, then it holds for all $t > 0$ that*

$$\mathbb{P}\left[Z \geq 2\mathbb{E}[Z] + \sqrt{\frac{2Vt}{N}} + \frac{2Ut}{N}\right] \leq e^{-t}.$$

### F.2 Some Results for Analyzing t-NNs

In this subsection, we present several fundamental statements described as lemmas for analyzing t-NNs. First, we give Lemmas 36-38, which are used in the analysis of the t-product layers.

**Lemma 36.** *Let $\sigma(\cdot) : \mathbb{R} \to \mathbb{R}$ be a $L_\sigma$-Lipschitz function, i.e., $|\sigma(x) - \sigma(y)| \leq L_\sigma|x - y|, \forall x, y \in \mathbb{R}$. If it is applied element-wisely to any two real vectors $\mathbf{x}$ and $\mathbf{y}$, then it holds that*

$$\|\sigma(\mathbf{x}) - \sigma(\mathbf{y})\|_{l_p} \leq L_\sigma \|\mathbf{x} - \mathbf{y}\|_{l_p}.$$

**Lemma 37.** *The following inequalities hold:*

$$\|\underline{\mathbf{T}}\|_{\mathrm{sp}} \leq \|\underline{\mathbf{T}}\|_{\mathrm{F}}, \quad \text{and} \quad \|\underline{\mathbf{W}} *_M \underline{\mathbf{x}}\|_{\mathrm{F}} \leq \|\underline{\mathbf{W}}\|_{\mathrm{sp}} \|\underline{\mathbf{x}}\|_{\mathrm{F}} \leq \|\underline{\mathbf{W}}\|_{\mathrm{F}} \|\underline{\mathbf{x}}\|_{\mathrm{F}}.$$

*Proof.* According to the definition of the $M$ transform in Eq. (1) and the orthogonality of $\mathbf{M}$, we have

$$\|\underline{\mathbf{T}}\|_{\mathrm{sp}} = \left\|\widetilde{\mathbf{T}}_M\right\| \leq \left\|\widetilde{\mathbf{T}}_M\right\|_{\mathrm{F}} = \|M(\underline{\mathbf{T}})\|_{\mathrm{F}} = \|\underline{\mathbf{T}}\|_{\mathrm{F}},$$

and

$$\|\underline{\mathbf{W}} *_M \underline{\mathbf{x}}\|_{\mathrm{F}} = \left\|\widetilde{\mathbf{W}}_M \cdot \widetilde{\mathbf{x}}_M\right\|_{\mathrm{F}} \overset{(i)}{\leq} \left\|\widetilde{\mathbf{W}}_M\right\| \|\widetilde{\mathbf{x}}_M\|_{\mathrm{F}} = \|\mathbf{W}\|_{\mathrm{sp}} \|\mathbf{x}\|_{\mathrm{F}} \leq \|\mathbf{W}\|_{\mathrm{F}} \|\mathbf{x}\|_{\mathrm{F}},$$

where inequality $(i)$ holds because $\|\mathbf{AB}\|_{\mathrm{F}} \leq \|\mathbf{A}\| \|\mathbf{B}\|_{\mathrm{F}}$ for any matrices $\mathbf{A}, \mathbf{B}$ with appropriate dimensions. $\qquad\square$

**Lemma 38** (The t-product layer is Lipschitz continuous). *Suppose the activation function is $L_\sigma$-Lipschitz, then a layer of t-product layer $h(\underline{\mathbf{x}}) = \sigma(\underline{\mathbf{W}} *_M \underline{\mathbf{x}})$ is at most $L_\sigma \|\underline{\mathbf{W}}\|_{\mathrm{F}}$-Lipschitz.*

*Proof.* According to the Lipschitzness of the activation function, we have

$$\begin{aligned}
\|h(\underline{\mathbf{x}}_1) - h(\underline{\mathbf{x}}_2)\|_{\mathrm{F}} &= \|\sigma(\underline{\mathbf{W}} *_M \underline{\mathbf{x}}_1) - \sigma(\underline{\mathbf{W}} *_M \underline{\mathbf{x}}_2)\|_{\mathrm{F}} \\
&\leq L_\sigma \|\underline{\mathbf{W}} *_M \underline{\mathbf{x}}_1 - \underline{\mathbf{W}} *_M \underline{\mathbf{x}}_2\|_{\mathrm{F}} \\
&= L_\sigma \|\underline{\mathbf{W}} *_M (\underline{\mathbf{x}}_1 - \underline{\mathbf{x}}_2)\|_{\mathrm{F}} \\
&= L_\sigma \|\underline{\mathbf{W}}\|_{\mathrm{F}} \|\underline{\mathbf{x}}_1 - \underline{\mathbf{x}}_2\|_{\mathrm{F}}.
\end{aligned}$$

$\qquad\square$

We then present Lemma 39 and Lemma 40 which are used in upper bounding the input and output of t-NNs in adversarial settings.

**Lemma 39.** *Given a fixed example $\underline{\mathbf{x}} \in \mathbb{R}^{d \times 1 \times c}$, if an adversary $\underline{\mathbf{x}}'$ satisfies $R_{\mathrm{a}}(\underline{\mathbf{x}} - \underline{\mathbf{x}}') \leq \xi$, then it holds that*

$$\|\underline{\mathbf{x}}'\|_{\mathrm{F}} \leq B_x + \xi \mathsf{C}_{R_{\mathrm{a}}}.$$

**Lemma 40.** *The $L_\infty$-norm of any $f(\underline{\mathbf{x}}; \underline{\mathbf{W}}) \in \mathfrak{F}^{\mathrm{adv}}$ defined on the set of input examples $\mathcal{X}$ is upper bounded by*

$$\sup_{\tilde{f} \in \mathfrak{F}^{\mathrm{adv}}} \left\|\tilde{f}\right\|_{L_\infty} \leq B_{\tilde{f}} := B_{\underline{\mathbf{W}}} B_{x, R_{\mathrm{a}}, \xi}. \tag{71}$$

*The diameter of $\mathfrak{F}^{\mathrm{adv}}$ is can be upper bounded as follows*

$$D_{\tilde{f}} := 2 \sup_{\tilde{f} \in \mathfrak{F}^{\mathrm{adv}}} \left\|\tilde{f}\right\|_S \leq 2 B_{\underline{\mathbf{W}}} B_{x, R_{\mathrm{a}}, \xi}. \tag{72}$$

*Proof.* For any $f \in \mathfrak{F}$, given an example $(\underline{\mathbf{x}}, y) \in \mathcal{X} \times \{\pm 1\}$, let $\underline{\mathbf{x}}^* \in \arginf_{R_{\mathrm{a}}(\underline{\mathbf{x}} - \underline{\mathbf{x}}') \leq \xi} y f(\underline{\mathbf{x}}')$ be one adversarial example. Then, we have

$$\begin{aligned}
|\tilde{f}(\underline{\mathbf{x}}, y)| &= |\inf_{R_{\mathrm{a}}(\underline{\mathbf{x}} - \underline{\mathbf{x}}') \leq \xi} y_i f(\underline{\mathbf{x}}')| \\
&= |f(\underline{\mathbf{x}}^*)| \\
&= |\mathbf{w}^\top \mathrm{vec}\left(\mathbf{h}^{(L)}(\underline{\mathbf{x}}^*)\right)| \\
&\leq \|\mathbf{w}\| \left\|\sigma(\underline{\mathbf{W}}^{(L)} *_M \mathbf{h}^{(L-1)}(\underline{\mathbf{x}}_i^*))\right\|_{\mathrm{F}} \\
&= \|\mathbf{w}\| \left\|\sigma(\underline{\mathbf{W}}^{(L)} *_M \mathbf{h}^{(L-1)}(\underline{\mathbf{x}}^*)) - \sigma(\mathbf{0})\right\|_{\mathrm{F}} \\
&\leq \|\mathbf{w}\| \left\|\underline{\mathbf{W}}^{(L)} *_M \mathbf{h}^{(L-1)}(\underline{\mathbf{x}}^*)\right\|_{\mathrm{F}} \\
&\leq \|\mathbf{w}\| \left\|\underline{\mathbf{W}}^{(L)}\right\|_{\mathrm{F}} \left\|\mathbf{h}^{(L-1)}(\underline{\mathbf{x}}^*)\right\|_{\mathrm{F}} \\
&\leq \cdots \\
&\leq \mathbf{w} \prod_{l=1}^{L} B_l \left\|\mathbf{h}^{(0)}(\underline{\mathbf{x}}^*)\right\|_{\mathrm{F}} \\
&\leq B_{\mathbf{w}} \prod_{l=1}^{L} B_l B_{x, R_{\mathrm{a}}, \xi} \\
&=: B_{\tilde{f}},
\end{aligned}$$

which also implies

$$D_{\tilde{f}} = 2 \max_{\tilde{f} \in \mathfrak{F}^{\mathrm{adv}}} \left\| \tilde{f} \right\|_S \le 2B_{\mathbf{w}} \prod_{l=1}^{L} B_l B_{x,R_{\mathrm{a}},\xi}.$$

$\square$

Lemma 41 helps upper bounding the F-norm of residuals after low-tubal-rank approximation of the weight tensors of t-NNs under Assumption 13.

**Lemma 41.** *Given constants $a > 0, \alpha > 1$, suppose a sequence $\{z_j\}_{j=1}^{\infty}$ satisfying polynomial decay $z_j \le aj^{-\alpha}$, then for any positive integer $n$, we have*

$$\sum_{j>n} z_j \le \frac{an^{1-\alpha}}{\alpha - 1}.$$

*Proof.* We compute the sum of the sequence using integration as follows:

$$\sum_{j>n} z_j \le \int_n^{\infty} at^{-\alpha}\mathrm{d}t = \frac{a}{\alpha - 1}t^{1-\alpha}\Big|_n^{\infty} \le \frac{an^{1-\alpha}}{\alpha - 1}.$$

$\square$

