# OpenReview forum: "Transformed Low-Rank Parameterization Can Help Robust Generalization for Tensor Neural Networks"
_NeurIPS.cc/2023/Conference — NeurIPS 2023 poster_

### Official Review · Reviewer_pKvo · 2023-06-29

**Soundness:** 4 excellent
**Presentation:** 4 excellent
**Contribution:** 3 good
**Rating:** 7
**Confidence:** 2

**Summary:**

The paper provides a thorough investigation of the generalization behavior of t-NNs for the first time, which closes the gap between the practical success of t-NNs and their theoretical analysis. To be specific, the authors derived the upper bounds for the generalization gaps. The authors also propose that compressing t-NNs with a transformed low-rank structure will result in more efficient adversarial training and tighter bounds. In addition to that, the authors show that adversarial training with GF in highly over-parameterized settings results in t-NNs with approximately transformed low-rank weights, and derive the sharp adversarial generalization bounds in this scenario as well.

**Strengths:**

- The paper provides a thorough investigation of the generalization behavior of t-NNs for the first time.
- The authors provide the bound for t-NNs generalization gaps with firm proof.
- The authors further explore the generalization bound of t-NNs with a transformed low-rank structure and show that there exists a training framework that will result in approximately transformed low-rank weights.
- The work is well motivated and with firm theoretical evaluation.

**Weaknesses:**

- Only theoretical evaluation is provided. Some empirical evaluations regarding the approximately transformed low-rank weights are expected.

**Questions:**

- Instead of using adversarial training with GF in highly over-parameterized settings to train a approximately transformed low-rank t-NN, is it possible to apply some extra regularizations in training to achieve a better low-rank transformation, for example, LoRA?

**Limitations:**

The authors have adequately addressed the limitations.

---

> ### Author Rebuttal · Authors · 2023-08-09
>
> We are truly grateful for your favorable evaluation of our work. Your recognition of our strengths, including the comprehensive investigation into t-NNs' generalization behavior, rigorous derivation of generalization bounds, and exploration of transformed low-rank structures, reinforces our commitment to unleash the potential of t-NNs for machine learning.
> We would also like to take this opportunity to address the weakness you pointed out and provide answers to the question you raised.
>
> **Weakness:**
> *Only theoretical evaluation is provided. Some empirical evaluations regarding the approximately transformed low-rank weights are expected.*
>
> **Response:**
> Thank you very much for your thoughtful suggestion.
> While our paper primarily emphasizes theoretical findings, we recognize and appreciate the importance of incorporating numerical evidence to reinforce our conclusions. To that end, we have actively undertaken numerical evaluations, particularly utilizing the effective rank of the M-block-matrix of a weight tensor as a metric for approximate low-rankness transformation in Experiment II. Please review the preliminary numerical assessments presented in the rebuttal section titled "Contributions and Numerical Evaluations," along with the accompanying PDF file.
>
> **Question:**
> *Instead of using adversarial training with GF in highly over-parameterized settings to train a approximately transformed low-rank t-NN, is it possible to apply some extra regularizations in training to achieve a better low-rank transformation, for example, LoRA?*
>
> **Response:**
> Yes, it is possible to apply additional regularizations during training to achieve a better low-rank representation in t-NNs. Instead of relying solely on adversarial training with gradient flow in highly over-parameterized settings, these extra regularizations can potentially promote and enforce low-rankness in the network.
>
> To validate the concern regarding the addition of an extra regularization term, we performed a preliminary experiment. In this experiment, we incorporated the tensor nuclear norm (Ref. [45] in our paper) as an explicit regularizer to induce low-rankness in the transformed domain. Specifically, we add the tensor nuclear norm regularization to the t-NN with three t-product layer (D=128) in Experiment II with a regularization parameter 0.01, and keep the other settings the same as Experiment II. We explore how the effective ranks of tensor weights evolve with the epoch number with/without tubal nuclear norm regularization.
>
> The initial experimental results are depicted in Figure 8 of the attached PDF in "Contributions and Numerical Evaluations". According to the results Figure 8, it becomes evident that the introduction of the explicit low-rank regularizer significantly enforced low-rankness in the transform domain of the weight tensors, thereby confirming the suggestion.

---

> > ### Comment · Reviewer_pKvo · 2023-08-15
> >
> > The authors have addressed my concerns. Please consider adding the experiment part to the final version of the paper. I will keep my score unchanged.

---

> > > ### Author Response · Authors · 2023-08-15
> > >
> > > Thank you for recognizing our responses. We will incorporate the experimental results in the final version as per your suggestion.

---

### Official Review · Reviewer_83W1 · 2023-07-05

**Soundness:** 3 good
**Presentation:** 3 good
**Contribution:** 3 good
**Rating:** 7
**Confidence:** 2

**Summary:**



The paper "Transformed Low-Rank Parameterization Can Help Robust Generalization for Tensor Neural Networks" considers neural networks with t-product layers, that encode the weights in tensor format equipped with the t-product as the corresponding tensor operation and ReLU activations.
The paper is an analytical work that derives upper bounds for the generalization error in t-NNs, using norm bound estimates. Further, the influence of low-rankedness of the Tensor structures on robust generalization is inspected in the setting of exact low-rankedness and  approximately transformed low-rankedness. Results show that low-rank t-NNs improve robust generalization properties. In particular, the Adverserial generalization bound for low-rank tNNs is shown to scale with the tubal rank of the tensors, instead of their dimension as in the full-rank setting.
Additional findings show that in an overparametrized setting t-NNs yield approximately low-rank weights during adversarial learning.

**Strengths:**

- Although this is a rather technical paper, the authors manage to present the findings comprehensively by rigorously introducing the relevant concepts with a precise notation.
- An extensive Appendix is provided to give an introduction to t-SVDs, and overview of previous work and technical details.
- The theoretical results of the work are discussed and compared with previous works.

**Weaknesses:**

- No code and no numerical examples are provided to support the presented results.

Although this is a rather theoretical work, a small test case can improve the presentation of this paper tremendously.
I think this is a strong paper that gives a multitude of insights.

A brief explanation of the intuition behind t-NN layers would also be helpful.

**Questions:**


- What is the intuition of the transformed t-product, i.e. what is the reason that the M matrix is needed? What is the special case of M being the identiy?
- Theorem 6: Can this statement be related to "standard" matrix valued neural networks with low-rank factorizations?
- Theorem 10: Are you able to validate the statement in a numerical example. It would be interesting to see, how the rank of a layer changes as J gets larger.

**Limitations:**

The authors discuss the limitations of their work, which manifest in rather loose bounds on the generalization error. They provide a suggestion on how to mitigate this issue in future work.

---

> ### Author Rebuttal · Authors · 2023-08-09
>
> We wish to express our appreciation for your positive and thorough evaluation of our technical paper. Your acknowledgment of our dedication to presenting the research findings with rigor, the incorporation of an extensive appendix, and the thoughtful comparison with existing works has provided us with profound encouragement.
> We are devoted to unlocking the potential of t-NNs for machine learning and are fully dedicated to rectifying the weaknesses and addressing the questions you have pointed out.
>
> **Weakness 1** (No code or numerical examples).
>
> **Response:**  Please refer to the initial numerical assessments in the rebuttal session titled "Contributions and Numerical Evaluations." We'll share the code after a thorough numerical evaluation.
>
> **Weakness 2** (A small test case to improve the presentation).
>
> **Response:** We acknowledge the importance of incorporating a practical example and are exploring the potential to include a relevant illustration that complements our theoretical results.
>
> **Weakness 3** (Intuition behind t-NN layers)*.*
>
> **Response:**  The intuition behind t-NN layers lies in their ability to handle multi-channel data more efficiently compared to traditional deep learning layers. Unlike standard fully connected layers that collapse multi-dimensional data into vectors or matrices, t-NNs maintain the multi-channel structure of the data as tensors. This unique property of t-NNs allows them to capture complex correlations and dependencies in the data more effectively. By preserving the inherent structure of tensors, t-NN layers can better grasp the global context and interrelationships within the data, leading to improved performance in various applications ([25,30,31,40]).
>
> Additionally, t-NNs leverage specialized mathematical operations like the t-product and t-SVD in the transformed domain [14], which further enhances their ability to process complex data efficiently. The transformed domain allows t-NNs to focus on essential features and patterns in the data while disregarding noise and irrelevant local details, resulting in improved stability and robustness, especially in the face of adversarial attacks [40].
>
> **Question 1**
>
> **Q1.1** (Intuition of the transformed t-product, i.e, the need of matrix M)
>
> **Response:**  The introduction of the transformed domain in the t-product is a pivotal aspect of t-NNs, offering distinct advantages in handling complex data.
> By using the transform matrix M, t-NNs can map tensors to various domains, like Fourier or DCT. These domains offer unique data representations, aiding t-NNs in leveraging frequency domain low-rankness and global feature attention.
>
> Operating in the transformed domain enables t-NNs to capture global features and structures in high-dimensional data, enhancing their performance in various applications. This domain also helps filter noise and irrelevant details, making t-NNs more robust against adversarial attacks. Global feature attention mechanisms further improve t-NNs' pattern identification.
>
> Moreover, the transformed domain reduces parameters, enhancing computational efficiency for high-dimensional data. In conclusion, t-NNs benefit from transformed domains to model complex data effectively, enhance robustness, and improve efficiency, making them promising in machine learning.
>
> **Q1.2**  (M being the identity).
>
> **Response:** Indeed, when the transform matrix M is an identity matrix, the transformed domain becomes equivalent to the original domain, and the t-product reduces to the standard parallel matrix multiplication of the frontal-slices of factor tensors. In this scenario, we lose the opportunity to leverage the advantages offered by the transformed domain representations, which is a situation to be avoided in t-NNs. Thus, we usually avoid an identity transform matrix M to ensure that t-NNs can exploit the benefits of the transformed domain representations for enhanced performance, robustness, and efficiency in modeling complex data.
>
> **Question 2** (Theorem 6 related to "standard" matrix neural networks with low-rank factorizations).
>
> **Response:** Yes!  When the channel number c=1, the derived bound exactly recovers the corresponding bounds for classical fully connected neural networks. This result demonstrates the compatibility of our approach with traditional neural networks and highlights the versatility of our low-rank parameterization in various settings.
>
> Furthermore, our findings reveal that the adversarial generalization bound under low-rank parameterization exhibits a superior scaling behavior compared to standard (full-rank) neural networks. The improved scaling implies that networks with low-rank parameterization may require a smaller number of training examples to achieve the desired accuracy, leading to potential benefits such as reduced energy consumption, storage requirements, and computational cost.
>
> **Question 3** (Numerical evaluation for Theorem 10).
>
> **Response:** Yes, we have actively begun the numerical evaluations for Theorem 10. We totally agree that it is interesting to see how the rank of a layer changes as J gets larger, and are more than willing to conduct such numerical evaluations. However, we would like to acknowledge that our empirical findings indicate a substantial time investment. Even for a t-NN encompassing merely three t-product layers, an extensive commitment ranging from about 500 to 2,000 hours on our available computational resources is required to comprehensively study the implicit bias phenomena. As such, we have embarked on initial numerical experiments specifically concerning three t-product layers to explore the implicit bias towards transformed low-rankness. You can find preliminary numerical results in "Experiment II" in "Contributions and Numerical Evaluations."

---

> ### Author Response · Authors · 2023-08-15
>
> We sincerely thank you for recognizing and appreciating our work. Your valuable feedback and suggestions have been noted, and we have tried to address all the concerns you've pointed out. As we update our final version, we're guided by your recommendations. We genuinely hope our explanations align with your expectations. If any details need further clarity, please let us know.

---

> > ### Comment · Reviewer_83W1 · 2023-08-16
> >
> > I thank the authors for providing extensive feedback and addressing open questions. Thank you also for including the numerical experiments in the final paper. I will keep my score unchanged to 7 (accept).

---

> > > ### Author Response · Authors · 2023-08-20
> > >
> > > We genuinely appreciate your recognition of our efforts to address the questions and incorporate the numerical experiments. Your feedback is invaluable to us, and we're grateful for it.

---

### Official Review · Reviewer_yC9Y · 2023-07-26

**Soundness:** 2 fair
**Presentation:** 1 poor
**Contribution:** 1 poor
**Rating:** 4
**Confidence:** 1

**Summary:**

The paper analyzes the generalization ability of t-product layers (t-NNs) by deriving upper bounds on generalization error in standard and adversarial settings.

**Strengths:**

The paper advances the theoretical understanding of t-NNs and derives their generalization behavior in two practical settings.

**Weaknesses:**

The paper focuses on and is written about t-product layers, referenced as [7, 28] in the first lines of the introduction. The first reference is a patent, and the second is a paper from 2018 (from the same authors as the patent), introducing the concept more or less academically. The submission assumes the reader's familiarity with these concepts right from the start, which is a mistake, and does not attempt to clarify or recap the importance of these concepts up until Sec. 3. From this standpoint, the writing should be improved.

The importance of the usage of t-product layers in the Deep Learning community is not supported with sufficient evidence, leaving the question of the practical utility of the studied framework unanswered.

**Questions:**

Is Definition-1 (t-product) equivalent to the batch matrix-multiplication operation, where `c` corresponds to the batch dimension?
Is Definition-3 (t-SVD) equivalent to the SVD of a block-diagonal representation from Definition 2?
Why do these concepts need special terms? What is the advantage of such complex structures over the traditional deep learning layers?


**Limitations:**

Discussed in the Conclusion.

---

> ### Author Rebuttal · Authors · 2023-08-09
>
> **Weakness 1** (Writing).
>
> **Response:**  Thanks for your comment on the writing.
> Firstly, we would like to address the need for references to [7] and [28].
> - We appreciate your highlighting the reference to patent [7] for the scientific rigor and integrity of our work.  However, patent [7] (granted in Dec. 2022) has garnered a certain level of attention, evident from citations in two IEEE T-GARS papers (Google Scholar) and four patents (Google Patents). Additionally, we draw attention to instances where scientific papers [R2.1] have cited important patents. Therefore, we respectfully assert that citing [7] does not have any negative consequences.
> - The arXiv paper [28] introduced t-NNs and presented noteworthy advancements on MNIST and CIFAR-10 datasets, outperforming standard matrix networks. It furnishes compelling evidence of the potential of t-NNs in real-world benchmark tasks. As a result, t-NNs have found extensive applications in machine learning over recent years, as evidenced by [25, 30, 31, 40]. Thus, [28] should also be cited.
>
> Secondly, we apologize for any confusion and wish to clarify that we "expect" rather than "strictly assume" readers' familiarity with t-product layers in the introduction. As aptly noted by Reviewer 3 in the Strengths, "Although this is a rather technical paper, the authors manage to present the findings comprehensively by rigorously introducing the relevant concepts with precise notation." We choose to briefly introduce the significance and recent advancements in t-NNs rather than the basic concepts within the introduction, due to the complexity of technical terms associated with t-NNs. To provide a smoother understanding for readers, we then revisit and recap the foundational concepts in subsequent sections.
>
> However, we totally understand that this may delay readers' eager comprehension of the concepts of t-NNs. To mitigate this concern, we will revise the introduction to provide a clearer and more comprehensive explanation of t-product layers, including the basic ideas on a conceptual level. Additionally, we will clarify and recap necessary notions of t-NNs in Section 2 rather than Section 3.
>
> [R2.1] Florsheim E B, et al. Immune sensing of food allergens promotes avoidance behaviour. Nature, 2023.
>
> **Weakness 2** (Practical utility of the studied framework).
>
> **Response:**  We respectfully disagree with this comment.  The importance of incorporating t-product layers in the deep learning community is substantiated by substantial evidence, showcasing their promising advantages across a range of domains. These advantages are clearly demonstrated in applications such as graph data augmentation, dynamic graph learning, code semantics embedding, seismic data processing,  and more [25, 30, 31, 40, R2.2].
>
> We reemphasize that T-NNs show great potential in advancing machine learning. They excel in capturing global features and complex patterns through transformed domain representations. The integration of low-rank structures, along with global feature attention, augments resilience and efficiency. Moreover, the utilization of low-rank decomposition streamlines the optimization of parameters, thus reducing storage requirements and enhancing computation speed.
>
> [R2.2] Yang J, et al. Toward interpretable graph tensor convolution neural network for code semantics embedding. ACM Transactions on Software Engineering and Methodology, 2023.
>
> **Question 1** (T-product vs batch matrix-multiplication operation).
>
> **Response:**  No, the t-product is not a direct equivalent of batch matrix multiplication in the original domain according to Def. 1. Instead, it corresponds to batch matrix multiplication in the transformed domain. Additionally, this unique property of the t-product allows for efficient and effective data and parameter processing by leveraging the advantages of both batch matrix multiplication and the transformed domain. By performing computations in the transformed domain, the t-product enables more complex and intricate operations to be carried out, leading to enhanced performance and versatility in various applications.
>
> **Question 2** (T-SVD vs SVD of a block-diagonal representation from Def. 2).
>
> **Response:**  Yes. Our Eq. (3) reveals that t-SVD is equivalent to the SVD of a block-diagonal representation per Def. 2.
>
> **Question 3** (Special terms of concepts).
>
> **Response:**  The t-product and t-SVD are distinct terms that differentiate them from traditional matrix operations and factorizations. They are vital in t-NNs due to the multi-channel data structure, requiring new tensor-specific mathematical operations. These concepts efficiently manipulate tensors while maintaining their structure, enhancing global correlation capture. The t-product and t-SVD enable specialized tensor operations, addressing tensor-related challenges and unlocking t-NNs' potential for better performance in processing multi-dimensional data.
>
> **Question 4** (The advantage of such complex structures over the traditional deep learning layers).
>
> **Response:**  T-NNs excel over traditional deep learning layers in handling complex multi-channel data. While conventional FC layers are limited to vectors and matrices, t-NNs utilize specialized tensor-centric operations (like t-product and t-SVD) to efficiently capture intricate correlations and preserve multi-channel structures. This approach allows t-NNs to better understand global patterns within the data, leading to enhanced performance across various applications.
> Additionally, t-NNs possess other advantageous features, including global feature attention mechanisms, robustness against adversarial attacks, and parameter compression, further enhancing their capabilities. Leveraging these complex structures enables t-NNs to process high-dimensional multi-channel data more effectively, making them a promising choice for addressing the challenges posed by modern data types and advancing model performance and generalization.

---

> ### Author Response · Authors · 2023-08-15
>
> Thank you for taking the time to review our manuscript and for sharing your feedback. We've carefully addressed the points you brought up and will make necessary adjustments to our paper. We hope our responses meet your expectations. Should you have further questions or need clarification on any matter, please don't hesitate to inform us.

---

> > ### Comment · Reviewer_yC9Y · 2023-08-20
> > **Acknowledgement**
> >
> > I thank the authors for elaborating on the points raised in the initial feedback. I will keep my score.

---

### Official Review · Reviewer_EpHo · 2023-07-30

**Soundness:** 3 good
**Presentation:** 3 good
**Contribution:** 2 fair
**Rating:** 5
**Confidence:** 4

**Summary:**

The paper studies the generalization error in the standard and the adversarial settings of t-NNs, neural networks with layers and features parametrized via t-vectors and t-products.


**Strengths:**

- The provided generalization bounds for t-NNs show that neural networks with low-rank parameters have the potential for a better generalization.
- This type of generalization analysis for tNNs seems new

**Weaknesses:**

1. Although I appreciate this type of analysis for t-NN did not appear before, the t-product is essentially a composition of operations obtained through a suitable block reshaping of the parameter kernel, thus extending the results from standard fully connected linear layers seems like a natural and relatively direct result. In light of this, the novelty of this paper seems limited

2. I would have liked to see a few numerical experiments validating the theoretical findings and the tightness of upper bounds -- also to address the claimed limitations at the end of the paper


**Questions:**

- the authors should clarify in the main text how the obtained results differentiate with respect the corresponding bounds for linear fully connected nets (for example the bound in Thm 6 and Thm 10) and should highlight what the main differences in the proof/analysis of these bounds are with respect to the linear case

- I think the rank-adaptive approach for selecting the ranks of the network mentioned on line 186 deserves more attention - in particular, i do not really understand in what way rank-adaptivity can be exploited in the analysis provided by Thm 6. See also [x],[y] for alternative recent rank-adaptive strategies for linear layers.

- The fact that adversarial training leads to low-rank weights and vice-versa reducing parameters has a positive/negative effect of robustness (as mentioned at line 250) is subject to ongoing active research and there is a variety of different observations/results on this subject (see eg [a-d]). I would rephrase here in the light of these papers. In any case, these empirical studies are done for networks in the standard form not for t-NNs. Using that evidence to comment on findings for t-NN seems to support my weakness no. 1

[x] H. Yang, M. Tang, W. Wen, F. Yan, D. Hu, A. Li, H. Li, and Y. Chen. Learning low-rank deep neural networks via singular vector orthogonality regularization and singular value sparsification (IEEE/CVF 2020)
[y] S. Schotthöfer, E. Zangrando, J. Kusch, G. Ceruti, and F. Tudisco. Low-rank lottery tickets: finding efficient low-rank neural networks via matrix differential equations (NeurIPS 2022)
[a] https://openreview.net/pdf?id=SJGrAisIz
[b] https://arxiv.org/abs/1912.02386
[c] https://arxiv.org/abs/2306.01485
[d] https://link.springer.com/content/pdf/10.1007/s10994-021-06049-9.pdf?pdf=button

**Limitations:**

The authors have added a "limitations" paragraph stating that the obtained generalization bounds may be somewhat conservative, which I agree with. I am not sure I understand the proposed solution to the limitations though.

---

> ### Author Rebuttal · Authors · 2023-08-09
>
> **W1** (Novelty).
>
> **Re:** We appreciate the reviewer for acknowledging the novelty of our analysis for t-NNs. We believe that our analysis can bridge the gap from the existing applications to a more rigorous understanding of t-NNs in machine learning. Notably, t-product's distinction from linear products in traditional FNNs underscores their structural and algebraic dissimilarity. This forms the basis for our unique approach and sets our analysis apart from FNNs.
>
> Specifically, the unique structure of t-NNs, including their t-product and transformed low-rankness, precludes the direct adaptation of generalization bound proofs (e.g., Lemma 3 and Thm 6) from [5, 42, 43], designed for FNNs. To address this, we derive several lemmas tailored to the special characteristics of t-NNs. For instance, Lemma 3 involves a t-product-based "peeling argument," implemented by reshaping weight tensors using Lemma 16. This sets the foundation for the crucial Lemma 17, supporting the argument. Furthermore, the proofs of Thm 6 require control over t-product layer outputs and the covering number of low-tubal-rank tensors, achieved through Lemmas 37 and 38, and Lemma 33, respectively.
>
> Likewise, proofs for adversarial generalization bounds for t-NNs with approximate transformed low-rank parameterization can't straightforwardly stem from the analysis of [34] for standard generalization of FNNs. Challenges arise due to capacity control mismatches under adversarial attacks on t-NNs. To address this, our proof involves introducing $(\delta, r)$-approximate low-tubal-rank parameterization for capacity measurement, controlling localized Rademacher complexity of the Minkowski difference of adversarial-counter parts of two t-NN classes for Thm 12, and employing low-tubal-rank approximations for precise capacity control in Thm 14.
>
> **W2** (Numerical experiments).
>
> **Re:** Please review the initial numerical evaluations in rebuttal "Contributions and Numerical Evaluations".
>
> **Q1** (Differences of results, proof, analysis of the bounds from FNNs).
>
> **Re:**
> Our theoretical results significantly differ from corresponding FNN bounds:
>
> - The standard (resp. adversarial) generalization bound in Lemma 3 (resp. Thm 5) differs from the counterpart for FNNs [5] (resp. [42,43]) due to the role of the channel number $\mathsf{c}$ in t-NNs' parameter complexity. Additionally, Thm 5's bound covers broader adversary classes compared to $l_p$-attacks in [42, 43].
> - Thm 6's notable divergence from [42, 43] lies in its inclusion of weight low-rankness in the adversarial generalization bound. This aspect hints at the potential for enhanced robustness in generalization.
> - We introduce a novel concept in our analysis of GF for adversarial training (AT): the implicit bias of approximately transformed low-rankness in t-NNs. This significantly extends and enhances the findings from implicit bias in AT for FNNs in [22], which concentrates solely on convergence to a KKT point with exponential loss. Our research goes further by investigating approximately transformed low-rankness for t-NNs with broader loss functions in AT.
> - The key distinction in the adversarial generalization bounds of Section 4.3 from non-adversarial bounds for FNNs in [34] is the inclusion of the localized Rademacher complexity of the Minkowski difference between adversarial-counter parts of approximately and exactly low-tubal-rank t-NNs in Thm 12.
>
> Our analysis and proofs diverge from linear cases:
>
> - Unlike the analysis for FNNs' generalization bounds based on Rademacher complexity in [5, 42, 43], we derive specific lemmas for standard and adversarial generalization bounds in t-NNs. This is due to the unique structure of (low-rank) t-product layers. We reformulate the t-product through an operator-like expression in Lemma 16, paving the way for pivotal Lemma 17, supporting the t-product-based "peeling argument." Additionally, we introduce Lemmas 37, 38, and Lemma 33 to handle t-product layer output norms and covering low-tubal-rank tensors.
> - Proving the implicit bias of GF for AT of t-NNs in Thm 10, specifically the approximately transformed low-rankness, is nontrivial in comparison to the proof in [22] for the implicit bias of adversarial training for FNNs. As we consider more general loss functions for t-NNs in contrast to the exponential loss for FNNs in [22], we first derive a more general convergence result to the direction of a KKT point for t-NNs Lemma 9, and then goes deeper by using a constructive approach to establish the approximately transformed low-rankness in the proof of Thm 10.
> - Differing from [34] which focuses on standard FNN generalization, our approach delves into t-NNs' adversarial generalization. We achieve this by introducing the $(\delta, r)$-parameterization, bounding localized Rademacher complexity for a Minkowski set in adversarial settings, and using low-tubal-rank approximations for tensor weights.
>
> **Q2** (Rank adaptivity for Thm 6).
>
> **Re:** The analysis of Thm 6 doesn't require rank adaptivity due to explicit constraints on weight tensor ranks. Yet, we appreciate exploring alternative rank-adaptive methods like [x, y] due to the potential value for practical adversarial training of t-NNs.
>
> **Q3** (Empirical studies on robustness and low-rankness for standard NNs rather than t-NNs).
>
> **Re:** We recognize that the relationship between weight low-rankness and enhanced robustness is actively researched, yielding diverse observations and results. Direct comparisons should be approached cautiously, as these studies primarily concern standard network architectures and may not seamlessly apply to t-NNs. We'll incorporate insights from [a-d] into our discussion at line 250, highlighting the distinctions between standard networks and t-NNs with careful references. Additionally, we're planning to introduce new experiments (e.g., Experiment II in "Contributions and Numerical Evaluations") that employ t-NNs' own results to substantiate our claims.

---

> ### Author Response · Authors · 2023-08-15
>
> Thank you for the valuable comments and suggestions. We have tried to address each of your concerns in detail and will make revisions to the final version based on your recommendations. May we ask if our responses have adequately addressed all your queries? If there are any points that require further clarification or explanation from us, please let us know.

---

> > ### Comment · Reviewer_EpHo · 2023-08-20
> >
> > Dear authors, thank you for your response. I wish to maintain my score.

---

### Author Rebuttal · Authors · 2023-08-09

## **Contributions and Numerical Evaluations**

We thank all the reviewers for their valuable comments and suggestions. We first clarify our contributions and novelty again, and then report the initial numerical evaluations as suggested by Reviewers (R1, R3, R4).

---

**Contributions & Novelty**

With the rise of t-NNs  in machine learning, our paper introduces a pioneering theoretical framework for t-NNs, enabling us to comprehend both their standard and robust generalization behaviors for the first time. The main contribution and novelty of this paper lie in an in-depth theoretical analysis of t-NNs, revealing key properties and robustness of this specialized type of neural networks.

- *Theoretical Characterization of Generalization Behavior:* Through the introduction of lemmas specifically designed for t-NNs, this paper establishes upper bounds on the generalization error for t-NNs in both standard and adversarial contexts.
- *Robustness Analysis of t-NNs:* The analysis shows that t-NNs with exactly and approximately transformed low-rank weights exhibit lower adversarial generalization bounds, highlighting the benefits of transformed low-rank weights in improving robustness and efficiency.
- *Impact of Adversarial Learning on Weight Tensors:* The investigation reveals a novel observation that weight tensors in over-parameterized t-NNs tend to exhibit an approximation of transformed low-rankness.
- *Influence of Transformed Low-rank Weights on Robust Generalization:* Through the precise derivation of adversarial generalization bounds, the importance of integrating transformed low-rank weights is emphasized as a means to strengthen the robustness of t-NNs.

---

**Experiment I**

To validate the generalization bound in Thm 6, we have conducted experiments on the MNIST dataset to explore the relationship between adversarial generalization gaps (AGP), weight tensor low-rankness, and training sample size. We consider binary classification of 3 and 7,  with FSGM attacks of strength 20/255. The t-NN consists of three t-product layers and one FC layer, with weight tensor dimensions of 28×28×28 for $\underline{\textnormal{W}}^{(1)}$ to $\underline{\textnormal{W}}^{(3)}$, and 784 for the FC weight **w**. As an input to the t-NN, each MNIST image of size 28×28 is treated as a t-vector of 28×1×28.

Thm 6 emphasizes: (i) lower weight tensor rank leads to smaller AGP bound, and (ii) AGP bound diminishes at a rate of $1/\sqrt{N}$ as $N$ increases. We explored this by conducting experiments, controlling the upper bounds $\textnormal{r}$ of the tubal-rank to 4 and 28 for low and full tubal-rank cases, and systematically increasing the number of training samples.

Fig. 1 in the PDF presents initial results. The curves indicate that t-NNs with lower rank weight tensors have smaller robust generalization errors. Interestingly, the adversarial generalization errors seem to follow a linear relationship with $1/\sqrt{N}$, approximately validating the generalization error bound in Theorem 6 by approximating the scaling behavior of the empirical errors.

---

**Experiment II**

We carried out experiments to confirm two theoretical statements related to the analysis of GF-based adversarial training.

*Statement 2.1*  Thm 10 reveals that, under specific conditions, well-trained t-NNs with highly over-parameterized adversarial training using GF show nearly transformed low-rank parameters.

*Statement 2.2*  Lem 22 asserts that the empirical adversarial risk approaches zero, and the F-norm of the weights grows infinitely as \( t \) approaches infinity.

In continuation of Experiment I, we focus on binary classification on MNIST under FGSM attacks. The t-NN is structured with three t-product layers and one FC layer, with weight dimensions set to D×28×28 for $\underline{\textnormal{W}}^{(1)}$, D×D×28 for $\underline{\textnormal{W}}^{(2)}$ and $\underline{\textnormal{W}}^{(3)}$, and 28D for the FC weight $\textbf{w}$. Our experiments involve setting values of D to 128 and 256, respectively, and we track the effective rank of each weight tensor, the empirical adversarial risk, and the F-norm of the weights as the number of epochs progresses. Since implementing gradient flow with infinitely small step size is impractical in real experiments, we opt for SGD with a constant learning rate and batch-size of 80, following the setting on fully connected layers in [22].

Note that fully observing implicit bias generally requires 10,000 to 40,000 epochs as shown in [22] for FNNs. This would take around 500 to 2,000 hours on our devices. *Due to time constraints in the rebuttal phase, we can only offer preliminary experimental results as initial support for our research statements.*

For Statement 2.1, we present preliminary results illustrating the progression of the effective ranks of the M-block-diagonal matrix of tensor weights in Figs. 2 and 3 for the settings D=128 and D=256, respectively. Notably, these results show that the effective ranks decrease as more epochs are executed, thereby confirming the influence of implicit bias on transformed low-rankness, as described in Statement 2.1.

For Statement 2.2, we present initial numerical findings depicting the progress of the empirical adversarial risk and the F-norm of the weights in Figs. 4 and 6 for D=128, and Figs. 5 and 7 for D=256, respectively. Unfortunately, due to time limitations, the program was only capable of running for less than 1/10th of the expected epochs. Nevertheless, these preliminary results exhibit a consistent pattern with the theoretical descriptions outlined in Statement 2.2 and the numerical results reported in [22] for adversarial training and [23] for standard training of FNNs. Specifically, we observe a decreasing trend in the empirical risk function and an increasing trend in the weight tensor's F-norm, which align with the expected behavior based on our theoretical framework and corroborate the numerical results presented in [22, 23].

---

### Author Response · Authors · 2023-08-20

We extend our heartfelt gratitude to both the reviewers and the area chairs for the time and dedication given to our manuscript. Your feedback and insights are deeply valued and are essential to our work's refinement. As the author-reviewer discussion deadline approaches, should there be any additional concerns or clarifications needed, please do not hesitate to inform us. We are dedicated to enhancing our work in light of your feedback. Best regards.

---

### Decision · Program_Chairs · 2023-09-21

**Decision:**

Accept (poster)

**Comment:**

In recent applications, multi-channel learning has garnered significant attention, particularly concerning neural networks utilizing t-product layers (t-NNs). These t-NNs have demonstrated promising performance due to novel feature mapping in the transformed domain. This paper aims to develop theory for such t-NNs. The authors claim to do this by deriving upper bounds on the generalization error of t-NNs in both standard and adversarial settings. Their analysis reveals that t-NNs, when compressed with an exact transformed low-rank parameterization, can achieve tighter adversarial generalization bounds compared to non-compressed models. Furthermore, the paper establishes sharp adversarial generalization bounds for t-NNs employing approximately transformed low-rank weights. These findings emphasize the potential of transformed low-rank parameterization in improving the robust generalization of t-NNs, providing insights for future research and development in this domain.

The reviewers liked the theoretical results for t-NNs and derives the generalization behavior. They expressed some concerns about lack of code/empirical evaluations, clarity of writing, and significance of the contributions. These concerns seem to be alleviated for most of the reviewers by the authors response. Hence I recommend acceptance.